# Satellite repeat transcripts modulate heterochromatin condensates and safeguard chromosome stability in mouse embryonic stem cells

Clara Lopes Novo [1,2,3✉], Emily V. Wong[4], Colin Hockings[5], Chetan Poudel [5], Eleanor Sheekey [1], Meike Wiese [6], Hanneke Okkenhaug[7], Simon J. Boulton [3,8], Srinjan Basu [6], Simon Walker[7], Gabriele S. Kaminski Schierle [5], Geeta J. Narlikar [4] & Peter J. Rugg-Gunn [1,6✉]

Heterochromatin maintains genome integrity and function, and is organised into distinct nuclear domains. Some of these domains are proposed to form by phase separation through the accumulation of HP1α. Mouse heterochromatin contains noncoding major satellite repeats (MSR), which are highly transcribed in mouse embryonic stem cells (ESCs). Here, we report that MSR transcripts can drive the formation of HP1α droplets in vitro, and modulate heterochromatin into dynamic condensates in ESCs, contributing to the formation of large nuclear domains that are characteristic of pluripotent cells. Depleting MSR transcripts causes heterochromatin to transition into a more compact and static state. Unexpectedly, changing heterochromatin's biophysical properties has severe consequences for ESCs, including chromosome instability and mitotic defects. These findings uncover an essential role for MSR transcripts in modulating the organisation and properties of heterochromatin to preserve genome stability. They also provide insights into the processes that could regulate phase separation and the functional consequences of disrupting the properties of heterochromatin condensates.

[1] Epigenetics Programme, The Babraham Institute, Babraham Research Campus, Cambridge CB22 3AT, UK. [2] Tommy's National Miscarriage Research Centre at Imperial College London, London W12 0NN, UK. [3] The Francis Crick Institute, 1 Midland Road, London NW1 1AT, UK. [4] Department of Biochemistry and Biophysics, University of California San Francisco, San Francisco, CA, USA. [5] Department of Chemical Engineering and Biotechnology, University of Cambridge, Cambridge CB3 0AS, UK. [6] Wellcome – Medical Research Council Cambridge Stem Cell Institute, University of Cambridge, Cambridge CB2 1QR, UK. [7] Imaging Facility, The Babraham Institute, Babraham Research Campus, Cambridge CB22 3AT, UK. [8] Artios Pharma Ltd., B940, Babraham Research Campus, Cambridge CB22 3FH, UK. ✉email: clara.novo@gmail.com; peter.rugg-gunn@babraham.ac.uk

Functional compartmentalisation of the genome segregates repetitive, gene-poor regions into constitutive heterochromatin[1]. Establishing and maintaining the appropriate regulation of heterochromatin is essential for preserving nuclear architecture, genome stability and DNA repair, and for silencing transposon expression[2,3]. In interphase cells, constitutive heterochromatin from different chromosomes cluster in cytologically defined nuclear bodies called chromocenters. These structures are characterised by the presence of histone H3 lysine 9 trimethylation (H3K9me3) and of heterochromatin protein 1α (HP1α)[4–6]. In the context of purified components, phosphorylation or DNA binding drives soluble HP1α into phase-separated droplets[7,8]. Furthermore, earlier studies showed that human HP1 proteins display dynamics on the order of seconds within heterochromatin puncta[9,10]. These observations in cells support the possibility that chromocenters could form by phase separation through the localised accumulation of HP1α. A phase separation-based model has important implications for how heterochromatin is assembled and regulated. However, the underlying biological processes that could promote HP1α-mediated phase separation in vivo, and the functional consequences of disrupting the phase separation of heterochromatin domains, remain as yet unknown.

The regulation of constitutive heterochromatin is developmentally controlled[11] and this regulation is perturbed during cell stress and ageing[12]. In mouse, constitutive heterochromatin adopts an unusual configuration in both embryonic stem cells (ESCs) and early embryonic cells compared to most other cell types, as it has reduced levels of H3K9me3, dynamically binds heterochromatin-associated proteins, and has a less repressed heterochromatin structure[13–22]. Such distinct properties also extend to cytological differences, for example, constitutive heterochromatin is structured into much larger and fewer chromocenters in ESCs compared to most somatic cells[14,15,23,24]. How and why heterochromatin in ESCs adopts this unusual configuration is unknown, although one clue could be that ESCs transcribe high levels of noncoding major satellite repeat (MSR) elements, which constitute the predominant DNA sequences within chromocenters[13,15,19,25]. MSR transcripts remain close to their sites of transcription at chromocenters[19,26], and biochemical data indicate that MSR RNA can interact with heterochromatin-associated proteins, such as HP1α, SUV39H2, MeCP2 and SAFB[27–30]. Importantly, the control of MSR transcription is integrated into the transcription factor networks that sustain ESCs, which suggests that the regulation of heterochromatin in ESCs is an active process that is tightly controlled and coupled to cell state[15,19]. Evidence that MSR transcripts potentially have a functional role in ESCs comes from studies of early mouse development, which demonstrate that MSR RNA is required to establish chromocenter formation and for embryo development to proceed[31–34]. An important goal, therefore, is to understand the role that repetitive, noncoding RNAs, such as MSR transcripts, play in promoting the biophysical properties of heterochromatin formation and stability.

Here, we report that noncoding satellite RNA alters the physical properties and nuclear organisation of heterochromatin in mouse ESCs. We find that MSR transcripts regulate a permissive and dynamic environment within individual chromocenters and promote the formation of large heterochromatin foci that are a hallmark of pluripotent cells. We further show that MSR RNA can drive the formation of HP1α droplets in vitro and is required for HP1α organisation within heterochromatin domains in vivo. The reduction of MSR transcripts prompted chromocenter properties to transition to a less dynamic and more stable heterochromatin state, and triggered the reorganisation of chromocenters into more numerous, smaller and compact foci. This

aberrant chromocenter reorganisation led to rapid chromosome instability, as reflected by the appearance of chromosome end-fusions and pericentric gaps. These findings thus uncover an unexpected protective function for heterochromatin condensates in preserving genome stability.

## Results

**Constitutive heterochromatin forms liquid-like condensates in mouse embryonic stem cells.** Studies of HP1α behaviour in *Drosophila* and in mammalian cells have proposed that heterochromatin is possibly held within liquid-like, phase-separated compartments[7,8]. To define the biophysical properties of constitutive heterochromatin in mouse ESCs, we used time-lapse imaging to track the movement and dynamics of individual chromocenters in live cells. For that, we generated a stable ESC line that expressed a doxycycline-inducible, monomeric mClover fusion protein that has a transcription activator-like effector (TALE) targeting the MSR sequence (Fig. 1a, b)[35]. Fluorescent microscopy confirmed that the induced TALE-MSR-mClover signal localised to chromocenters throughout the cell-cycle (Supplementary Fig. 1a and Supplementary Movie 1), similar to previous observations[35]. As expected from prior studies[15,35], the targeting of TALE-MSR-mClover to MSR elements did not affect chromocenter organisation (Supplementary Fig. 1b, c). Importantly, our time-lapse imaging of TALE-MSR-mClover revealed that individual chromocenters separated and coalesced rapidly, with the movement of chromocenter extrusions occurring within minutes (Fig. 1c and Supplementary Movies 2a (unsegmented signal), 2b (segmented signal), 2c (unsegmented signal) and 2d (segmented signal) and 3 (cross section)). The ability to undergo such dynamic processes are features attributed to liquid-like membraneless condensates[36].

Liquid-like condensates are also characterised by the rapid, internal movement of molecules[37,38]. And so, to further assess the dynamics of molecules within chromocenters of ESCs, we used fluorescence recovery after photo-bleaching (FRAP) to measure the mobile (liquid) and immobile (static) TALE-MSR-mClover-bound chromatin fractions. Because TALEs can bind with strong affinity (low nM $K_d$ values) to their target DNA, both in vitro and in vivo[39,40], this assay allows us to measure the physical properties of chromocenters, including both the movement of heterochromatin and the dynamics of chromatin-associated proteins, and is unaffected by changes to the heterochromatin state. One chromocenter per nucleus was bleached and the recovery of the GFP signal in this region over time was quantified. Fluorescence quickly recovered at photo-bleached chromocenters, reaching 50% of the initial intensity within ~30 s after bleaching (Fig. 1d and Supplementary Movie 4), with comparable dynamics to other molecules that were previously reported to be within liquid-like condensates[41–43]. In addition, an immobile, stable component of ~25% remained after photo-bleaching (Fig. 1e), which is in line with measurements in other cell types[8,40]. The recovery could arise from the dissociation of bleached TALE-MSR-mClover molecules and their replacement with unbleached TALE-MSR-mClover molecules or from movement of the chromatin region itself. Compared to the time-scales that we observe, previous studies in ESCs have reported substantially slower recovery rates of bleached histones ($t_{1/2} = $ ~100 s) and faster recovery of soluble heterochromatin proteins, such as HP1α-GFP ($t_{1/2} = $ ~3.5 s)[14]. Thus, we favour the interpretation that this measures a combination of both replacement and movement. Based on these results, we conclude that DNA binding factors, such as TALE-MSR-mClover, bound to constitutive heterochromatin in ESCs display behaviours that

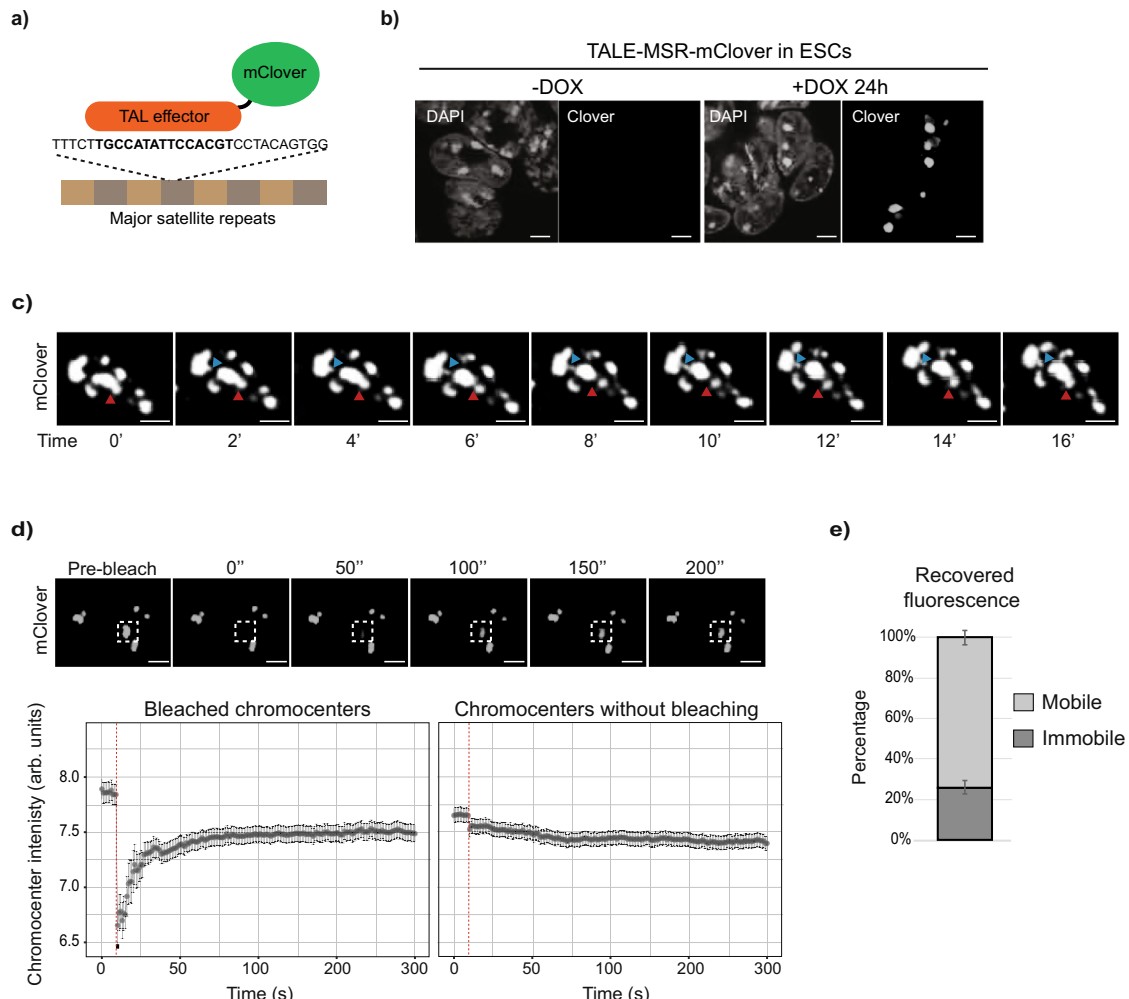

**Fig. 1 Constitutive heterochromatin forms liquid-like condensates in embryonic stem cells. a** Schematic showing the recruitment of doxycycline-inducible TALE-mClover to MSR DNA to enable live-cell visualisation of chromocenter dynamics. **b** Representative images from >10 independent experiments showing the TALE-MSR-mClover signal and DAPI counterstain in ESCs without and with doxycycline induction. Scale bars, 5 µm. **c** Time-lapse microscopy experiment in doxycycline-induced TALE-MSR-mClover ESCs. Projections of an ESC nucleus over time reveal the ability of mClover-labelled chromocenters to separate and coalesce (red arrow) and to form protrusions (blue arrow). Scale bars, 5 µm. **d** Quantification of fluorescence recovery after photo-bleaching individual chromocenters. Top panel shows representative images of the FRAP experiment; the dashed squares indicate the photo-bleached region. Lower panel shows the recovery of the mClover signal (not normalised, $\log_2$ median intensity ± SEM) over time at either bleached (left) or unbleached (right) chromocenters. Data represents $n = 63$ photo-bleached chromocenters, collected at least from $N = 3$ independent photo-bleaching experiments and the red dashed line shows the bleaching event. Scale bars, 10 µm. **e** Histogram depicting the percentage ± SEM of mobile and immobile components of chromocenters ($n = 47$ chromocenters), as calculated from the end-value of the recovered fluorescence intensity. See also Supplementary Fig. 1 and Supplementary Movies 1, 2a, 2b, 2c, 2d, 3, 4 and 5.

are consistent with being part of a phase-separated compartment that consists of both dynamic and stable components.

**Satellite transcripts regulate constitutive heterochromatin behaviour in embryonic stem cells.** Chromocenters are distinctively large and less numerous in ESCs[14,23,24], which suggests that an as-yet unknown component acts to promote the fusion of these liquid-like heterochromatin condensates. Because MSR RNA levels are substantially higher in ESCs relative to most other cell types[13,15,19,25,44], and because pull-down experiments indicate that MSR transcripts can interact with HP1α[26,28], we hypothesised that MSR transcripts might contribute to the formation of heterochromatin condensates. To directly test this hypothesis, we applied a commonly used approach for noncoding RNA studies; we depleted both strands of MSR transcripts using sequence-specific locked nucleic acid (LNA) DNA gapmers[31].

LNA DNA gapmers deplete target transcripts by acting at transcriptional[45] and post-transcriptional steps[46]. After transfecting ESCs with LNA DNA gapmers that target the MSR transcripts (MSR gapmers), MSR RNA levels were reduced by ~50%, compared to cells transfected with control gapmers, as shown by RT-qPCR analysis (Fig. 2a). The decreased level of MSR transcripts after LNA DNA gapmer transfection is comparable to that of MSR transcriptional levels in more differentiated cell types (Supplementary Fig. 2a)[15]. The levels of other noncoding transcripts, such as of minor satellites or LINEs, were unchanged after MSR gapmer treatment (Fig. 2a). In addition, we verified that the depletion of MSR RNA in ESCs did not affect the expression of undifferentiated cell markers or promote cell differentiation (Supplementary Fig. 2b, c).

We next used live-cell imaging of TALE-MSR-mClover to measure chromocenter dynamics in ESCs after depleting MSR transcripts. We confirmed that following the depletion of MSR

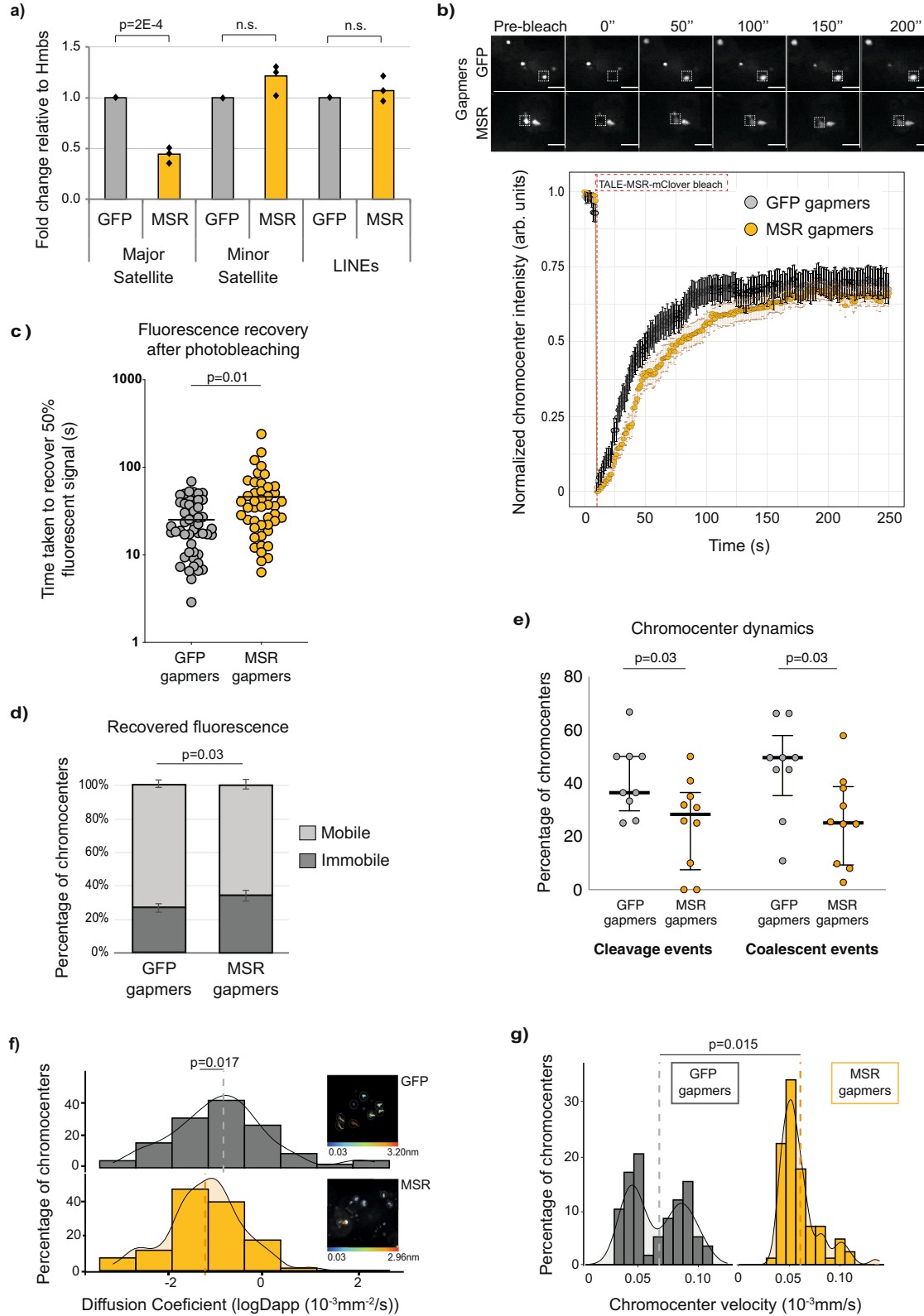

transcripts, TALE-MSR-mClover still localised to chromocenters, as expected (Supplementary Fig. 2d). FRAP experiments revealed that the fluorescence recovery time at photo-bleached chromocenters doubled in MSR RNA-depleted cells compared to controls, indicating that the dynamic movement of TALE-MSR-mClover molecules within chromocenters was reduced (Fig. 2b, c and Supplementary Movie 5). Notably, the fluorescence recovery

immediately after photo-bleaching was substantially slower in the MSR RNA-depleted cells (Fig. 2b), which is consistent with the TALE proteins being within a less dynamic chromatin[14]. In agreement, the proportion of mobile TALE-MSR-mClover slightly decreased in the MSR RNA-depleted cells (Fig. 2d), suggesting that MSR RNA levels affect the proportion of dynamic and stable components within chromocenters. Additionally,

**Fig. 2 Major satellite transcripts regulate heterochromatin behaviour in embryonic stem cells. a** Depleted MSR transcript levels following the transfection of ESCs with LNA-DNA gapmers that target MSR but not control sequences (*GFP*). Transcripts of other repeats were unaffected. Data show the mean values for $N = 3$ independent experiments, compared using an unpaired, two-sided *t*-test. **b** Upper, TALE-MSR-mClover FRAP images in GFP or MSR gapmer-transfected ESCs. Dashed squares indicate photo-bleached areas at 0 s. Lower, Normalised median GFP intensity ± SEM of photo-bleached chromocenters over time after GFP or MSR gapmer transfection. Red line, moment of photo-bleaching. $N = 5$ independent photo-bleaching events ($n = 47$ and $n = 45$ bleached-chromocenters in the GFP and MSR-gapmer conditions, respectively). Scale bars, 10 μm. **c** Increased time to recover 50% of TALE-MSR-mClover signal at bleached chromocenters in ESCs with depleted MSR RNA levels. Each circle represents one photo-bleached chromocenter. $N = 5$ independent experiments ($n = 49$ chromocenters for GFP and $n = 45$ for MSR) compared using a two-sided Mann–Whitney test. **d** Mobile and immobile components of chromocenters (percentage ± SEM). $N = 5$ independent photo-bleaching experiments ($n = 47$ chromocenters for GFP and $n = 45$ for MSR) and the proportions were compared using a two-sided Mann–Whitney test. **e** Percentage of interphasic chromocenters per cell engaging in cleavage or coalescence events after GFP or MSR-gapmer treatment. Dots represent the percentage of chromocenters per cell ($n = 79$ and $n = 183$ chromocenter trajectories analysed after GFP and MSR gapmer treatment, respectively) with a track duration of at least 10 min (images acquired every 30 s for 20 min). Bars show median with interquartile range. Gapmer conditions compared using an unpaired, two-sided *t*-test. **f, g** Fitted apparent anomalous diffusion coefficients (**f**) and velocity (**g**) calculated from mean-squared displacement (MSD) of individual chromocenter trajectories. Interphasic chromocenters from $N = 3$ independent experiments ($>n = 90$ cells analysed) were tracked by live-imaging (acquired every second for 310 frames) after GFP ($n = 61$ chromocenters) and MSR ($n = 85$) gapmer treatment, respectively. Vertical lines show the mean for each gapmer treatment, compared using an unpaired two-sided *t*-test. Inserts (**f**): individual chromocenter trajectories; colour gradients indicate distances moved through the 310 s of acquisition. See also Supplementary Fig. 2 and Supplementary Movie 6.

chromocenters tracked by live-cell imaging over a fixed period of time engaged less frequently in coalescence or cleavage events in ESCs with depleted MSR RNA levels, compared to control cells (~25% of chromocenters coalesce/cleave after MSR-gapmer treatment, compared to ~45% in GFP-gapmer-treated ESCs, Fig. 2e). To further characterise effects in chromocenter mobility, we calculated the average mean-squared-displacement for individual interphasic chromocenter trajectories and extracted the apparent diffusion coefficient (Dapp) and the velocity over a fixed period of time. In agreement with a decreased chromocenter mobility following MSR RNA depletion, control ESCs showed higher anomalous diffusion coefficient (Fig. 2f) and a prevalence of fast moving chromocenters (Fig. 2g), compared to the chromocenters of cells when MSR transcripts were depleted. These data suggest that MSR RNA levels regulate the mobility of chromocenters in ESCs.

To validate and extend our analysis, we further assessed chromocenter dynamics by tagging an intrinsic component of heterochromatin, HP1α. We expressed a fusion protein between HP1α and monomeric Eos3.2[47] in ESCs and confirmed its localisation to heterochromatin (Supplementary Fig. 2e). We performed FRAP experiments to evaluate the effect of MSR transcript depletion on heterochromatin dynamics. Consistent with our TALE experiments, we observed an increase in the time taken to recover 50% of the fluorescent signal following MSR gapmer treatment compared to the control treatment (13 vs. 8 s, respectively; Supplementary Fig. 2f, g and Supplementary Movie 6). Additionally, we detected an increase in the immobile portion of HP1α after MSR gapmer treatment, compared to the control gapmer (26% vs. 15%, respectively, Supplementary Fig. 2h). Taken together, these data lead us to propose that MSR RNA promotes a biophysically dynamic environment within individual chromocenters.

**Satellite RNA promotes phase-separation of HP1α.** We next investigated a potential mechanism by which MSR RNA might contribute to the biophysical properties of heterochromatin. As MSR RNA binds HP1α[26,28], we hypothesised that MSR RNA might affect the ability of HP1α to phase-separate. Indeed, adding in vitro transcribed MSR RNA (272 nucleotides containing a single repeat sequence) to recombinantly purified human HP1α induced droplet formation in an HP1α concentration-dependent manner (Fig. 3a). Furthermore, MSR RNAs containing multiple repeat sequences promoted droplet formation at progressively lower HP1α concentrations, comparable to, and in some cases

better than, enzymatically synthesised polyuridylic acid (polyU) at identical nucleotide concentrations (Fig. 3a and Supplementary Fig. 3a). Within a given RNA size, both strands of RNA showed comparable values of critical concentration (the minimum concentration of HP1α needed to form detectable droplets), indicating that both strands are equally efficient at promoting droplet formation. However, transcripts corresponding to the forward strand consistently yielded larger droplets as compared to the reverse strand (Fig. 3a and Supplementary Fig. 3a). Of note, the HP1α protein in these experiments is not phosphorylated and does not phase separate by itself under the conditions used[7]. These results, therefore, suggest that MSR RNAs are capable of binding to HP1α and promote thematically similar behaviour as has been observed for HP1α with DNA in vitro[7,8]. This model provides a potential molecular basis for understanding why HP1α-associated heterochromatin foci are smaller after MSR RNA levels are depleted (Fig. 3a). And, conversely, it offers an explanation as to why the induction of MSR transcription in *cis* in differentiated cell types can drive the fusion of chromocenters into larger aggregates[15].

We next used structured illumination microscopy (SIM) and confocal imaging to test whether a reduction in MSR RNA levels affects HP1α structure at chromocenters. Upon MSR transcript depletion, SIM experiments showed that HP1α formed a smaller and more compact core at chromocenters, and seemed to have a stronger overlap between HP1α and H3K9me3 signals (Fig. 3b, Supplementary Fig. 3b and Supplementary Movies 7a (control gapmer) and 7b (MSR gapmer)). Analysis of confocal images confirmed there was increased colocalisation of HP1α and H3K9me3 following MSR gapmer treatment (Fig. 3c). Consistent with this finding and with an increased portion of immobile HP1α at chromocenters (Supplementary Fig. 2h), we detected by chromatin immunoprecipitation (ChIP) increased HP1α at the underlying satellite DNA following MSR RNA depletion (Fig. 3d). We next investigated whether the increase of chromatin-associated HP1α at chromocenters affected chromatin compaction by using SiR-DNA fluorescence lifetime methodology that measures chromatin compaction levels in defined genomic compartments[48]. Following MSR RNA depletion, we observed a global increase in chromatin compaction that was more significant at heterochromatin than at euchromatin regions of the nucleus (Fig. 3e and Supplementary Fig. 3c). Together, these results lead us to conclude that high levels of MSR RNA in ESCs promote the fusion and aggregation of chromocenters into large foci, and also prevent premature

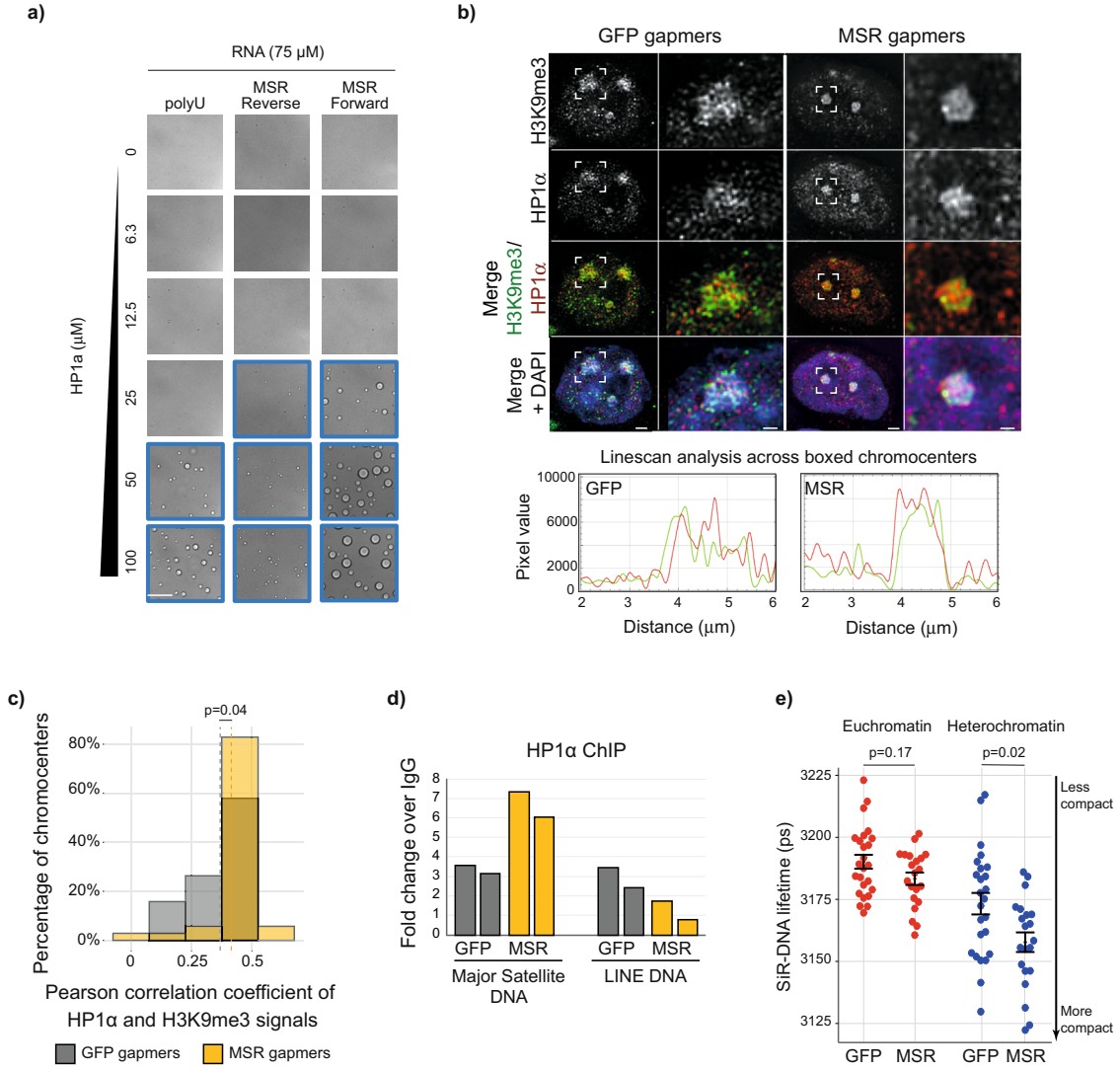

**Fig. 3 Satellite RNA promotes phase-separation of the heterochromatin protein, HP1α. a** In vitro transcribed MSR RNAs, or synthetic polyU RNA, were added to increasing concentrations of recombinant HP1α protein. Images framed in blue show the conditions that allowed for the formation of HP1α droplets. All images show the same magnification; scale bar, 100 μm. **b** Representative images of 3D reconstructions acquired with SIM, showing details of DAPI, H3K9me3 and HP1α organisation at chromocenters in ESCs that were transfected with GFP or MSR gapmers (scale bars, 2 μm). White boxes indicate zoomed-in areas shown to the right (scale bars, 0.5 μm). Linescan analyses of the boxed chromocenters are shown below and reveal the overlap of heterochromatin marks at chromocenters (HP1α, red line; H3K9me3, green line). **c** Histogram showing the Pearson's correlation coefficients of H3K9me3 and HP1α signal intensities on a pixel-by-pixel basis at chromocenters, as measured using confocal analysis of chromocenter mid-sections. Vertical dashed lines show the mean Person correlation coefficient for $n = 1981$ and $n = 5523$ chromocenters quantified in 19 (761 cells) and 34 (1045 cells) images acquired after $N = 2$ independent GFP and MSR gapmer transfection experiments, respectively. Samples were compared with a two-sided Mann–Whitney test. **d** Histograms of ChIP results show the fold-change over IgG of HP1α bound at MSR DNA or LINE DNA upon transfection of ESCs with GFP or MSR gapmers. $N = 2$ biologically independent samples are shown. **e** Plot shows the fluorescence lifetime of the far-red dye SiR-DNA in both euchromatin and heterochromatin regions of nuclei following the transfection of ESCs with GFP or MSR gapmers. Each point represents the mean lifetime intensity per image from three independent experiments, each with >8 images per sample. Shorter fluorescence lifetime means increased chromatin compaction. Samples were compared with a two-sided Mann–Whitney test; error bars in black show mean ± SEM. See also Supplementary Fig. 3 and Supplementary Movies 7a and 7b.

heterochromatinisation and compaction of chromatin within these domains.

**Satellite RNA contributes to heterochromatin organisation in embryonic stem cells.** Having established the effect of MSR RNA in promoting rapid chromatin dynamics within constitutive heterochromatin, we next investigated the role of MSR transcripts in chromocenter organisation. Chromocenters can be detected as DAPI-bright nuclear foci, as shown by the overlap of DAPI and

MSR-TALE-mClover signals (Supplementary Fig. 4a) or with major satellite, but not minor satellite, DNA-FISH probes (Supplementary Fig. 4b). It is possible that DAPI staining could miss subcentromeric satellite foci. We therefore examined this possibility using DNA-FISH with major satellite probes and by visualising MSR-TALE-mClover signals (Supplementary Fig. 4a), but we did not detect any subcentromeric satellite foci (Supplementary Fig. 4b). Analysis of chromocenter properties revealed that MSR RNA depletion triggered the rapid reorganisation of heterochromatin into more numerous, and smaller

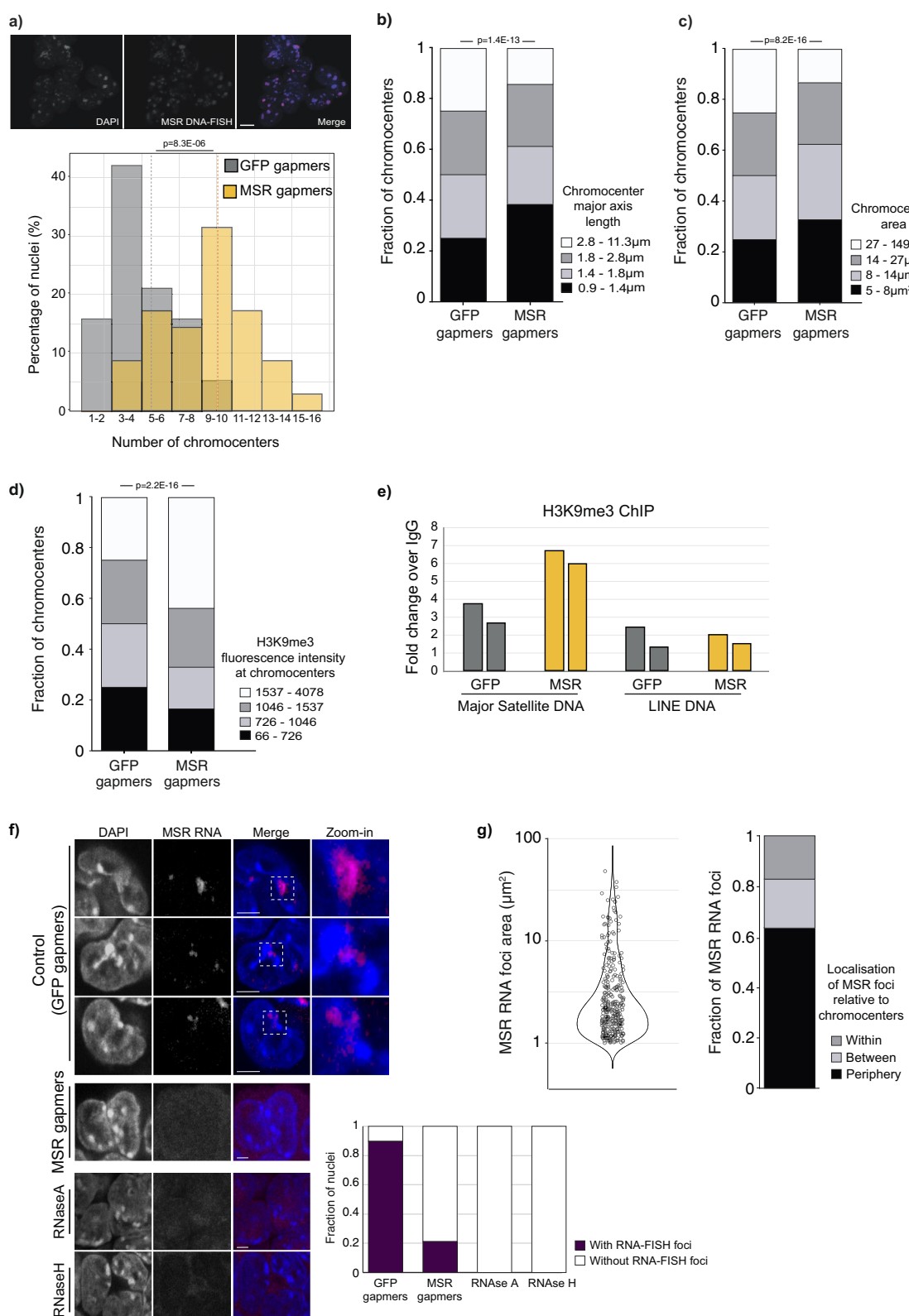

chromocenters (Fig. 4a–c and Supplementary Fig. 4c), an organisational pattern characteristic of differentiated cell types. Differences in chromocenter organisation when comparing between MSR and GFP gapmer-treated cells was also confirmed by line-scan DAPI-analysis (Supplementary Fig. 4d). In addition, immunofluorescence microscopy and ChIP revealed the increased accumulation of H3K9me3 at heterochromatin domains after

MSR RNA depletion (Fig. 4d, e), relative to controls. These results extend support that changes occur to the state of heterochromatin at chromocenters upon MSR RNA depletion.

To further understand the potential role of MSR RNA in chromocenter organisation, we performed RNA-FISH experiments and imaged throughout the nuclei of ESCs. Nearly all (>90%) ESC nuclei contained MSR RNA foci (Fig. 4f). We

**Fig. 4 Degradation of satellite transcripts induces chromatin remodelling in embryonic stem cells. a** Top: ESCs labelled with DAPI (blue) or major satellite DNA-FISH probe (red). Bottom: Percentage of nuclei containing different numbers of chromocenters scored by DAPI in ESCs transfected with GFP ($n = 774$ nuclei) or MSR ($n = 1084$ nuclei) gapmers. $N = 3$ independent experiments, compared using a two-sided Mann–Whitney test. Dashed vertical lines, mean chromocenter number per condition. Scale bar, 10 μm. **b, c** Major axis length (**b**) and area (**c**) of chromocenters. Measurements were made of individual chromocenters at their mid-point in $n = 774$ and $n = 1084$ nuclei, following the transfection of ESCs with GFP or MSR gapmers, respectively. Data were binned into quartiles and the changes in the MSR fractions were compared to the GFP control. $N = 3$ independent experiments, compared using a two-sided Mann–Whitney test. **d** Quantification of H3K9me3 immunofluorescence intensity at chromocenters that were defined as DAPI-large foci, at the optimal focal plane for each chromocenter. A total of $n = 3209$ and $n = 2849$ chromocenters were quantified following $N = 2$ independent GFP and MSR gapmer transfection experiments, respectively. Data were binned into quartiles, and the fractions of H3K9me3 intensities at chromocenter in the MSR condition compared using a two-sided Mann–Whitney test. **e** ChIP-qPCR data show the fold-change over IgG of the H3K9me3 signal at MSR DNA or LINE DNA following GFP or MSR gapmer ESC transfections. $N = 2$ biologically independent samples are shown. **f** RNA-FISH single-section images show the distribution of major satellite RNA in ESCs following treatment with GFP gapmers (control) or MSR gapmers, or upon RNaseA or RNaseH treatment. Scale bars, 5 μm. White boxes indicate zoomed-in areas shown to the right. Chart shows the proportion of nuclei with MSR RNA foci in the different treatment conditions ($n = 127$ nuclei in GFP and $n = 114$ in MSR gapmer samples; and $n = 50$ in each of the RNAse-treated samples, from $N = 2$ independent experiments). **g** Left, MSR RNA foci areas at their mid-point in ESC nuclei. Right, their localisation in relation to chromocenters. MSR RNA foci ($n = 331$) were measured in $n = 127$ nuclei from $N = 2$ biologically independent GFP gapmer-treated experiments. See also Supplementary Fig. 4 and Supplementary Movie 8 .

observed variable patterns of MSR RNA-FISH signals when comparing between individual cells, particularly in foci size and signal distribution in relation to chromocenters (Fig. 4f, g, Supplementary Fig. 4e and Supplementary Movie 8). We observed MSR RNA signal within the interior of chromocenters (~17% of MSR RNA foci), although more commonly, MSR RNA foci seemed to reside at the periphery of chromocenters (64%) and were sometimes located in the space between adjacent chromocenters (19%) (Fig. 4f, g and Supplementary Fig. 4e). We confirmed that the proportion of nuclei with MSR foci was strongly reduced following MSR gapmer transfection, and also when treated with RNAse, demonstrating the specificity of the RNA FISH signal (Fig. 4f). Taken together, the distribution of major satellite transcripts at the periphery of chromocenters, and sometimes within chromocenters, support a potential role for MSR RNA in maintaining the dynamic and distinctive organisation of heterochromatin domains in ESCs.

**Chromocenter architecture protects chromosome stability in embryonic stem cells**. Disrupted heterochromatin maintenance is often associated with the onset of chromosome instability, elevated DNA damage, and with defective mitosis. In somatic cells, this type of heterochromatin perturbation is typically triggered by the weakening of heterochromatin-associated processes[3]. ESCs are unusual, however, in that they can seemingly tolerate a permissive and uncompacted heterochromatin state without adverse consequences[49,50]. We therefore investigated whether a distinct heterochromatin state is not only tolerated in ESCs but perhaps also required to maintain proper chromosome segregation. We observed—shortly after MSR RNA depletion—that a large proportion of metaphase-stage nuclei showed clear hallmarks of chromosome instability (Fig. 5a, b). Specifically, ~20% of nuclei had chromosome fusions (mainly Robertsonian translocations), ~4% contained chromosome breaks, and ~30% showed signs of fragility at pericentromeric regions; defects that were rarely observed in control samples. Furthermore, immunofluorescence microscopy revealed a significant increase in the number of γH2AX foci in chromocenters after gapmer treatment (Fig. 5c), which indicates the presence of elevated damage at satellite DNA following MSR RNA depletion.

We then tested this association using a different system, this time by forcing an increase in chromocenter heterochromatinisation using a doxycycline-inducible TALE-MSR-KRAB protein[51]. The directed recruitment of the KRAB domain to pericentromeric regions increased the levels of heterochromatin markers (HP1α, H3K9me3 and SUV39H1) at major satellite DNA (Fig. 5d) and

lead to a sharp decrease in MSR transcript levels (Fig. 5e). Consistent with our above observations, the activation of TALE-MSR-KRAB in ESCs substantially impaired chromosome segregation, with ~20% of metaphase-stage nuclei containing fused chromosomes, ~10% with chromatid breaks, ~50% with pericentromeric gaps and >50% of analysed mitotic chromosomes showing defects in sister-chromatid cohesion (Fig. 5f, g). In contrast, samples taken from the same ESC line but without doxycycline induction were karyotypically normal, with no signs of genetic instability (Fig. 5g). These results establish that the unique heterochromatin organisation in ESCs plays an important protective role to safeguard chromosomal stability.

## Discussion

Our study demonstrates that noncoding satellite RNA modulates the physical properties and nuclear organisation of heterochromatin. Satellite repeat elements are highly transcribed in ESCs[13,15,19,25], but it remains unknown whether such transcripts are simply a by-product of the permissive chromatin environment that defines pluripotent cells or whether they themselves fulfil a functional role in genome regulation[52]. We now demonstrate that satellite repeat transcripts maintain constitutive heterochromatin in a more dynamic and mobile state that organises into large chromocenters in vivo.

Our findings show that depleting MSR RNA alters heterochromatin dynamics, increases heterochromatinisation and alters chromocenter organisation (Fig. 6a). These results also provide an interpretation for a previous observation that an unknown nuclear RNA component can sustain the higher-order structure of pericentric heterochromatin[53], and presents evidence that the ability to maintain chromocenter stability resides at least in part with the transcript itself. Using RNA-FISH, we found that MSR transcripts in ESCs tend to localise close to the periphery of chromocenters, a pattern corroborated by recent studies[54,55]. As satellite repeats are under the transcriptional control of pluripotency factors in ESCs[15], these findings provide a direct connection between cell state and heterochromatin regulation. Thus, a key developmental switch controlled by the decline in pluripotency factor availability as cells differentiate, triggers the downregulation of MSR transcripts and the reorganisation of chromocenters towards a configuration that is typical of somatic cells. An important implication arising from our work is that MSR RNA levels can be titrated through regulation of their transcription, degradation and clearance. This could enable the fine tuning of heterochromatin properties into different material states perhaps to regulate chromatin accessibility, genome

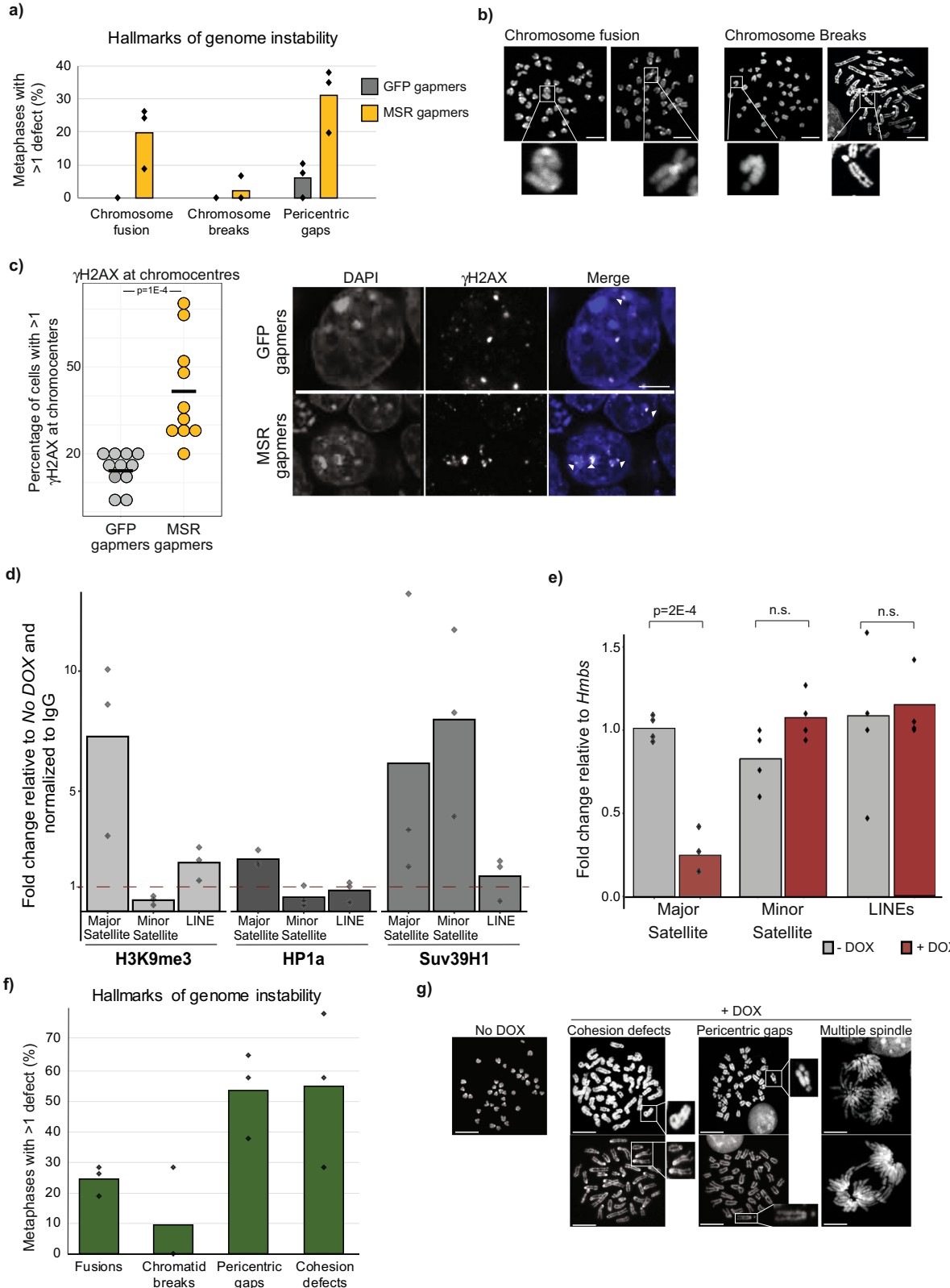

compartmentalisation and/or chromosome structure. An important future direction for research will be to investigate how this pathway integrates with other processes, including protein phosphorylation and alternative ligands, to direct heterochromatin properties and dynamics[56].

Mechanistically, we show that MSR RNA is a ligand that promotes the ability of HP1α to phase separate in vitro. General

modelling of protein-nucleic acid condensation suggests that the addition of nucleic acids reduces the thermodynamic and kinetic barriers of phase separation[57,58], consistent with our observations here. Similarly, RNA molecules can promote liquid–liquid phase-separation by lowering the critical concentration required for proteins to phase separate and/or to modulate the physical properties of phase-separated condensates[37,59]. A limitation of

**Fig. 5 Chromocenter architecture protects chromosome stability in embryonic stem cells. a** Histogram showing that ESCs transfected with LNA-DNA gapmers that target MSR transcripts have a high proportion of metaphases with hallmarks of genetic instability, such as chromosome end-fusions and pericentric gaps. At least 31 metaphases from $N = 3$ independent gapmer transfections. Data points show percentages for each biological replicate. **b** Representative images of DAPI-stained metaphases, exemplifying the cytogenetic defects observed. Scale bars, 10 µm. At least 31 metaphases from $N = 3$ independent gapmer transfections were analysed. **c** Left: histogram showing the percentage of cells with γH2AX foci at chromocenters. Each point represents the percentage of cells with at least one γH2AX foci at the chromocenter, per image analysed. Right, representative images of γH2AX at chromocenters (white arrow). Scale bars, 5 µm. A total of $n = 317$ and $n = 420$ ESCs were scored for GFP and MSR gapmer transfection, respectively, and the two samples were compared with a two-sided Mann–Whitney test. **d** Histogram shows ChIP fold-enrichment over IgG of heterochromatin marks (H3K9me3, HP1α and SUV39H1) bound at major satellite, minor satellite or LINE DNA after TALE-MSR-KRAB doxycycline-induction relative to no doxycycline control. Data was collected from $N = 3$ independent experiments. **e** Histogram showing the expression levels of repetitive elements in ESCs with (+ DOX, red) or without (– DOX, grey) TALE-MSR-KRAB induction by doxycycline. Data was collected from $N = 4$ biologically independent samples and compared with an unpaired two-sided Wilcoxon test (Mann–Whitney test). **f** Histogram showing the genetic defects observed after TALE-MSR-KRAB induction in ESCs. At least 30 metaphases from $N = 3$ biologically independent samples were scored. Data points show percentages for each sample. **g** Representative images of DAPI-stained metaphases with a normal karyotype (– DOX) or cytogenetic defects seen after TALE-MSR-KRAB induction by doxycycline (+ DOX). Scale bars, 5 µm. At least 30 metaphases from $N = 3$ biologically independent samples were scored.

our work is that the concentration of MSR RNA within cells is currently unknown, and obtaining this information would help to better understand and model how the transcripts might contribute to phase separation processes. Future experiments could also investigate the effects of combining DNA/nucleosomes with MSR RNA in changing the ability of HP1α to phase separate. Lastly, it remains important to determine whether a reduction in MSR RNA can trigger specific transitions in the material properties of heterochromatin condensates, with parallels to enhancer condensates[60], and whether the relocalisation of HP1α and other processes is also required.

In addition to modulating heterochromatin state, our work raises the possibility that MSR transcripts could have a second key function: to prevent the untimely and deleterious compaction of chromatin within heterochromatin domains. It remains unexplained how heterochromatin in ESCs is maintained in a more permissive and uncompacted state, despite residing within H3K9me3 and HP1α marked territories[16,17,26,28,61]. We speculate that higher levels of MSR RNA in ESCs might affect the function and dynamics of HP1α at heterochromatin regions. This hypothesis is consistent with prior biochemical experiments indicating that HP1α binds RNA and that the MSR RNA purine-rich forward strand forms different types of secondary structures than the reverse strand[26–28,62]. We found that the depletion of MSR RNA triggered the increased binding of HP1α at satellite DNA and increased chromatin compaction within chromocenters. These data also agree with previous findings that transcriptional downregulation of MSR RNA upon ESC differentiation is closely coupled to progressive compaction of chromatin[15,19,63], whilst activating MSR transcription in fibroblasts results in the decompaction of chromocenters[15,19]. Our in vitro data lead to a model whereby MSR RNA modulates HP1α phase separation, which could trigger changes in HP1α binding dynamics and heterochromatin organisation (Fig. 6b). Indeed, we observed large, gapmer-sensitive MSR RNA foci at the periphery of chromocenters, in between adjacent chromocenters, and sometimes within chromocenters, which is a localisation pattern that is consistent with a role for the transcripts in regulating chromocenter dynamics and organisation in ESCs, in particular fusion/fission events. Alternatively, another plausible model is that altering MSR RNA levels induces local chromatin changes, such as chromatin compaction and heterochromatin-associated histone modification, and these changes affect the ability and dynamics of HP1α to bind to their target sites (Fig. 6b). The models are not exclusive and could potentially work together. Future studies are required to resolve these possibilities.

HP1 molecules can rapidly exchange in and out of chromocenters in fully differentiated mouse cells[63]. Such rapid exchange

is consistent with the low surface tension of phase-separated condensates that arise from electrostatically driven multivalent interactions, such as those between DNA and HP1[56,64]. Excitingly, the results of this study raise the potential for a dose-dependent effect of other charged ligands of HP1, such as MSR transcripts, in regulating heterochromatin by modulating the ability of HP1α to form multivalent interactions. Conversely, the chromatin-chromatin crosslinker ability of HP1α can also contribute to the maintenance of constitutive heterochromatin as a polymer gel[65]. Thus, modulation of HP1α abilities by ligand type and levels could transition chromocenters between different material states, like the liquid-like aggregates described in early embryonic development[8] and in pluripotent stem cells (and as reported in this study), to gel-like or 'ordered collapsed globules' in differentiated cells[63]. Indeed, the differential mobility of distinct fractions of the same molecule (for example, the mobile and immobile fraction of HP1α), and of different types of molecules (DNA, RNA and proteins), contribute to the material state of the whole chromocenter domain. Our work raises the possibility that the availability of MSR transcripts might modulate the material state of chromocenters. The increased immobile fraction detected by both TALE-MSR-mClover and HP1α-Eos may indicate longer binding residence time on chromatin, which would imply an alteration to a more stable material state of chromocenters upon MSR transcript depletion. It remains important to define whether HP1α alone could be the chromatin-binding factor that bridges and collapses chromatin into globules or if other factors, possibly cell-specific, are also involved.

Pericentromeric heterochromatin forms one of the sub-domains of centromeres, and preserving heterochromatin stability, including HP1α function, is required for appropriate chromosome condensation and segregation in most cell types[66–70]. Curiously, ESCs can tolerate mutations that weaken heterochromatin pathways, while the same perturbation in somatic cells causes significant karyotypic defects[49,50]. Unexpectedly, we found in ESCs that MSR transcript depletion and strengthened heterochromatinisation in ESCs led to the rapid appearance of mitotic defects, highlighting the functional differences of heterochromatin regulation that exists between pluripotent and somatic cells. It is currently unclear why ESCs require this unusual form of heterochromatin regulation to maintain chromosome stability. One possibility is that this requirement arose concomitantly with the very short G1 phase of the cell cycle in ESCs[71,72], perhaps to accommodate the rapid reassembly of chromocenters before the start of the next replication round. Changes to the biophysical properties of chromocenters could perturb the correct timing of DNA replication and/or transcription, which could quickly lead to

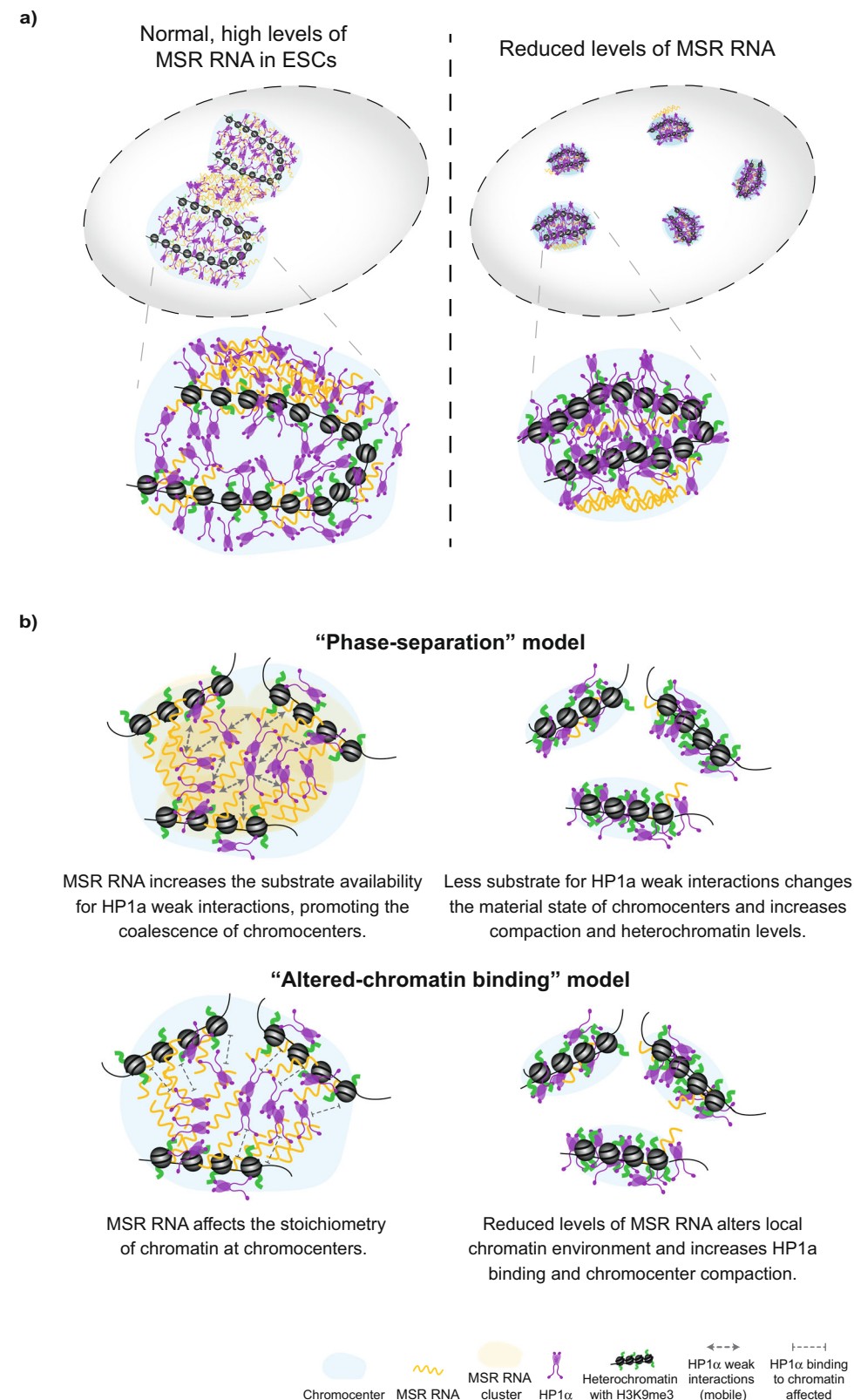

a)

Normal, high levels of MSR RNA in ESCs | Reduced levels of MSR RNA

b)

**"Phase-separation" model**

MSR RNA increases the substrate availability for HP1a weak interactions, promoting the coalescence of chromocenters.

Less substrate for HP1a weak interactions changes the material state of chromocenters and increases compaction and heterochromatin levels.

**"Altered-chromatin binding" model**

MSR RNA affects the stoichiometry of chromatin at chromocenters.

Reduced levels of MSR RNA alters local chromatin environment and increases HP1a binding and chromocenter compaction.

Chromocenter    MSR RNA    MSR RNA cluster    HP1α    Heterochromatin with H3K9me3    HP1α weak interactions (mobile)    HP1α binding to chromatin affected

the observed genetic instability defects. Additionally, increased chromatin compaction at chromocenters in ESCs may impact the accessibility of DNA repair machinery, leading to the accumulation of DNA damage. Highly compact H3K9me3-rich heterochromatin domains accumulate mutations[73] and have slow rates of DNA repair[74,75]. Finally, changes to pericentromeric architecture could impact chromosome segregation by directly interfering with the kinetochore assembly at these regions[67,76].

More generally, our findings imply that an association exists between the biophysical state of heterochromatin and the preservation of chromosome stability. Notably, noncoding transcripts are accumulated in other processes in which the genome is remodelled, such as in the formation of senescence-associated

**Fig. 6 Models of the proposed role for MSR transcripts in chromocenter organisation and dynamics. a** Schematic summarising the contribution of MSR RNA in maintaining a permissive heterochromatin structure in ESCs. The depletion of MSR RNA levels triggers the increased compaction and heterochromatinisation of chromocenters. **b** Two, non-exclusive, models to potentially explain the role of MSR RNA in controlling the dynamics of heterochromatin-associated proteins at chromocenters. In the 'phase separation' model, MSR transcripts catalyse weak interactions between HP1α molecules (as shown in vitro) and this could enable the formation of a 3D hub that phase-separates pericentromeric regions into larger, coalescenced chromocenters. Upon MSR RNA depletion, which normally occurs during ESC differentiation, the substrate for weak interactions is reduced, altering the material state of chromocenters, and leading to the observed changes in heterochromatin compaction and organisation. Alternatively, in the 'altered-chromatin binding' model, reduced MSR RNA levels may change the local chromatin environment (compaction and heterochromatin-associated histone modifications), and these changes affect the dynamics of HP1α binding to their target sites.

heterochromatin foci, cellular stress, embryo development and in human ageing and disease[32,77–81]. Based on the findings reported here, further work is needed in these other contexts to investigate the role of RNA molecules in controlling heterochromatin properties and genome function.

## Methods

**Cell culture**. Male, E14Tg2a mouse ESCs (129P2/OlaHsdl passages 19–28)[82] were cultured in ESC media containing DMEM supplemented with 15% FBS, 1 mM sodium pyruvate, 0.1 mM 2-mercaptoethanol, 0.1 mM non-essential amino acids, 2 mM glutamax and 1000 U/ml LIF. Cells were maintained at 37 °C on mitotically inactivated mouse embryonic fibroblasts and transferred for two passages onto gelatin-coated surfaces prior to collection. ESCs were routinely verified as being mycoplasma-free using a PCR-based assay (Sigma). The E14Tg2a line is not on the list of commonly misidentified cell lines (International Cell Line Authentication Committee).

Stable, doxycycline-inducible ESC lines were made by transfecting E14Tg2a cells using Lipofectamine 2000 (Thermo Fisher) with 1 μg PB-TET-TALE-MSR-mClover-ires-mCherry[15] or with PB-TET-TALE-MSR-KRAB-ires-GFP plasmids, together with 1 μg pCAG-rtTA-Puro and 2 μg pCyL43 piggyBac transposase, followed by selection with 1.2 μg/ml puromycin. ESC lines were flow sorted to purity based on GFP or mClover expression following transient doxycycline induction, and were expanded for up to >10 passages in the absence of doxycycline. Doxycycline was applied at 0.5–1 μg/ml for 24 h to induce transgene expression.

mEos3.2–HaloTag-HP1α was cloned into the mammalian expression vector pEF.myc.ER-E2-Crimson (Addgene #38770). E14 ESCs expressing mouse mEos3.2–HaloTag-HP1α fusion protein was generated as previously described[47] by transfecting appropriate plasmids into E14 cells using Lipofectamine 2000 transfection reagent (ThermoFisher Scientific), followed by selection in 600 μg/ml geneticin (Life Technologies). After 2 weeks of geneticin selection, cells were sorted using a Sony SH800S cell sorter for the expression of mEos3.2.

**Depletion of major satellite repeat transcripts**. ESCs were grown to 70% confluency on gelatin-coated plates and were transfected with LNA gapmer oligos (Exiqon) at a concentration of 100 nM, using Lipofectamine RNAiMaX Reagent (ThermoFisher cat# 13778030), following the manufacturer's instructions. To ensure the robust depletion of target transcripts, the LNA gapmer transfections were repeated at 48 h and 72 h after the first transfection. Cells were collected for further analysis 24 h after the final transfection. The LNA gapmer sequences are: LNA DNA gapmer GFP (gagaAAGTGTGACAagtg), LNA DNA gapmer Major Satellite 1 (acatCCACTTGACGActtg) and LNA DNA gapmer Major Satellite 2 (tattTCACGTCCTAAagtg), where the lowercase letters denote the LNA nucleotides[31]. The LNA DNA gapmer GFP does not recognise the sequence of mClover, which is the GFP variant in the TALE-MSR-mClover construct.

**RT-qPCR**. For most RT-qPCR experiments, RNA was extracted using the RNeasy Mini Kit (Qiagen), reverse transcribed using Superscript II (Life Technologies) and random primers (Promega), and subjected to qPCR analysis, as previously described[83]. To analyse major satellite and other repeat classes, total RNA was extracted using TRIzol (Life Technologies) and treated with two rounds of 1U DNase I (Fermentas) per 1 μg RNA in the presence of RiboLock RNase inhibitor (Fermentas) to remove genomic DNA. RNA (1 μg) was reverse transcribed using Superscript II (Life Technologies) and random primers (Promega), in the presence of RiboLock RNase inhibitor. cDNA was amplified with SYBR Green Jumpstart Taq Ready Mix (Sigma) using primers from[84,85]. Primer sequenced; Major Satellite For, GACGACTTGAAAAATGACGAAATC; Major Satellite Rev, CAT-ATTCCAGGTCCTTCAGTGTGC; Minor Satellite For, TGATATA-CACTGTTCTACAAATCCCGTTTC; Minor Satellite Rev, ATCAATGAGTTACAATGAGAAACATGGAAA; LINE L1 For, CTGGCGAG-GATGTGGAGAA; LINE L1 Rev, CCTGCAATCCCACCAACAAT; Nanog For, ATGCCTGCAGTTTTTCATCC; Nanog Rev, GAGGCAGGTCTTCAGAGGAA; Klf4 For, ACACTTGTGACTATGCAGGCTGTG; Klf4 Rev, TCCCAGTCA-CAGTGGTAAGGTTTC; Hmbs For, CGTGGGAACCAGCTCTCTGA; Hmbs

Rev, GAGGCGGGTGTTGAGGTTTC; T For, TCAGCAAAGTCAAACTCAC-CAACA; T Rev, CCGAGGTTCATACTTATGCAAGGA.

**Chromatin immunoprecipitation**. Crosslinked ChIP experiments were performed as previously described[86]. Cells were fixed with 2 mM DSG (Sigma) for 45 min and then with 1% formaldehyde for 12 min. Sonicated chromatin (250 μg; 200–500 bp fragments) was pre-cleared with blocked beads for 2 h at 4 °C and incubated at 4 °C overnight with either 5 μg HP1α antibody (1/200 dilution, ab77256, Abcam) or 1 μg rabbit IgG (1/200 dilution, Jackson ImmunoResearch). Chromatin-antibody complexes were incubated for 6–8 h at 4 °C with protein A/G magnetic Dynabeads (Life Technologies), washed and the crosslinks were reversed. Native ChIP experiments for H3K9me3 (1/200 dilution, ab8898 Abcam) were performed as previously described[83]. ChIP DNA was analysed by qPCR using primers from[84,85].

**In vitro transcription of major satellite repeats**. MegaScript T3 and T7 kits (ThermoScientific cat# AM1338/1334) were used to transcribe forward and reverse transcripts of differing lengths from major satellite repeat DNA, according to the manufacturer's instructions. To obtain transcripts with one single consensus repeat, oligos containing the forward or the reverse major satellite consensus sequences were synthesised with the T7 (TAATACGACTCACTA-TAGGGACCTGGAATATGGCGAGAAAACT) or the T3 sequences (AAT-TAACCCTCACTAAAGGGTTCAGTGTGCATTTCTCATTTTTC), respectively (GeneScript) and hybridised. Two repeats were transcribed using the pCR4 Maj9–2 template, which was digested with SpeI or NotI for T7 or T3 Megascript kits, respectively[84] (a gift from Thomas Jenuwein). Eight repeats were transcribed from the pySat template[87] (a gift from Niall Dillon, Addgene plasmid # 39238), which was digested with NotI or SalI for T7 or T3 Megascript kits, respectively. Prior to in vitro transcription, linearised plasmids were cleaned with 3 M Sodium acetate and 100% ethanol. Linear templates (1 μg) were in vitro transcribed according to manufacturer's instructions, and RNAs were cleaned by LiCl precipitation.

**HP1α droplet-formation assay**. Polyuridylic acid (polyU, Santa Cruz Bio-technology, cat. no. sc-215733A) was dissolved in MilliQ water, then dialysed extensively against additional MilliQ water in a 3 kDa cutoff dialysis membrane (SpectraPor) to remove co-purified salts, ethanol precipitated and washed prior to final resuspension in MilliQ water. In vitro transcribed MSR RNAs and commercial polyU were quantitated using the total base hydrolysis method of RNA degradation[88] and UV-absorbance quantitation at 260 nm[89] to yield concentration in units of nucleotides. All RNAs were diluted in assay buffer (20 mM HEPES, pH 7.2, 75 mM KCl, 1 mM DTT) prior to use. Recombinant human HP1α protein was purified from *E. coli* as previously described[7]. Prior to use in droplet assays, protein was dialysed in the assay buffer (20 mM HEPES, pH 7.2, 75 mM KCl, 1 mM DTT). Droplet assays were performed using a 384 Greiner Sensoplate (no. 781892) according to the protocol by Keenen and colleagues[90] with a Nikon TiEclipse microscope fitted with a ×20 DIC objective. Briefly, 2× solutions of HP1α (two-fold serial dilutions from 200 μM down to 1.6 μM, and 0 μM) were mixed in equal volume with a 2× solutions of RNA (150 μM nucleotide). The HP1α–RNA mixture was then vigorously pipet-mixed before transferring into the 384-well plate. Mixtures were allowed to settle for an hour prior to droplet imaging.

**Microscopy imaging and analysis**. ESCs were cultured on glass coverslips pre-coated with 0.1% gelatin. For immunofluorescence experiments, cells were pre-permeabilised with 0.1% Triton X-100 in PBS for 2 min, fixed with 3% paraf-ormaldehyde in PBS for 10 min at room temperature (RT), washed three times with PBS for 5 min, permeabilised with 0.1% Triton X-100 in PBS for 10 min, and washed three times with PBS for 5 min. Cells were then incubated with primary antibody (H3K9me3, 1/400 dilution 39161 Active Motif; HP1α, 1/200 dilution ab77256 Abcam; GFP, 1/400 dilution ab290 Abcam; OCT4, 1/200 dilution sc-5279 Santa-Cruz, pH2AX Ser139, 1/200 dilution 05-636-1 Sigma) in blocking buffer (5% milk in PBS) and incubated either for 1 h at RT or overnight at 4 °C. They were then washed three times with PBS for 5 min and incubated with 1/500 dilution of secondary antibodies (anti-rabbit AlexaFluor 488 Abcam ab150077; anti-goat AlexaFluor 488 Abcam, ab150129; anti-goat AlexaFluor 594 Abcam ab150132;

anti-mouse AlexaFluor 594 Abcam ab150108; anti-rabbit AlexaFluor 647 Abcam ab150075) for two hours at RT. Coverslips were mounted onto slides with media containing DAPI (Vectashield H-1200, Vector Laboratories) and were imaged using a Nikon A1-R confocal microscope and a 60X oil objective.

For DNA-FISH experiments, cells were fixed with 3.7% paraformaldehyde for 15 min at room temperature, washed three times with PBS for 5 min and permeabilised for 5 min with 0.5% Triton X-100 in PBS. Cells were washed three times with PBS for 5 min. Probes (500 ng) were denatured for 5 min at 90 °C in 50% formamide, 10% dextran sulfate, 1 mg/ml yeast tRNA in 2×SSC). Cells were denatured in 70% formamide/2×SSC for 5 min at 85°C, washed twice with cold 2×SSC and incubated overnight at 37 °C with the denatured probe. Cells were washed four times with 2×SSC for 20 min, four times with 4×SSC for 10 min, blocked with 3% BSA in 4×SSC for 30 min and washed once with 4×SSC for 10 min. Probes were detected with a streptavidin-TexasRed conjugate (Amersham Biosciences; Buckinghamshire, UK). The sequence of probes used are: for Major Satellite [biotin]CTCGCCATATTTCACGTCCTAAAGTGTGTATTTCTC and for Minor Satellite [biotin]CACTGTAGAACATATTAGATGAGTGAGTTACA CTGA.

For RNA-FISH, cells were incubated for 2 min in ice-cold CSK buffer with vanadyl-ribonucleoside complex (10 mM VRC), then fixed with 3.7% paraformaldehyde, dehydrated through an alcohol series and air-dried. Cells were then incubated with Major Satellite probe (50% formamide, 1 mg tRNA, 1 mg salmon sperm DNA and 10 mM VRC) and detected as above. RNase controls were performed in cells treated with the same protocol, except that VRC was not added to the CSK buffer and cells were incubated with each RNAse for 1 h at 37 °C prior to probe hybridisation. Cells were washed 3 times with 0.1× SSC at 60 °C and once with 2× SSC at room temperature. Slides were dehydrated with ethanol and after air-dried, stained with DAPI before imaging. Imaging was performed with an inverted Nikon TiE microscope with a spinning disk module (Yokogawa CSU-X1), equipped with a ×100 oil immersion objective. Z-stacks were acquired with a frame size of 1000 × 1000, a pixel depth of 12 bits, and a z-distance of 0.3 μm between optical sections.

To account for technical variation between different slides, image acquisition conditions were maintained across biological replicates from the same experiment.

For HP1α-EoS3.2 imaging, a sequential image acquisition was performed, starting with the laser 488. When indicated, a Z-stack of images was collected with 0.2 μm spacing and projected using maximum intensity.

Image files were prepared using Fiji software[91]. Regions of interest were defined using CellProfiler v4.2.0[92] or with Imarisv9.5.1 (BitPlane, Oxford). Briefly, individual nuclei and then chromocenters were segmented using an adjusted threshold based on the background subtraction and to allow for the best separation between signal and background noise, and which was set constant for all images and conditions analysed. The MeasureColocalization module in CellProfiler v4.2.0 was used to quantify the correlation between H3K9me3 and HP1a intensities on a pixel-by-pixel basis, and on mid-sections of chromocenters. The Major Axis Length was calculated as the length of the major axis of the ellipse (chromocenter) and the Area as the number of pixels in the chromocenter region. Linescan analysis was performed as previously described[17], using ImageJ in single optical planes where DAPI foci (chromocenters) were at optimal focal planes and chromocenter width was largest. For RNA-FISH, a gaussian filter was applied and regions of interest defined with the "Surfaces" feature in Imaris v9.5.1. The localisation of RNA FISH foci in relation to chromocenters was defined with the colocalisation feature in Imaris v9.5.1 (% of RNA-FISH colocalised with defined DAPI regions), as follows: 'periphery', when the FISH signal overlaps <50% with one chromocenter; 'between', when the FISH signal overlaps <50% with at least two chromocenters; and 'within' when >50% of the FISH signal overlaps with a chromocenter.

Super-resolution structured illumination microscopy images were acquired with a Nikon dual mode super-resolution microscope with structured illumination (SIM). A series of raw images with 120 nm step intervals were acquired and reconstructed into a high-resolution image shown using the Nikon 3D-SIM reconstruction algorithm that uses minimal parameters and does not require a set threshold to define the limit of the signal.

**Live-cell imaging**. TALE-MSR-mClover cells were grown on glass coverslips coated with 0.1% gelatin. Cells were imaged using an Andor Revolution spinning disk confocal microscope, which was equipped with a thermostatically controlled stage maintained at 37 °C with a ×60 oil immersion objective. A z-stack of images was collected with 0.5 μm spacing and projected using maximum intensity. Three biological replicates per cell line, gapmer treatment or doxycycline treatment, were performed in each experiment.

To analyse the dynamics of chromocenters (cleavage/coalescence events), chromocenters were visualised in image stacks (>40 z-stacks, 0.5 μm each) captured at every 30-s intervals for 20 min. The dynamics of these chromocenters was analysed in Fiji[91]. At each time-point, individual chromocenters were segmented and analysed using the 3D object analyser[93]. Foci centroids were then used to track the dynamics of individual chromocenters using TrackMate v6.0.3[94]. Cleavage was defined where a track branched into two sub-tracks, whilst coalescence events were defined where two independent tracks merged into one. The percentage of tracks engaging in split/coalescence events was scored. Chromocenters tracked for <10 min were excluded from the analysis. For chromocenter displacement, images

were acquired every second for 310 s (310 frames) and individual chromocenters tracked with the TrackMate plugin (v3.8.0) in the Fiji software[94]. Only chromocenters tracked for all series (310 frames) were analysed. Apparent anomalous diffusion coefficients were extracted from the mean square displacement (MSD) of chromocenter trajectories, as described[95].

**Fluorescence recovery after photo-bleaching**. FRAP experiments were performed on an Andor Revolution spinning disk confocal microscope. For TALE-MSR-mClover ESCs, images were acquired every second for 310 s (310 frames). The first ten frames were collected before the bleach pulse for baseline fluorescence. A 10 × 10 region surrounding a single chromocenter per cell was selected for bleaching puncta using 2% laser power (488 nm). Fluorescence intensity image analysis was performed using the TrackMate plugin (v3.8.0) in the Fiji software[94]. After normalisation of the intensity signal, FRAP curves were generated from the fluorescence intensity of each chromocenter, at each timepoint (every second).

For mEos3.2–HaloTag-HP1α ESCs, two images were acquired every second for baseline fluorescence and, after a bleach pulse, images were acquired every 5 s for 61 sec. A 10 × 10 region surrounding one chromocenter (B) and a control nuclear region (background) was selected for bleaching puncta using 5% laser power (488 nm). Fluorescence intensity image analysis was performed using the TrackMate plugin (v3.8.0) in the Fiji software[94]. For each chromocenter, the fluorescent intensity signal was corrected by subtracting the fluorescence intensity of a 10 × 10 background foci from the fluorescence intensity of a 10 × 10 region surrounding one bleached (B CHR) and also from one not bleached chromocenter (Ctrl CHR). The fluorescent intensity of the bleached chromocenters was then corrected with the intensity signal of non-bleached chromocenters, to account for the bleaching effect of long-time acquisition, and finally normalised to the pre-bleached intensity signal. FRAP curves were generated from the fluorescence intensity of each chromocenter, at each timepoint. To calculate the half-time [ln(2)/K] of fluorescence recovery after photo-bleaching, data was fitted to a one-phase association equation $[Y = Y0 + (Plateau-Y0) * (1-\exp(-K*x))]$, where $Y0$ is the fluorescence after photo-bleaching (GraphPad software). To correct for loss of fluorescence due to the bleaching pulse the mobile fraction was calculated as $[F_m = (Plateau-First\ post-bleach)/(Pre-bleach-First\ post-bleach)]$ and the immobile fraction was measured as remaining fluorescence intensity unrecovered at plateau phase. FRAP experiments were repeated five times and data were acquired from one bleached and from at least one unbleached chromocenter per cell.

**Chromocenter compaction analysis**. For SiR-DNA-based analysis of chromatin compaction by FLIM[48], cells were plated in 35 mm glass bottom dishes (P35-1.5-14-C, Mattek Corporation, MA, USA) the day before imaging. Cells were stained with 1 μM SiR-DNA (Spirochrome Ltd., Stein am Rhein, Switzerland) and with 10 μM verapamil (Spirochrome Ltd.) in cell culture medium for 1 h, before changing to cell culture medium that contained 10 μM verapamil and 20 mM HEPES pH 7.4 for imaging. Fluorescence lifetime imaging (FLIM) was performed on a home-built confocal platform (Olympus Fluoview FV300), which was integrated with time-correlated single photon counting (TCSPC) to measure fluorescence lifetime in every image pixel. Output from a pulsed supercontinuum source (WL-SC-400-15, Fianium Ltd., UK, repetition rate 40 MHz) was filtered using a bandpass filter FF01-635/18 to excite SiR-DNA. Fluorescence emission from the sample was filtered using 700/70 nm (Comar Optics, UK) before passing onto a photomultiplier tube (PMC-100, Becker & Hickl GmbH, Berlin, Germany). Photons were recorded in time-tagged, time-resolved mode that permits the sorting of photons from each pixel into a histogram according to their arrival times. The data was recorded by a TCSPC module (SPC-830, Becker and Hickl GmBH). Photons were acquired for 2 min to make a single 256 × 256 FLIM image. Photon count rates were always kept below 1% of the laser repetition rate to prevent pulse pile-up. Photobleaching was verified to be negligible during acquisition. Approximately 10 representative images with several nuclei per field of view were acquired for each condition. FLIM images were analysed using FLIMfit v4.12.1 and fitted with a monoexponential decay function with no scattered light parameter (using identical parameters for all images). The instrument response function for each data set was collected at the same time, and based on reflected light (with filters removed). Whole nuclei were segmented based on intensity (debris and mitotic nuclei were manually removed). A two-level mask that separated heterochromatin spots and euchromatin was created with Icy software spot detection tool[96,97]. Pixels in these two chromatin regions were binned and fitted separately to obtain two lifetime values for each image (one for euchromatin, one for heterochromatin). Statistical analysis was carried out using an unpaired two-samples Mann–Whitney test.

**Quantification and statistical analysis**. Statistical parameters including the exact value of $n$, SD, SEM and statistical significance are reported in the figures and the figure legends. For the majority of experiments, statistical significance is determined by the value of $p < 0.05$ by unpaired two-samples Wilcoxon test (Mann–Whitney test).

**Reporting summary**. Further information on research design is available in the Nature Research Reporting Summary linked to this article.

## Data availability

The data that support this study are available from the corresponding authors upon reasonable request. Source data are provided Supplementary information Source data are provided with this paper.

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

## Acknowledgements

We are grateful to Niall Dillon, Maria Elena Torres-Padilla, Thomas Jenuwein and the Wellcome Sanger Institute for kindly providing plasmids. We thank several Babraham Institute Facilities for their assistance, including the Flow Core, the Bioinformatics Group, and particularly Simon Andrews and Anne Segonds-Pichon for data analysis and statistical support. We thank members of P.J.R.-G.'s group, Rosana Collepardo and Jerelle Joseph for helpful discussions, Aurora Idilli for help with RNA-FISH imaging, Gavin Kelsey and Wolf Reik for comments on the manuscript, and the Babraham Institute Science Policy and Oversight Committee for their support. C.N., H.O., S.W. and P.J.R.-G. were funded by grants from the Biotechnology and Biological Sciences Research Council (BBS/E/B/000C0421, BBS/E/B/000C0422 and BB/M022285/1 to P.J.R.-G. and by the Core Capability Grant to The Babraham Institute), and European Commission Network of Excellence EpiGeneSys (HEALTH-F4-2010-257082 to P.J.R.-G.). S.J.B. and C.N. were supported by the Francis Crick Institute, which receives core funding from Cancer Research UK (FC0010048), the UK Medical Research Council (FC0010048) and the Wellcome Trust (FC0010048). S.J.B. is also funded by a European Research Council (ERC) Advanced Investigator Grant (TelMetab to S.J.B.); and Wellcome Trust Senior Investigator and Collaborative Grants. C.N. is grateful to the Tommy's National Miscarriage Centre for funding support. E.V.W. was supported by an F32 fellowship from the NIH. C.H., C.P. and G.S.K.S. were supported by the Wellcome Trust, Alzheimer's Research UK, the Michael J Fox Foundation, Infinitus China Ltd, and by the European Union's Horizon 2020 Research and Innovation Programme under grant agreement number 722380. S.B. is thankful to Trinity College, University of Cambridge and the Wellcome Trust-MRC Cambridge Stem Cell Institute for funding (2023151/Z/16/Z). E.S. was supported by a British Society for Developmental Biology Gurdon/The Company of Biologists Summer Studentship. G.J.N. was supported by an MIRA grant from the NIH (R35 GM127020 to G.J.N.).

## Author contributions

Conceptualisation, C.L.N. and P.J.R.-G.; Methodology, C.L.N., C.H., E.W., C.P., H.O., M.W., S.W.; Investigation, C.L.N., E.W., C.H., E.S.; Visualisation, C.L.N.; Writing—Original Draft, C.L.N. and P.J.R.-G.; Writing—Review & Editing, C.L.N., E.W., G.J.N. and P.J.R.-G.; Funding acquisition, C.L.N., S.J.B., G.S.K.S., G.J.N., P.J.R.-G.; Supervision, C.L.N., S.B., G.S.K.S., G.J.N. and P.J.R.-G.

## Competing interests

The authors declare no competing interests.
