## [Peer Review File · Nature Communications]

REVIEWER COMMENTS

Reviewer #1 (Remarks to the Author):

In this manuscript, Novo et al investigated the roles of major satellite repeat (MSR) RNAs in the regulation of constitutive heterochromatin domains (chromocenters) in mouse ES cells (mESCs). The authors applied transcription activator-like effector (TALE) system to trace MSR sequences, thereby analyzed biophysical properties of heterochromatin condensates. Via in vitro droplet formation assay, they showed that forward MSR RNA promoted HP1a phase-separation. Furthermore, the authors indicated that depletion of MSR transcripts in mESCs resulted in the more compacted heterochromatin, and the increase of hallmarks of chromosomal instability.

The roles of MSR transcripts in heterochromatin organization had been recently studied by other groups (eg. PMIDs 28760199, 31677973, 33296675). Although this study was well-conducted and results are interesting, some conclusions remain premature, and some data are inconsistent.

Major points:

1. The authors concluded that "depleting MSR transcripts causes heterochromatin to transition into a more compact and static state", mainly based on imaging and FRAP data. They showed that HP1a and H3K9me3 foci became smaller and tighter in MSR reduced cells (Fig 3B), but condensates indicated by TALE-MSR-GFP were larger (Fig 2B); this inconsistency should be explained. It is important to compare these markers in the same cells to obtain more convincing data. Moreover, as shown in the lower panel of Fig. 3B, the colocalization between H3K9me3 and HP1a was somewhat decreased in MSR RNA reduced cells, suggesting these smaller foci could be less chromatin-associated. A recent study showed that the condensate properties detected by DNA are different from those detected by chromatin-associated proteins including HP1a (PMID: 33326747). The authors also examined HP1a and H3K9me3 binding at MAR using ChIP-qPCR (Figs. 3E and 4E). However, they normalized ChIP data to IgG, which is heavily variable among antibodies; the input (total DNA) should be better for normalization. To evaluate the degree of chromatin compaction at MSR, more heterochromatin makers (e.g. H4K20me3, SUV39 etc.) should be examined and complementary assays, such as DNase & MNase digestion would be needed.
2. The authors concluded that "satellite RNA scaffolds heterochromatin organization" in mESCs, but there was no sufficient data supporting this strong claim. Although Fig 3A demonstrated nicely that forward MSR RNA promotes phase-separation of HP1a, the molecular mechanism was not examined. For example, why did only forward RNA promote phase-separation? What is the sequence difference between forward and reverse? What are the DNA motifs recognized by HP1a? Transcripts from which direction are more abundant in chromocenters in mESCs? Do they colocalized with HP1a in the cells? TALE-MSR-GFP can be applied to image DNA but not RNA; RNA-FISH should be applied to detect MSR RNAs in the cells.
The results of Fig. S3A are interesting. Could the authors explain why did 2 repeats of MSR promote the formation of larger HP1a droplets than 1 repeat and 8 repeats did?
3. It is interesting that MSR RNA reduction cause the increase of rH2AX staining at chromocenters (Fig. 5C). If chromocenter became more compacted as the authors suggested, how can DNA be damaged? This is related to point 1.
4. Fig. 5D: The ChIP qPCR data are highly variable among replicates. As I mentioned above, the ChIP data should be normalized to input not IgG. Further, it is better to present control and Dox+ data separately, so that one can evaluate the enrichment among repeat elements.
5. Seeing that the authors applied TALE-MSR-KRAB system to silence major satellites. The authors might want to examine the changes of chromocenter and compared with the results

obtained by gapmers.

Minor questions:

1. Figure 1D. Why the data during recovery stage is discontinued?
2. Figure S2A. To quantify transcript levels, at least three independent experiments are needed.
3. Figure S3B. This figure looks strange. HP1 α is the reader of H3K9me3, but why they are not co-localized? This is inconsistent with that of Fig 3B.

Reviewer #2 (Remarks to the Author):

The work of Novo et al reports on role of the major satellite repeat transcripts (MSRtr) in maintenance of semi-open chromatin state of chromocenters in mouse ESCs and chromosome stability. Using TALE-GFP targeted to MSR, they show dynamics of the chromatin within chromocenters, as well as dynamics of chromocenters themselves, which resembles liquid-like membraneless condensates behavior. Depletion of MSRtr leads to decrease in dynamics of chromocenter chromatin, concomitant with increase of HP1 α , H3K9me3 and chromatin compaction, as well as to increase of chromocenter number. Based on the ability of MSRtr to bind HP1 α , which was demonstrated by droplet assay in vitro, the authors speculate that satellite transcripts scaffold HP1 α in chromocenters and prevent its binding to satellite DNA impeding chromocenter condensation. The authors noticed that MSRtr depletion results not only in chromocenter structure changes but also influences chromosome integrity, in particular, leading to Robertsonian translocations and pericentromeric undercondensation. The paper is well written, sufficiently illustrated and data are comprehensively discussed. The messages of the paper are interesting to the nuclear biology field and the paper is potentially a good fit to Nature Communication journal, if some technical issues listed below are attended.

I have two major concerns for the experimental part.

(1) Firstly, when the authors describe chromocenter structure or their number per nucleus, their assessment is based on DAPI staining and thus can miss small chromocenters (e.g., Fig.4) or immunostaining that stains not only chromocenters (e.g., Fig.3). The methods of choice in such assays are those which specifically detect subcentromeric satellite – either DNA-FISH with MSR probe or targeting MSR with TALE-GFP (see also minor comment 9)

(2) Secondly, if the author believe in the role of MSRtr in organization of chromocenters, they need to demonstrate presence and distribution of the RNA in chromocenters by RNA-FISH. This is especially important in the view of the authors' suggestion that MSRtr compete with MSR for HP1 α binding. If their suggestion is correct, then the MSRtr signals must be really abundant and distributed through the entire chromocenter volume in untreated ESC, while this signal will be significantly reduced after MSRtr depletion. In addition, it is important to carry control RNA hybridizations after RNasing.

I also would like to mention that the supplementary videos are uninformative or wrongly interpreted (see below).

Minor comments:

(1) Supplementary Figure 1C:

The histogram shows percentage of nuclei with 1, 2 and 3 chromocenters. As one can see from other figures in the paper, as a rule, ESCs, have more than 3 chromocenters.

(2) Supplementary Figure 1D:

How the authors define a diameter of a chromocenter? Even on projections, chromocenters have very irregular shape. Besides, to define a border of a chromocenter, one needs to threshold an image. How thresholds were set up for different slides with different acquisition conditions? It is not clear from the legend, e.g. how many nuclei were assessed in the three independent experiments – 3 x 16 or 16 in general? In any case, this statistic is pretty poor.

(3) Figure 1C:

Are those projections or single optical section? In the latter case, the changes in shape of chromocenters can be explained by a chromocenter (nucleus) movement with parts of chromocenter coming out and in focus.

(4) Video 2a and 2b suppose to show “rapid movement of chromocenters” however both of them show merely cell movements, whereas relative positioning of chromocenters does not change.

(5) Video 2 and 3 suppose to show separation and coalescence of chromocenters, however, the videos do not show these phenomena

(6) Page 8:

“...a molecular basis for understanding why HP1 α -associated heterochromatin disperses into smaller condensates after MSR RNA levels are depleted (Figure 3B)”. The Figure 3B does not show this phenomenon but rather illustrates changes in chromocenter structure.

(7) Page 9 (Figure 3B):

To compare overlapping signals of H3K9me3 chromatin and chromatin binding HP1a, the authors have chosen to compare volumes of the signal. This is not an appropriate method, since volume rendering strongly depends on a threshold, which in this case has to be set separately for two different channels, individual cells and different acquisitions. The colocalization has to be shown and estimated on single optical sections through mid of chromocenters.

(8) Figure 3D:

This figure does not convince that brightness of the fluorescent signal is different in two conditions. Authors have to explain (a) how the HP1a fluorescence was measured – on projections or single sections, for individual cells or group of cells – and (b) how can they be sure about equal conditions of staining and image acquisition from two different slides, which is essential for such comparison.

(9) Figure 4A:

Firstly, how many nuclei were analyzed in this experiment? Secondly, the histogram shows number of chromocenters from 1 to 6. However, the very image on the left shows 8 chromocenters. I am working with various mouse cells types half of my scientific career but I never saw mouse somatic cells with just one chromocenter. I think the authors are mistaking because they rely only on DAPI staining that might be misleading, especially in case of ESCs. To make a correct assessment of chromocenter number, a DNA-FISH with a probe for MSR, specifically detecting subcentromeric heterochromatin, has to be employed.

(10) Figure 4B,D and C:

How DAPI and H3K9me3 intensities were measured? – see also (8). I wonder how the authors measured chromocenter diameters? – see also comment (2)

(11) Figure 5B:

The authors cannot claim that the very left panel shows a fusion of arms of two different chromosomes – the chromosomes simply got close to each other during spreading; fused chromosomes have different configurations on spreads. Fusion of chromosomes by their centromere regions is called Robertsonian translocation and should be named like this in the MS.

(12) Figure 5F: “cohesin defects” has to be changed to “cohesion defects”

Reviewer #3 (Remarks to the Author):

In this work, Lopes Novo et al investigate the role of major satellite repeat transcripts in organization and biophysics of the constitutive heterochromatin domain in mouse embryonic stem cells. They demonstrate that the presence of these repetitive transcripts affects compaction and material state of heterochromatin. Most interestingly, they connect these changes in biophysical state to functional outcomes of heterochromatin disruption, including genome instability and DNA damage accumulation. This work is novel in that it is attempting to connect molecular mechanism to biophysical state to functional outcome of an important chromatin compartment that is of broad interest. If the authors address the points raised below, I recommend it for publication in Nature Communications.

Overall, the use of novel tools like TALE-MSR-GFP allows for unique investigations of heterochromatin structure and function in live cells. However, I urge caution to the authors with regard to interpretation of data taken on this synthetic protein in terms of what it means for the endogenous heterochromatin domain. For example, changes in FRAP recovery of the TALE-MSR-GFP upon gapmer knockdown of MSR RNA does indicate that presence of the RNA affects mobility of the TALE-MSR-GFP protein, but is not necessarily generalizable to other proteins. This criticism can be remedied by placing more emphasis on changes measured in proteins that are endogenously present in the domain (i.e. HP1 α), or by taking FRAP measurements of other proteins present in heterochromatin (i.e. SUV39H1). Additionally, I encourage the authors to further elaborate on the discussion of multiple populations of the same molecule with different mobility (e.g. mobile and immobile fraction of HP1 α) and differential mobility of types of molecules in the domain (DNA vs. RNA vs. protein) that lead to the emergent material state of the whole domain.

Specific comments on figures below:

Figure 1: The chromocenters marked by TALE-MSR-GFP in these ESCs are highly dynamic (1C). In addition to FRAP, which gives measurement of mobility of individual components, the timing of fusion and rounding up of whole domains can give measurements of the viscosity of the domain and surrounding nucleoplasm, which will further support the ‘gel-like’ or ‘liquid-like’ picture presented in Figure 6. From these data or similar movies of fission / fusion events, perhaps a viscosity can be measured following methods similar to those used for nucleoli in Caragine, Haley & Zidovska, eLife 2019 (<https://elifesciences.org/articles/47533>). This method requires high resolution images for accurate measurement of the bud neck, and I am not sure what types of microscopy are available to the authors, so I consider this an informative but not essential experiment for revision.

1D: Is there a normalization happening for this FRAP trace? In Fig 2 the FRAP data seems normalized but not here in 1D.

1E: Please indicate the time post-bleach at which the immobile fraction was calculated in the figure legend.

Figure 2:

2B: Please indicate both on the figure and in the legend what protein is being bleached (TALE-MSR-GFP). Also, in the legend make sure to report the total number of bleach areas used to calculate the FRAP curve, similar to how it was reported for 1E. I see only number of trials reported.

Major question: Does the GFP gapmer used here as a control knock down levels of TALE-MSR-GFP, and could this affect the protein's mobility? The methods section suggests that the GFP gapmer does not match the mClover sequence used in the TALE-MSR-mClover. Was any control experiment done (i.e. western blot of TALE-MSR-mClover with and without gapmers)?

Please also write in the text that the actual fluorophore is mClover and the gapmer should not bind.

Additionally, is the TALE-MSR-GFP protein capable of binding dsRNA? Later the authors suggest a molecular mechanism for MSR RNA abundance modulating HP1's mobility by competing for binding with DNA. Does that model also apply to the synthetic protein?

Figure 3:

Major point: for these cell-specific assays like SIM, is there a marker of which individual cells received the gapmer? Or a control of what percent of cells receive the transfection?

3A: Because forward and reverse transcripts of MSR have different effects on HP1 α 's phase behavior in vitro, we might expect them to differentially influence its behavior in vivo. Have the authors treated cells with a single gapmer that would knock down only forward or only reverse transcripts? Also, have the authors tried mixing both strands with HP1 α in vitro? Are MSR transcripts thought to create dsRNA in vivo?

3B, bottom: If you make the scale of the x axis (Distance) the same for left and right, it will highlight the size/compaction difference between the two conditions more.

3C: There are no statistics on the volume colocalization method used here and the variance is high, are these changes statistically significant? Could you use another method like radially averaged autocorrelation to confirm?

3F: Please include in the figure legend that shorter fluorescence lifetime means increased chromatin compaction.

Figure 4:

4A: Please indicate number of nuclei per trial in figure legend for 4A. I assume the same raw data was used to quantify different metrics in 4A, B and C?

4B, C, D: Within the figure, chromocenter is spelled 'chromocentre', please use one consistent spelling.

4E: The authors indicate that the pattern of more, small chromocenters instead of few larger ones could be indicative of differentiation, but also show earlier in S2B and S2C that pluripotency markers are unaffected. So are these uncoupled phenotypes?

Figure 5:

5B: In the right-most image, the width / length of the mitotic chromosomes is significantly different from the other three examples here; what is going on?

5C: Why are the quantifications grouped by image instead of displaying a point for each nucleus, or a violin plot representing cells equally? Without seeing the images or a summary of number of cells per frame, the reader doesn't know whether frames with a few cells are getting disproportionate representation.

5G: again the chromosomes look wildly different in terms of compaction, length/width. Is this a phenotype? It is not mentioned in the text.

Figure 6:

The model presented in figure 6 suggests that increasing the amount of MSR RNA fluidizes the domain but is difficult to interpret from the image alone—consider highlighting purposefully drawn changes with labels like 'increased compaction'. Also, the color and placement of "Liquid-like" and "gel-like" chromocenter colors is confusing, it seems to insinuate formation of a liquid-like compartment surrounding a gel-like core, but I think it's really just representing mobile and immobile fractions of FRAP populations.

Discussion:

What do the authors interpret for both types of changes in FRAP data, T1/2 and mobile/immobile fraction? Increase in immobile fraction can indicate longer binding residence time on chromatin, while changes in T1/2 can more closely represent diffusion characteristics. Do the authors interpret these as both contributing to the material state of the heterochromatin domain? The discussion and interpretation of material state and its contribution to function are a bit under-developed. Again I encourage additional measurements of material state, for example viscosity, through the method mentioned above. And how would disruption of material state lead to these measured differences in chromatin organization and DNA damage accumulation? The authors elaborate beautifully on molecular mechanism of dynamics but the connection from molecular mechanism to chromosomal stability and function is slightly disappointing.

Reviewer #4 (Remarks to the Author):

In this paper by Novo et al, the authors study the properties of chromocenters in mouse embryonic stem cells (ESCs) and investigate their response upon transfection of major satellite repeat (MSR) gapmers, which target and deplete MSR transcripts. The authors show that chromocenters in ESCs can undergo fusion/fission, and that MSR gapmer transfection leads to increased turnover of an ectopically expressed TALE that binds chromocenters, increased colocalization of H3K9me3 and HP1a at chromocenters, an increased number of chromocenters in the cell, and the appearance of hallmarks of genome instability. They combine these results with in vitro work showing that MSR transcripts promote phase separation of HP1. Based on their results, the authors suggest that MSR transcripts promote a liquid-like state at chromocenters in ESCs, and that this liquid-like state is lost upon MSR gapmer-induced depletion of MSR transcripts. Furthermore, they suggest that loss of the liquid-like state is responsible for the effects seen in response to MSR gapmers listed above.

While I find that this is a potentially interesting paper, I am not convinced that the effects observed upon MSR gapmer transfection are necessarily linked to the capability of MSR transcripts to promote a liquid-like chromocenter state in the cell. I find that the authors need more data if they want to convincingly make this point, and it would also be important to compare their data to other models besides a "liquid condensate" model to test in a more unbiased way which model explains the data best. Specific suggestions are listed below.

Major points

- The fusion/fission events in Fig. 1C are maybe the most convincing piece of data showing that chromocenters in ESCs are liquid-like. It would be important to quantify this (how many fusions/fissions per chromocenter per time?) and to show how this number changes if MSR gapmers are transfected.

- I do not follow the conclusion that the increased turnover of TALEs upon transfection of MSR gapmers means that MSR transcripts promote a "dynamic, physical environment within chromocenters". An alternative (and potentially simpler) explanation would be that TALEs bind stronger to their target (MSR) DNA when MSR gapmers have been transfected, for example, because less MSR transcripts are produced and therefore less RNA Polymerase molecules run along the MSR DNA to counteract TALE binding. This should be considered/tested as an alternative explanation, for example by using a protein that binds other sequences in the chromocenter (= in the same dynamic, physical environment), like minor satellite repeats that do not change their transcriptional activity with the gapmers used here. Furthermore, if MSR transcripts create a liquid condensate, as implied by the authors, there would actually be a good chance that the viscosity of this condensate is higher than the viscosity of the liquid that fills the cell, so that the mobility of molecules within that condensate would decrease and not increase (the liquid that fills the cell is not much more viscous than water, see for example this review by Luby-Phelps that discusses viscosity measurements made with different techniques: PMID 10553280).

- To better understand the effects seen upon MSR gapmer transfection, it would be important to make sure that these effects come indeed from the depletion of MSR transcripts in the cell, and not from some other effects that MSR gapmers might induce. For example, MSR gapmers could also alter the chromatin environment at chromocenters by binding to MSR DNA (potentially in transcription bubbles) or to nascent RNA transcripts, thus regulating Pol II occupancy, or by titrating away proteins etc. This could also account for some of the effects seen upon MSR gapmer transfection, especially as the reduction of MSR transcript levels seems moderate according to Fig. 2A (about 2-fold). Can the gapmer-induced effects be rescued by providing additional MSR transcripts? Do chromocenters become more liquid-like if MSR transcripts are added to differentiated cells? Can MSR transcript levels be modulated without transfecting MSR gapmers, for example, by expressing a TALE that is fused to a transcriptional repressor like KRAB, and are similar effects observed?

- How do the concentrations and the stoichiometry in the in vitro assays shown in Fig. 3A relate to those in the cell? This is a critical piece of information in order to understand if MSR transcripts are likely to create a liquid-like condensate at chromocenters, similar to what is obtained in the test tube. If the absolute number/concentration of MSR transcripts in ESC chromocenters is not known, I think it should be measured. Otherwise, it is hard to decide if it is likely that MSR transcripts act "directly" as a component of a condensate, or rather indirectly in some other way (for example, by changing the chromatin environment/histone marks/proteome at chromocenters).

- For the final model in Fig. 6, I am wondering how pronounced the differences between the two states actually are. What is the functional consequence of chromocenters showing fusion/fission in the left case (termed liquid-like, ESCs) but not in the right case (termed gel-like, differentiated cells), and what is the functional consequence of proteins binding a few seconds longer in one of the cases? Taking one step back, heterochromatin appears quite dynamic in ESCs and also in differentiated cells when it comes to binding of HP1 and other heterochromatin proteins that exchange within seconds as has been shown in the past and as is shown here, so the functional relevance of the distinction between two pretty similar (= two pretty dynamic) chromocenter states is not clear to me. In particular, it is

not clear if these states are really linked to the hallmarks of genomic instability, or if this is just a correlation, and it would be important to strengthen this aspect.

Minor points

- Would it be possible to determine the interfacial tension from the morphology of the fusion/fission intermediates shown in Fig. 1C? This could tell us something about the "condensate properties" of chromocenters.

- It is not uncommon that DNA-binding proteins with low nM Kd values show dynamic FRAP recoveries in cells, see for example mCherry-LacI in Fig. 1E in Chong et al, PMID 29930090 (LacI binds lacO with Kd < 1 nM, but it recovers when bleached in the indicated paper). This might have many reasons, like competition with cellular proteins that bind to the same DNA sequence etc, and I therefore do not see how this is a readout for the "physical properties of chromocenters", whatever the authors mean with this term. The turnover of a chromatin-associated protein at its target site is certainly not a readout of the local viscosity, as it will depend on many other factors (accessibility of the sequence, active processes counteracting binding, etc).

- Why is the FRAP recovery of the TALE construct in J1 ESCs (ref 40) much slower than that measured here?

- The FRAP curves in Fig. 2B look pretty similar to each other. Although the difference might be significant, it would be interesting to know what functional relevance the authors assign to this small difference.

- Figure S3 shows that droplets become smaller with RNAs that have more repeats. What does this mean, and what is the size of these RNAs in the cell? The size seems to be regulated (e.g., PMID: 17984319), and it might be useful to discuss this aspect.

- In Figure 3B/C, the number of analyzed cells seems much lower than in Figure 3D (based on the points shown in Figure 3C, it would be good to have the number in the legend). Could the large dataset used in Figure 3D to quantify HP1 fluorescence be used to analyze the colocalization that is assessed Figure 3B/C? It would be useful to have better statistics for the colocalization analysis.

- How many chromocenters were used for the analysis in Figure 4A? Also, I wonder why the number of analyzed chromocenters varies so much between panel B (100/105 chromocenters) and panel C (2849/3209 chromocenters).

We thank the four reviewers for their constructive and encouraging comments. We have addressed all of the points raised, and have prepared a point-by-point response. Please note that the additions made to the revised manuscript are highlighted in blue text in the accompanying manuscript file.

Reviewer #1

In this manuscript, Novo et al investigated the roles of major satellite repeat (MSR) RNAs in the regulation of constitutive heterochromatin domains (chromocenters) in mouse ES cells (mESCs). The authors applied transcription activator-like effector (TALE) system to trace MSR sequences, thereby analyzed biophysical properties of heterochromatin condensates. Via in vitro droplet formation assay, they showed that forward MSR RNA promoted HP1a phase-separation. Furthermore, the authors indicated that depletion of MSR transcripts in mESCs resulted in the more compacted heterochromatin, and the increase of hallmarks of chromosomal instability.

The roles of MSR transcripts in heterochromatin organization had been recently studied by other groups (eg. PMIDs 28760199, 31677973, 33296675). Although this study was well-conducted and results are interesting, some conclusions remain premature, and some data are inconsistent.

Major points:

1.1: The authors concluded that “depleting MSR transcripts causes heterochromatin to transition into a more compact and static state”, mainly based on imaging and FRAP data. They showed that HP1a and H3K9me3 foci became smaller and tighter in MSR reduced cells (Fig 3B), but condensates indicated by TALE-MSR-mClover were larger (Fig 2B); this inconsistency should be explained. It is important to compare these markers in the same cells to obtain more convincing data. Moreover, as shown in the lower panel of Fig. 3B, the colocalization between H3K9me3 and HP1a was somewhat decreased in MSR RNA reduced cells, suggesting these smaller foci could be less chromatin-associated. A recent study showed that the condensate properties detected by DNA are different from those detected by chromatin-associated proteins including HP1a (PMID: 33326747).

We agree with the reviewer that it is important to assess the effect of MSR RNA degradation in controlled experiments. For that, we have compared chromocenter features (size, heterochromatin marks, molecule mobility) in TALE-MSR-mClover expressing ES cells (ESCs) that were transfected in parallel with either GFP (control) or MSR-specific gapmers. Importantly, the expression of TALE-MSR-mClover itself does not affect the size of condensates, as shown by comparing chromocenter diameters in ESCs with and without TALE-MSR-mClover expression (Figure S1D).

The images in Figure 2B demonstrate the recovery of the TALE-MSR-mClover signal in the FRAP experiments. We agree that in the particular image shown, it does look as though the boxed chromocenter in the MSR gapmer-treated cells is slightly larger than the boxed chromocenter in the GFP gapmer-treated cells. However, when the size of >100 chromocenters were quantified by linescan analysis, the difference is clear whereby

chromocenters are significantly smaller following MSR gapmer treatment (Figure 4C). This result is consistent with the super-resolution image analysis in Figure 3B, where we observed that the heterochromatin marks HP1 α and H3K9me3 are more compact upon MSR gapmer treatment as compared to the GFP-gapmer control. Also, please note that in the lower panel of the previous Figure 3B, the x-axis scale was longer in the control compared to the MSR gapmer-treated condition. To avoid potential confusion, we have now updated Figure 3B to show the data on the same x-axis scales. By making this change, we also hope that it is now clearer that the HP1a and H3K9me3 signals colocalize in both conditions, and importantly that the intensity profiles are higher and show a stronger colocalisation in the MSR gapmer-treated cells compared to the control GFP gapmer-treated cells (Figure 3B). This finding is consistent with the increased chromatin association of both heterochromatin markers (shown in Figures 3E and 4E) and with higher levels of DNA compaction (Figure 3F). These results suggest that the smaller chromocenters in the MSR gapmer-treated cells are more chromatin-associated, as compared to the chromocenters of control ESCs.

The authors also examined HP1a and H3K9me3 binding at MAR using ChIP-qPCR (Figs. 3E and 4E). However, they normalized ChIP data to IgG, which is heavily variable among antibodies; the input (total DNA) should be better for normalization. To evaluate the degree of chromatin compaction at MSR, more heterochromatin makers (e.g. H4K20me3, SUV39 etc.) should be examined and complementary assays, such as DNase & MNase digestion would be needed.

We have determined the levels of heterochromatin by assessing two key markers (HP1 α , Figure 3E and H3K9me3, Figure 4E); both show very similar increases at major satellite chromatin following MSR gapmer treatment. Input and IgG samples account for different types of biases in ChIP experiments. Input samples account for inherent differences within chromatin regions that may interfere with the shearing step of the ChIP protocol, whilst IgG samples are important to identify non-specific binding of the beads or the chromatin itself. As we are comparing the ability of different proteins to bind to the same chromatin loci, in our opinion, the use of a non-specific antibody (IgG) is a more appropriate way to normalise the ChIP-qPCR data in this experiment. However, for completeness, we show in Reviewer Figure 1 below the same data but this time normalised to input. Regardless of the normalisation method used, treatment with MSR gapmers induces an increase in heterochromatin marks at pericentromeric repeats.

Reviewer Figure 1 (next page): Histograms of ChIP results show the fold-change over input of H3K9me3 (upper) and HP1 α (lower) bound at MSR DNA or LINE DNA upon transfection of ESCs with GFP or MSR gapmers. Two biological replicates are shown side by side.

To study chromatin compaction, we measured the fluorescence intensity and size of DAPI-stained foci (Figure 4B and 4C), a dye that strongly intercalates with DNA and is commonly used to assess chromocenter organisation. Moreover, we additionally used a recently developed method that is ideally suited to our experiments because it measures chromatin compaction and provides spatial resolution (Hockings et al.), thereby enabling us to measure chromatin compaction specifically at chromocenters (Figure 3F). As described by Hockings et al., this method has been thoroughly optimised in cell lines treated with agents that alter chromatin compaction (for example, HDAC inhibitors) and carefully validated in multiple distinct systems including chromatin compaction during ESC differentiation. Using this sensitive assay, our results show that pericentromeric heterochromatin, but not euchromatin, is compacted following treatment of ESCs with MSR gapmers, which is consistent with the associated changes in HP1α and H3K9me3 at major satellite sequences.

1.2: The authors concluded that “satellite RNA scaffolds heterochromatin organization” in mESCs, but there was no sufficient data supporting this strong claim. Although Fig 3A demonstrated nicely that forward MSR RNA promotes phase-separation of HP1α, the molecular mechanism was not examined. For example, why did only forward RNA promote phase-separation? What is the sequence difference between forward and reverse? What are the DNA motifs recognized by HP1α? Transcripts from which direction are more abundant in chromocenters in mESCs? Do they colocalized with HP1α in the cells? TALE-MSR-mClover can be applied to image DNA but not RNA; RNA-FISH should be applied to detect MSR RNAs in the cells. The results of Fig. S3A are interesting. Could the authors explain why did 2 repeats of MSR promote the formation of larger HP1α droplets than 1 repeat and 8 repeats did?

The reviewer raises several interesting points that we have used to expand our discussion (page 13).

We thank the reviewers for their close attention to the data presented in Figure S3A. When the reviewer mentions that “*only forward RNA promotes phase-separation*” we presume they are referring to the larger droplets formed with the forward RNA as compared to the reverse RNA. Critical concentration (i.e. the lowest concentration at which phase separation occurs) is typically used in the field as a quantitative indication of phase separation efficacy. By this measure the forward and reverse RNAs have comparable critical concentrations for each length. On the other hand, droplet size is thought to be determined by a combination of many different variables arising from the kinds and types of interactions made within droplets, including structural factors of the RNA itself (e.g. how compact the RNA is within a droplet).

To date no strong sequence specificity has been observed for binding of HP1 α to nucleic acids. However, since HP1 α compacts nucleic acids, we speculate that there may be some sequence bias related to the bendability of the DNA or RNA and that HP1 α may be responsive to structural determinants inherent to the DNA or RNA sequence. Such dependence on structure may in turn affect the sizes of droplets formed with the forward vs. reverse strands. Indeed, previous biochemical studies demonstrated that the forward strand interacts more strongly with heterochromatin-associated proteins, such as HP1 α and SUV39H2 (Maison et al., 2011; Muchardt et al., 2002). Additionally, HP1 α has been shown to preferentially bind to parallel G-quadruplexes as opposed to anti-parallel ones, indicating that HP1 α is responsive to structural determinants arising from RNA sequence (Roach et al., 2020).

In terms of abundance, previous studies have demonstrated that in mESCs the forward strand MSR transcripts are more abundant compared to the reverse strand (Camacho et al., 2017). Importantly, it is well established using RNA-FISH that MSR transcripts are retained locally at chromocenters, including in mESCs, and that the MSR transcripts overlap with HP1 α signal (Maison et al., 2011; Camacho et al., 2017; Tosolini et al., 2018). Thus, our *in vitro* data together with substantial evidence from the literature show that the ability of forward transcripts to bind HP1 α is consistent with a strand-specific function for forward pericentric transcripts in chromocenter organization.

In terms of the why the 2 repeats of MSR promote the formation of larger HP1a droplets than 1 repeat and 8 repeats, again we speculate that the specific structures promoted by the 2 repeats may drive larger droplet formation.

1.3: It is interesting that MSR RNA reduction cause the increase of rH2AX staining at chromocenters (Fig. 5C). If chromocenter became more compacted as the authors suggested, how can DNA be damaged? This is related to point 1.

Previous studies have shown that the level of chromatin compaction impacts the detection and/or repair of damage by the DNA repair machinery, exemplified by the accumulation of mutations at highly compacted, H3K9me3-rich heterochromatin domains (Schuster-Bockler & Lehner, 2012; Schep & van Steensel, 2021) and by their slower rates of DNA repair (Goodarzi et al., 2008; Noon et al., 2010). It is possible, therefore, that the increased

compaction of chromocenters following the reduction in MSR transcripts could result in the accumulation of DNA damage, as evidenced by increased gH2AX signal.

1.4: Fig. 5D. The ChIP qPCR data are highly variable among replicates. As I mentioned above, the ChIP data should be normalized to input not IgG. Further, it is better to present control and Dox+ data separately, so that one can evaluate the enrichment among repeat elements.

As requested, we have split the +Dox and No Dox data (please see Reviewer Figure 2 below). However, we think that separating the data could confuse the readers as there are many more conditions to compare and the differences are not as clear. Importantly, we do observe increased H3K9me3, HP1 α and SUV39H1 at major satellite DNA in all three replicates, however the magnitude of the ChIP signals vary between different replicates, as is very commonly observed when examining repetitive regions. For reasons of clarity, we would therefore prefer to show these data as fold change.

Reviewer Figure 2: Histograms of ChIP results show the fold-change over IgG of three heterochromatin marks (H3K9me3, HP1 α and SUV39H1) at major satellite, minor satellite and LINE DNA after TALE-MSR-KRAB doxycycline-induction (DOX; right) relative to no doxycycline control (No DOX; left). Each dot represents a biological replicate. The red dashed line indicates a value of 1.

1.5: Seeing that the authors applied TALE-MSR-KRAB system to silence major satellites. The authors might want to examine the changes of chromocenter and compared with the results obtained by gapmers.

We thank the reviewer for this suggestion. We used the TALE-MSR-KRAB system as it is a suitable approach to increase localised heterochromatin and therefore disrupt the transcription of targeted regions. Indeed, upon the induction of TALE-MSR-KRAB we observed a decrease in MSR transcripts. Importantly, however, the recruitment of KRAB and

the associated increased heterochromatinization also led to the strong coalescence of chromocenters into single, very large foci. Because the recruitment of the KRAB domain strongly impacts local chromatin and likely interferes with RNA Polymerase progression, these confounding effects raise caution in interpreting the changes to heterochromatin architecture that are induced by the KRAB system. Thus, we prefer to adopt a cautious approach and to focus on evaluating the changes in chromocenter organisation following gapmer treatment, as this method only interferes with transcript levels.

Minor questions:

1.6: Figure 1D. Why the data during recovery stage is discontinued?

In the FRAP experiments, we have performed live-cell imaging experiments by acquiring images every one second. In Figure 1D, each data point represents the value of fluorescence intensity obtained at one second intervals, both before and after the photobleaching event (indicated by the vertical red line). Continuing to acquire images after the photobleaching event allowed us to measure whether the fluorescence signal can recover within the photobleached foci and, if so, how much signal is recovered. We stop the image acquisition when the fluorescence signal reaches a plateau, and before any fluorescent intensity changes could be attributed to photobleaching.

1.7: Figure S2A. To quantify transcript levels, at least three independent experiments are needed.

We concur with the reviewer and we have added more experimental data to Figure S2A so that all mESC samples now have at least three independent replicates. These experiments are highly consistent and support the i) increased level of MSR transcripts in ESCs compared to fibroblast cells (also shown by others: Efroni et al., 2008; Tosolini et al., 2018; Percharde et al., 2018) and ii) the reduction in MSR transcripts that occurs following MSR gapmer treatment.

1.8: Figure S3B. This figure looks strange. HP1 α is the reader of H3K9me3, but why they are not co-localized? This is inconsistent with that of Fig 3B.

The images shown in Figure 3B are single optical sections from 3D-SIM, whilst the images shown in Figure S3B are both single optical sections (top panels, labelled "3D-SIM") and the rendered volumes derived from all nuclear optical sections acquired (bottom panels, labelled "3D-volumes"). The reviewer raises the interesting point that these heterochromatin marks would be expected to be colocalised. Colocalisation reflects the resolving power of the imaging system. 3D-SIM enhances the spatial resolution and allows detailed structural information on the interaction/distribution pattern of proteins known to colocalize by confocal microscopy. Using standard confocal microscopy (resolving power ~ 250 nm) the H3K9me3 and HP1 α signals appear to colocalise (Figure 3D), but using 3D-SIM (resolving power ~ 120 nm) the signals appear mainly separated. This indicates that most HP1 α and H3K9me3 labelled molecules are separated by a distance >120 nm but less than ~ 250 nm and it is an observation that we intend to investigate further.

Reviewer #2

The work of Novo et al reports on role of the major satellite repeat transcripts (MSRtr) in maintenance of semi-open chromatin state of chromocenters in mouse ESCs and chromosome stability. Using TALE-GFP targeted to MSR, they show dynamics of the chromatin within chromocenters, as well as dynamics of chromocenters themselves, which resembles liquid-like membraneless condensates behavior. Depletion of MSRtr leads to decrease in dynamics of chromocenter chromatin, concomitant with increase of HP1 α , H3K9me3 and chromatin compaction, as well as to increase of chromocenter number. Based on the ability of MSRtr to bind HP1 α , which was demonstrated by droplet assay in vitro, the authors speculate that satellite transcripts scaffold HP1 α in chromocenters and prevent its binding to satellite DNA impeding chromocenter condensation. The authors noticed that MSRtr depletion results not only in chromocenter structure changes but also influences chromosome integrity, in particular, leading to Robertsonian translocations and pericentromeric undercondensation. The paper is well written, sufficiently illustrated and data are comprehensively discussed. The messages of the paper are interesting to the nuclear biology field and the paper is potentially a good fit to Nature Communication journal, if some technical issues listed below are attended.

I have two major concerns for the experimental part.

2.1: Firstly, when the authors describe chromocenter structure or their number per nucleus, their assessment is based on DAPI staining and thus can miss small chromocenters (e.g., Fig.4) or immunostaining that stains not only chromocenters (e.g., Fig.3). The methods of choice in such assays are those which specifically detect subcentromeric satellite – either DNA-FISH with MSR probe or targeting MSR with TALE-GFP (see also minor comment 9)

In mouse cells, chromocenters can be easily identified by their DAPI-rich staining that clusters into large nuclear foci, as shown in the manuscript by co-localization with MSR-TALE-mClover (Figures 1A, S1A and S2D) and by DNA-FISH (shown in Reviewer Figure 3 below and new Figure S1C). Additionally, interphasic centromeric DNA forms separate, much smaller foci (Guenatri & Almouzni, 2004; Probst & Almouzni, 2008; please see Reviewer Figure 4 below). Importantly, chromocenters are quantitatively similar when examined using DAPI or TALE-MSR-mClover (please see Figures S1D-E), demonstrating there is strong concordance between the methods.

Reviewer Figure 3: DNA-FISH analysis of ESCs using probes for major (upper) and minor (lower) satellite DNA. Images are counterstained with DAPI.

Reviewer Figure 4: Images from Probst and Almouzni 2008 show distinct DNA-FISH signals for major and minor satellite sequences.

2.2: Secondly, if the author believe in the role of MSRtr in organization of chromocenters, they need to demonstrate presence and distribution of the RNA in chromocenters by RNA-FISH. This is especially important in the view of the authors' suggestion that MSRtr compete with MSR for HP1 α binding. If their suggestion is correct, then the MSRtr signals must be really abundant and distributed through the entire chromocenter volume in untreated ESC, while this signal will be significantly reduced after MSRtr depletion. In addition, it is important to carry control RNA hybridizations after RNasing.

We agree with the reviewer that the localization of MSR RNA in chromocenters is an important aspect of our model. Multiple studies, including in mouse ESCs, have established using RNA-FISH that the vast majority of MSR transcripts remain associated with pericentromeric regions of mouse chromosomes and that the transcripts are distributed through the entire chromocenter volume (Martens et al., 2005; Lu and Gilbert, 2007; Maison et al., 2011; Bulut-Karslioglu et al., 2014; Ishiuchi et al., 2015; Camacho et al., 2017). The RNA-FISH signal for MSR transcript disappears following RNase treatment (Maison et al., 2011). Additionally, MSR transcripts are essential for chromocenter formation (Probst et al., 2010; Casanova et al., 2013) and MSR RNA has a higher affinity compared to MSR DNA for HP1 α binding (Muchardt et al., 2002; Maison et al., 2011). Importantly, incubating cells with RNase A for 30 minutes prior to fixation results in the loss of HP1 α signal, suggesting that the complete degradation of MSR RNA leads to HP1 α dislodging from chromocenters (Maison et al., 2002). Our model builds on these previous findings to propose that MSR RNA levels are important in modulating the substrate for HP1 α interaction at chromocenters in mouse ESCs, whereby high MSR transcript levels decrease HP1 α binding to chromatin (HP1 α remains at chromocenter foci) and help to maintain an uncompacted local structure.

I also would like to mention that the supplementary videos are uninformative or wrongly interpreted (see below).

We address this comment below. Please see our response to comment 2.6.

Minor comments:

2.3: Supplementary Figure 1C. The histogram shows percentage of nuclei with 1, 2 and 3 chromocenters. As one can see from other figures in the paper, as a rule, ESCs, have more than 3 chromocenters.

We thank the reviewer for highlighting this. The data in the histogram is from analysis that was performed on single optical sections. We have changed the axis label and the figure legend to clarify this point.

2.4: Supplementary Figure 1D. How the authors define a diameter of a chromocenter? Even on projections, chromocenters have very irregular shape. Besides, to define a border of a chromocenter, one needs to threshold an image. How thresholds were set up for different slides with different acquisition conditions? It is not clear from the legend, e.g. how many nuclei were assessed in the three independent experiments – 3 x 16 or 16 in general? In any case, this statistic is pretty poor.

DAPI linescan analyses were performed in single optical sections where DAPI foci were at the mid-focal point. Fluorescence intensity histograms were generated with a linescan across the nucleus. The background signal (outside of the nucleus) was subtracted from the baseline fluorescence within the nucleus and from the signal within the chromocenter. Chromocenter borders were then defined at the base of each fluorescence peak above the nuclear background. In total, 16 and 25 nuclei were analysed from three independent inductions of TALE-MSR-mClover.

2.5: Figure 1C. Are those projections or single optical section? In the latter case, the changes in shape of chromocenters can be explained by a chromocenter (nucleus) movement with parts of chromocenter coming out and in focus.

The images shown are projections and we have clarified this in the figure legend. We have also improved the videos to better demonstrate the fusion/fission events.

2.6: Video 2a and 2b suppose to show “rapid movement of chromocenters” however both of them show merely cell movements, whereas relative positioning of chromocenters does not change.

Video 2 and 3 suppose to show separation and coalescence of chromocenters, however, the videos do not show these phenomena.

We agree that the videos submitted in our original manuscript were not as clear as they could be. We have generated new videos from the same data, but this time focusing only on these specific observations to visually help the readers with these results.

2.7: Page 8. “...a molecular basis for understanding why HP1a-associated heterochromatin disperses into smaller condensates after MSR RNA levels are depleted (Figure 3B)”. The Figure 3B does not show this phenomenon but rather illustrates changes in chromocenter structure.

We concur with the reviewer and have changed this phrase to “are smaller after MSR RNA levels are depleted”.

2.8: Page 9 (Figure 3B). To compare overlapping signals of H3K9me3 chromatin and chromatin binding HP1a, the authors have chosen to compare volumes of the signal. This is not an appropriate method, since volume rendering strongly depends on a threshold, which in this case has to be set separately for two different channels, individual cells and different acquisitions. The colocalization has to be shown and estimated on single optical sections through mid of chromocenters.

HP1 α is a reader of H3K9me3 and thus is highly co-localised when imaged by confocal microscopy, as can be seen in Figure 3D. In Figure 3B, we used the enhanced resolution of structured illumination microscopy (SIM) to have a better understanding of the distribution of each heterochromatin marker within chromocenters. We have clarified in the Methods section that we acquired a series of raw images with 120 nm step intervals that were then reconstructed into the high resolution image shown using a 3D-SIM algorithm. The Nikon 3D-SIM reconstruction algorithm used has minimal parameters and does not require a set threshold to define the limit of the signal and is therefore not subject to the potential bias that the referee suggests. At this higher resolution, we opted for measuring the overlap of H3K9me3 and HP1 α by rendering volumes for each channel with IMARIS software (with the same threshold in each channel) and quantifying how much volume from one marker falls within the volume of the other marker and vice-versa.

2.9: Figure 3D. This figure does not convince that brightness of the fluorescent signal is different in two conditions. Authors have to explain (a) how the HP1a fluorescence was measured – on projections or single sections, for individual cells or group of cells – and (b) how can they be sure about equal conditions of staining and image acquisition from two different slides, which is essential for such comparison.

We have updated the Methods section to clarify that the image acquisition conditions were maintained between slides (each slide was acquired from independent replicate experiments), so that the technical variation between replicates could be accounted for, including across replicates. The fluorescence intensity was measured on single sections in groups of cells, in chromocenters firstly defined by DAPI (most intense foci).

2.10: Figure 4A. Firstly, how many nuclei were analyzed in this experiment? Secondly, the histogram shows number of chromocenters from 1 to 6. However, the very image on the left shows 8 chromocenters. I am working with various mouse cells types half of my scientific career but I never saw mouse somatic cells with just one chromocenter. I think the authors are mistaking because they rely only on DAPI staining that might be misleading, especially in case of ESCs. To make a correct assessment of chromocenter number, a DNA-FISH with a

probe for MSR, specifically detecting subcentromeric heterochromatin, has to be employed.

A total of 55 and 36 nuclei (100 and 105 chromocenters, respectively) were quantified following the transfection of ESCs with GFP or MSR gapmers. The reviewer correctly points out that the x-axis was wrongly labelled (thank you), as it refers to the number of chromocenters per single focal plane rather than per nucleus. We have corrected this in the updated figure.

As a more general note, a previous study showed that the number of chromocenters per nucleus of mouse ESCs ranges between one and nine with a median number of four (Meshorer et al., 2016), which is very similar to the numbers that we report in our study. Also, as described in response to comment 2.1, we show that the number of pericentromeric foci are highly concordant when comparing data obtained from DAPI labelling, MSR DNA-FISH and MSR-TALE-mClover (new Figure 4A and Figure S1A, respectively). We agree with the reviewer that the low numbers of chromocenters are not observed in most mouse somatic cells, but rather it is an interesting feature that is characteristic of mouse pluripotent cells. As described in our manuscript, we believe that the higher levels of MSR transcripts in ESCs compared to somatic cells helps to modulate the properties of pericentromeric heterochromatin and chromocenter organisation.

2.11: Figure 4B,D and C. How DAPI and H3K9me3 intensities were measured? – see also (8). I wonder how the authors measured chromocenter diameters? – see also comment (2)

We agree that further clarification would be useful and we have included more details on this analysis in the Methods section. DAPI linescan analyses were performed in single optical planes where DAPI foci were at optimal focal planes. Fluorescence intensity histograms were generated with a linescan across the nucleus, and the background (outside of the nucleus) was subtracted from the baseline fluorescence within the nucleus and the chromocenter signal. Chromocenter borders were then defined at the base of each fluorescence peak above the nuclear background.

For H3K9me3, the fluorescence intensity was measured in regions of interest pre-defined by DAPI (most intense foci) in single sections with groups of cells. The conditions for image acquisition were maintained between slides (each slide was acquired from independent replicate experiments), so that the technical variation between replicates could be accounted for.

2.12: Figure 5B. The authors cannot claim that the very left panel shows a fusion of arms of two different chromosomes – the chromosomes simply got close to each other during spreading; fused chromosomes have different configurations on spreads. Fusion of chromosomes by their centromere regions is called Robertsonian translocation and should be named like this in the MS.

Figure 5F: “cohesin defects” has to be changed to “cohesion defects”.

We thank the reviewer for these helpful points, which have now been corrected in the revised manuscript.

Reviewer #3

In this work, Lopes Novo et al investigate the role of major satellite repeat transcripts in organization and biophysics of the constitutive heterochromatin domain in mouse embryonic stem cells. They demonstrate that the presence of these repetitive transcripts affects compaction and material state of heterochromatin. Most interestingly, they connect these changes in biophysical state to functional outcomes of heterochromatin disruption, including genome instability and DNA damage accumulation. This work is novel in that it is attempting to connect molecular mechanism to biophysical state to functional outcome of an important chromatin compartment that is of broad interest. If the authors address the points raised below, I recommend it for publication in Nature Communications.

Overall, the use of novel tools like TALE-MSR-mClover allows for unique investigations of heterochromatin structure and function in live cells. However, I urge caution to the authors with regard to interpretation of data taken on this synthetic protein in terms of what it means for the endogenous heterochromatin domain.

3.1: For example, changes in FRAP recovery of the TALE-MSR-mClover upon gapmer knockdown of MSR RNA does indicate that presence of the RNA affects mobility of the TALE-MSR-mClover protein, but is not necessarily generalizable to other proteins. This criticism can be remedied by placing more emphasis on changes measured in proteins that are endogenously present in the domain (i.e. HP1 α), or by taking FRAP measurements of other proteins present in heterochromatin (i.e. SUV39H1).

Additionally, I encourage the authors to further elaborate on the discussion of multiple populations of the same molecule with different mobility (e.g. mobile and immobile fraction of HP1 α) and differential mobility of types of molecules in the domain (DNA vs. RNA vs. protein) that lead to the emergent material state of the whole domain.

We thank the reviewer for encouraging us to elaborate on these interesting points, and in response we have added new text to the discussion (page 15).

We also would like to highlight that we have confirmed that MSR RNA affects the mobility of the heterochromatin-bound protein HP1 α , as measured by FRAP experiments using fluorescently-tagged HP1 α (Figures S2E-H). The results of this experiment are very consistent with the data that we obtained using TALE-MSR-mClover. Our rationale to use the TALE-MSR-mClover reporter was to assess the dynamics of pericentromeres with a reporter that would be unaffected by changes to the heterochromatic state that could be caused in response to altering MSR transcript levels. However, we do agree that assessing an intrinsic component of heterochromatin would strengthen our conclusions, and that is why we have also assessed chromocenter dynamics with HP1 α . By doing so, we confirmed that TALE-MSR-mClover is a legitimate reporter to follow the biophysical properties of chromocenters. Additionally, because TALE-MSR-mClover binds uniquely to our region of interest, this reporter system has a reduced background signal and a higher specificity when compared to the use of intrinsic components of heterochromatin, like HP1 α . As suggested by the reviewer, the use of complementary approaches further strengthens our conclusions that MSR transcript levels affect heterochromatin dynamics.

Specific comments on figures below:

3.2: Figure 1. The chromocenters marked by TALE-MSR-mClover in these ESCs are highly dynamic (1C). In addition to FRAP, which gives measurement of mobility of individual components, the timing of fusion and rounding up of whole domains can give measurements of the viscosity of the domain and surrounding nucleoplasm, which will further support the 'gel-like' or 'liquid-like' picture presented in Figure 6. From these data or similar movies of fission / fusion events, perhaps a viscosity can be measured following methods similar to those used for nucleoli in Caragine, Haley & Zidovska, eLife 2019 (<https://elifesciences.org/articles/47533>). This method requires high resolution images for accurate measurement of the bud neck, and I am not sure what types of microscopy are available to the authors, so I consider this an informative but not essential experiment for revision.

We appreciate the reviewer's comments on methods to infer viscosity from the microscopy analysis. However, as the reviewer correctly alluded, unfortunately our imaging facility is not currently equipped to perform this analysis.

3.3: Figure 1D. Is there a normalization happening for this FRAP trace? In Fig 2 the FRAP data seems normalized but not here in 1D.

The results in Figure 1D have not been normalised because we wanted to show separately the unprocessed data for the bleached and unbleached chromocenters. We have now clarified this in the figure legend. The intent of Figure 2B was slightly different. As commonly applied to this type of data, the FRAP data in Figure 2B have been normalised in order to overlay the signal recovery in cells treated with either GFP or MSR gapmers. In this context, normalisation further helps to visualise and easily compare the differences.

Figure 1E: Please indicate the time post-bleach at which the immobile fraction was calculated in the figure legend.

We have added the time post-bleach for the immobile fraction calculation to the figure legend (250 seconds post-bleach).

3.4: Figure 2B. Please indicate both on the figure and in the legend what protein is being bleached (TALE-MSR-mClover). Also, in the legend make sure to report the total number of bleach areas used to calculate the FRAP curve, similar to how it was reported for 1E. I see only number of trials reported.

Thank you. We have updated the manuscript according to the reviewer's comments.

3.5: Major question. Does the GFP gapmer used here as a control knock down levels of TALE-MSR-mClover, and could this affect the protein's mobility? The methods section suggests that the GFP gapmer does not match the mClover sequence used in the TALE-MSR-mClover.

Was any control experiment done (i.e. western blot of TALE-MSR-mClover with and without gapmers)?

Please also write in the text that the actual fluorophore is mClover and the gapmer should not bind.

Additionally, is the TALE-MSR-mClover protein capable of binding dsRNA? Later the authors suggest a molecular mechanism for MSR RNA abundance modulating HP1's mobility by competing for binding with DNA. Does that model also apply to the synthetic protein?

The GFP gapmers are very unlikely to affect TALE-MSR-mClover in these experiments. As mentioned in the Methods section, there is no overlap between the nucleotide sequences of GFP gapmers and the *mClover* sequence. We would also like to emphasise the TALE-MSR-mClover reporter is induced 24h after the third and final GFP gapmer transfection, and thus we do not expect any targeting of GFP gapmers to the reporter transcripts. In addition, GFP intensity was not reduced after GFP gapmer transfection compared to no transfected experiments, and thus we are confident that GFP gapmers do not target the TALE-MSR-mClover reporter. Lastly, the MSR target sequence in the TAL protein we use contains 5 'T' bases (out of 15 bases in total) and it is unlikely that the TAL protein could recognise the dsRNA containing 'U' as these positions. TAL proteins were shown to be unable to bind dsRNA (Ying et al., 2012). Thus, to our knowledge, TALE-MSR-mClover can only bind to DNA.

3.6: Figure 3. Major point: for these cell-specific assays like SIM, is there a marker of which individual cells received the gapmer? Or a control of what percent of cells receive the transfection?

Unfortunately not. To try and mitigate this, we performed three successive gapmer transfusions in each replicate experiment and confirmed an average 50% reduction of MSR transcripts for every single replicate.

3.7: Figure 3A. Because forward and reverse transcripts of MSR have different effects on HP1 α 's phase behavior in vitro, we might expect them to differentially influence its behavior in vivo. Have the authors treated cells with a single gapmer that would knock down only forward or only reverse transcripts? Also, have the authors tried mixing both strands with HP1 α in vitro? Are MSR transcripts thought to create dsRNA in vivo?

We did not perform single gapmer experiments. However, a study in early embryonic development, where each MSR strand was individually downregulated, showed that the reverse RNA strand is essential for clustering of pericentric satellites into chromocenters at the late two-cell stage (Casanova et al., 2013), suggesting strand-specific roles.

We have also followed the reviewer's suggestion to see how double-stranded MSR RNA would impact HP1 α -droplet formation in vitro, but unfortunately technical difficulties arose that could not be solved within a reasonable timeframe so that the in vitro droplet formation assay would be comparable to the previous experiments performed. However,

we now provide polyU/A dsRNA data (please see Reviewer Figure 5) showing that the dsRNA lowers the critical concentration of HP1 α -droplet formation.

Reviewer Figure 5: PolyA/U dsRNAs can contribute to HP1 α droplet formation *in vitro*. Images framed in green contain visible droplets. All images show the same magnification; scale bars, 100 μ m.

Figure 3B, bottom: If you make the scale of the x axis (Distance) the same for left and right, it will highlight the size/compaction difference between the two conditions more.

We thank the reviewer for this suggestion, which has now been added to the manuscript.

Figure 3C: There are no statistics on the volume colocalization method used here and the variance is high, are these changes statistically significant? Could you use another method like radially averaged autocorrelation to confirm?

To extract further structural information from the 3D-SIM images, we have performed an analysis that is similar to radially averaged autocorrelation: we have defined territories for each channel and quantified the percentage of the volume of the H3K9me3 channel that falls within the volume defined in the HP1 α channel and vice-versa. By applying this, we were able to correlate both signals and show that this correlation consistently increased upon MSR gapmer treatment.

Figure 3F: Please include in the figure legend that shorter fluorescence lifetime means increased chromatin compaction.

Good suggestion - thank you. We have added this to the revised manuscript.

3.8: Figure 4.

4A: Please indicate number of nuclei per trial in figure legend for 4A. I assume the same raw data was used to quantify different metrics in 4A, B and C?

The same raw data was used for all analysis shown in Figure 4. In total, we have analysed 100 and 105 chromocenters following the transfection of ES cells with GFP and MSR gapmers, respectively (55 nuclei in GFP-gapmer and 36 nuclei in MSR-gapmer conditions). We have now added this information to the Figure 4 legend.

4B, C, D: Within the figure, chromocenter is spelled 'chromocentre', please use one consistent spelling.

We thank the reviewer for spotting this and we have corrected the figure and legend.

Figure 4E: The authors indicate that the pattern of more, small chromocenters instead of few larger ones could be indicative of differentiation, but also show earlier in S2B and S2C that pluripotency markers are unaffected. So are these uncoupled phenotypes?

We thank the reviewer for this interesting question. Indeed, the pattern of more numerous and smaller chromocenters are characteristic of differentiated cells. In our experiments, a similar pattern is observed following MSR transcript depletion, but the ESCs have not initiated differentiation, at least within the experimental timeframe. Interestingly, we reported similar changes in chromocenter organization in *Nanog*-deficient ESCs and these cells also remain undifferentiated and pluripotent (Novo et al., 2016). Thus, it seems that the phenotypes can indeed be experimentally uncoupled. We propose that an untimely dysregulation of chromocenter organization has major deleterious effects in genome stability of ESCs.

3.9: Figure 5.

Figure 5B: In the right-most image, the width / length of the mitotic chromosomes is significantly different from the other three examples here; what is going on?

During the preparation of metaphase spreads in mouse ESCs (which have a large proportion of cells in S phase), a short 30 minute colcemid treatment was performed to generate sufficient numbers of cells in mitosis. As a result of this treatment to arrest metaphasic chromosomes, cells at different stages of cell division are blocked and this results in noticeable differences in the length and compaction of chromosomes, such as in the

examples mentioned by the reviewer. All images analysed were prepared from the same experiment but cropped differently to better highlight the mitotic defect shown.

Figure 5C: Why are the quantifications grouped by image instead of displaying a point for each nucleus, or a violin plot representing cells equally? Without seeing the images or a summary of number of cells per frame, the reader doesn't know whether frames with a few cells are getting disproportionate representation.

As ESCs have naturally high levels of gH2AX (Banath et al., 2009), we have only considered gH2AX that is colocalized at chromocenters. We have updated this figure to show the percentage of cells with at least one gH2AX signal at chromocenters. In total, 487 and 467 cells were analyzed in the GFP or the MSR gapmer conditions, respectively.

Figure 5G: again the chromosomes look wildly different in terms of compaction, length/width. Is this a phenotype? It is not mentioned in the text.

We think it is a technical aspect of metaphase preparation. Please see the response to comment 3.9.

3.10: The model presented in Figure 6 suggests that increasing the amount of MSR RNA fluidizes the domain but is difficult to interpret from the image alone—consider highlighting purposefully drawn changes with labels like 'increased compaction'. Also, the color and placement of "Liquid-like" and "gel-like" chromocenter colors is confusing, it seems to insinuate formation of a liquid-like compartment surrounding a gel-like core, but I think it's really just representing mobile and immobile fractions of FRAP populations.

Thank you for the good suggestions. We have updated the figure accordingly.

3.11 Discussion. What do the authors interpret for both types of changes in FRAP data, T1/2 and mobile/immobile fraction? Increase in immobile fraction can indicate longer binding residence time on chromatin, while changes in T1/2 can more closely represent diffusion characteristics. Do the authors interpret these as both contributing to the material state of the heterochromatin domain? The discussion and interpretation of material state and its contribution to function are a bit under-developed. Again I encourage additional measurements of material state, for example viscosity, through the method mentioned above. And how would disruption of material state lead to these measured differences in chromatin organization and DNA damage accumulation? The authors elaborate beautifully on molecular mechanism of dynamics but the connection from molecular mechanism to chromosomal stability and function is slightly disappointing.

We thank the reviewer for these very interesting comments, which we have added to our discussion (page 15, line 13).

Regarding the material state of chromocenters, our microscopy facility is not suitable to analyse viscosity with the method proposed by the reviewer. However, we have followed the reviewer's advice to extract more information about the behaviour of chromocenters. We found that MSR transcript deletion reduces the diffusion coefficient of chromocenters (new Figure 2F) and makes the velocity of chromocenters less varied and more uniform (new Figure 2G). Indeed, these new analyses suggest that MSR transcript levels may modulate the material state of chromocenters: high levels increase the mobility of chromocenters and of the molecules within these domains, whilst reducing the proportion of the immobile population of both HP1a and TALE-MSR-mClover.

We would really like to be able to put forward a molecular mechanism connecting heterochromatin architecture and chromosomal stability. Towards this goal, we have investigated several potential avenues but so far the data have not supported any specific links and so we are reluctant to speculate on this. For example, there do not seem to be any changes in cell cycle or in proliferation rate following LNA gapmer treatment. Similarly, we found that CENPA protein binding (at centromeres or even at pericentromeres) was unaltered in ESCs upon chromocenter reorganisation. We have also examined mitotic DNA replication to assess if there are drastic changes in replication timing at pericentromeres, but we found no clear evidence for this. As the reviewer may appreciate, pinpointing the underlying molecular mechanism of DNA repair defects is extremely challenging, even at repetitive regions. Additionally, we have previously shown that activating endogenous MSR transcription leads to rapid (within ~6 hours) changes in chromocenter organisation, suggesting that changes in the cell cycle are not required for these effects (Novo et al. 2016). In sum, we feel that, in general, these are exciting follow up avenues that fall outside the current scope of the manuscript.

Reviewer #4

In this paper by Novo et al, the authors study the properties of chromocenters in mouse embryonic stem cells (ESCs) and investigate their response upon transfection of major satellite repeat (MSR) gapmers, which target and deplete MSR transcripts. The authors show that chromocenters in ESCs can undergo fusion/fission, and that MSR gapmer transfection leads to increased turnover of an ectopically expressed TALE that binds chromocenters, increased colocalization of H3K9me3 and HP1a at chromocenters, an increased number of chromocenters in the cell, and the appearance of hallmarks of genome instability. They combine these results with in vitro work showing that MSR transcripts promote phase separation of HP1. Based on their results, the authors suggest that MSR transcripts promote a liquid-like state at chromocenters in ESCs, and that this liquid-like state is lost upon MSR gapmer-induced depletion of MSR transcripts. Furthermore, they suggest that loss of the liquid-like state is responsible for the effects seen in response to MSR gapmers listed above.

While I find that this is a potentially interesting paper, I am not convinced that the effects observed upon MSR gapmer transfection are necessarily linked to the capability of MSR transcripts to promote a liquid-like chromocenter state in the cell. I find that the authors need more data if they want to convincingly make this point, and it would also be important

to compare their data to other models besides a “liquid condensate” model to test in a more unbiased way which model explains the data best. Specific suggestions are listed below.

Major points

4.1: *The fusion/fission events in Fig. 1C are maybe the most convincing piece of data showing that chromocenters in ESCs are liquid-like. It would be important to quantify this (how many fusions/fissions per chromocenter per time?) and to show how this number changes if MSR gapmers are transfected.*

We thank the reviewer for prompting us to quantify these events. Indeed, although we observed cleavage and coalescence events in both conditions, the percentage of tracked chromocenters engaging in these events almost halved in cells treated with MSR gapmers compared to the control GFP gapmers (~25% vs. ~40%, respectively). We have added these data as new Figure 2E.

4.2: *I do not follow the conclusion that the increased turnover of TALEs upon transfection of MSR gapmers means that MSR transcripts promote a “dynamic, physical environment within chromocenters”. An alternative (and potentially simpler) explanation would be that TALEs bind stronger to their target (MSR) DNA when MSR gapmers have been transfected, for example, because less MSR transcripts are produced and therefore less RNA Polymerase molecules run along the MSR DNA to counteract TALE binding. This should be considered/tested as an alternative explanation, for example by using a protein that binds other sequences in the chromocenter (= in the same dynamic, physical environment), like minor satellite repeats that do not change their transcriptional activity with the gapmers used here. Furthermore, if MSR transcripts create a liquid condensate, as implied by the authors, there would actually be a good chance that the viscosity of this condensate is higher than the viscosity of the liquid that fills the cell, so that the mobility of molecules within that condensate would decrease and not increase (the liquid that fills the cell is not much more viscous than water, see for example this review by Luby-Phelps that discusses viscosity measurements made with different techniques: PMID 10553280).*

The reviewer raises several interesting points. We believe, however, that the alternative explanation proposed by the reviewer whereby TALE proteins bind more strongly to pericentromeric DNA when MSR transcripts are targeted by gapmers is unlikely. Firstly, gapmer treatment does not affect RNA polymerase recruitment to the region, instead, it triggers an RNase H-dependent degradation of MSR transcripts at the 3' terminal regions of the pre-mRNA and lncRNA, reducing RNA expression without affecting RNA Polymerase II association with the gene or transcription termination (Lai et al., 2020; Lee & Mendell 2020). Secondly, we have directly tested the alternative hypothesis by using HP1 α , a protein highly abundant at chromocenters, recruited by H3K9me3 and potentially MSR RNA, and thought to be structurally bound to heterochromatin through its hinge domain (Maison et al., 2011). FRAP experiments on fluorescently-tagged HP1 α confirmed that MSR reduction by gapmers also impacted the dynamics of HP1 α at chromocenters (Figures S2F-H). Additionally, centromeric DNA locates at the periphery of chromocenters and is thought to

form a distinct structural entity that, importantly, is much smaller than pericentromeric DNA at chromocenters (please see the DNA-FISH images in Reviewer Figure 6 below).

Reviewer Figure 6: DNA-FISH analysis of ESCs using probes for major (upper) and minor (lower) satellite DNA. Images are counterstained with DAPI.

4.3: To better understand the effects seen upon MSR gapmer transfection, it would be important to make sure that these effects come indeed from the depletion of MSR transcripts in the cell, and not from some other effects that MSR gapmers might induce. For example, MSR gapmers could also alter the chromatin environment at chromocenters by binding to MSR DNA (potentially in transcription bubbles) or to nascent RNA transcripts, thus regulating Pol II occupancy, or by titrating away proteins etc. This could also account for some of the effects seen upon MSR gapmer transfection, especially as the reduction of MSR transcript levels seems moderate according to Fig. 2A (about 2-fold). Can the gapmer-induced effects be rescued by providing additional MSR transcripts? Do chromocenters become more liquid-like if MSR transcripts are added to differentiated cells? Can MSR transcript levels be modulated without transfecting MSR gapmers, for example, by expressing a TALE that is fused to a transcriptional repressor like KRAB, and are similar effects observed?

The reviewer asks if it is possible to provide additional MSR transcripts. This is a difficult experiment to perform because the transcripts operate in *cis* near to their sites of transcription, and therefore cannot be provided ectopically. Combining MSR transcript knockdown concomitantly with providing additional MSR transcripts is also very difficult to control accurately, especially in a background of endogenous high transcript expression, as in ESCs. We have previously performed a similar experiment in cells that express lower endogenous levels of MSR transcripts (Novo et al., 2016). Here, we showed that elevating endogenous MSR transcript levels by recruiting activators to pericentromeric heterochromatin causes rapid changes in chromocenter organisation such that chromocenters adopt a pattern that is typical of ESCs (i.e. fewer and larger chromocenters). These data therefore strongly support our current findings that MSR transcripts themselves have a role in modulating chromocenter organisation. Additionally, as mentioned by the reviewer, in the current manuscript we have modulated MSR transcript levels in ESCs by using a TALE-MSR-KRAB system, please see Fig 5D-G, however we do not think this is a well-controlled system to measure chromocenter changes (please see response 1.5 above).

4.4: How do the concentrations and the stoichiometry in the in vitro assays shown in Fig. 3A relate to those in the cell? This is a critical piece of information in order to understand if MSR transcripts are likely to create a liquid-like condensate at chromocenters, similar to what is obtained in the test tube. If the absolute number/concentration of MSR transcripts in ESC chromocenters is not known, I think it should be measured. Otherwise, it is hard to decide if it is likely that MSR transcripts act “directly” as a component of a condensate, or rather indirectly in some other way (for example, by changing the chromatin environment/histone marks/proteome at chromocenters).

In our *in vitro* assays, we used a range of HP1 α concentrations that include the known *in vivo* concentrations to date (1-10 μ M HP1 α , from Lu et al, 2000, Müller et al, 2009). Further, the values of critical concentration (the minimal concentration for phase-separation) we observe *in vitro* (Fig. S3A, 3-25 μ M HP1 α) fall within the biological range above. In terms of stoichiometry, our prior *in vitro* data suggests an approximate stoichiometry of one HP1-dimer for roughly a 65 base stretch of nucleic acid (Larson et al, 2017 & Keenen et al. 2021). Regarding the concentration of MSR RNA, the mouse genome has around 6 Mb of major satellite repeats, which corresponds to more than 25,000 repeats (234 bp). Each repeat is around 10 pmol and so if just 1% of repeats are transcribed, we would expect 2.6 nmol of MSR transcripts. To more fully explain and interpret the findings, we have included in the revised discussion the possibility that the effect of MSR RNA could be indirect, as suggested by the reviewer.

4.5: For the final model in Fig. 6, I am wondering how pronounced the differences between the two states actually are. What is the functional consequence of chromocenters showing fusion/fission in the left case (termed liquid-like, ESCs) but not in the right case (termed gel-like, differentiated cells), and what is the functional consequence of proteins binding a few seconds longer in one of the cases? Taking one step back, heterochromatin appears quite dynamic in ESCs and also in differentiated cells when it comes to binding of HP1 and other heterochromatin proteins that exchange within seconds as has been shown in the past and as is shown here, so the functional relevance of the distinction between two pretty similar (= two pretty dynamic) chromocenter states is not clear to me. In particular, it is not clear if these states are really linked to the hallmarks of genomic instability, or if this is just a correlation, and it would be important to strengthen this aspect.

These are all very pertinent observations but very challenging to address experimentally. In the final model, we try to propose a model combining all of the main experimental observations presented in the manuscript. As the reviewer mentions, heterochromatin is quite dynamic in ESCs and differentiated cells when considering the binding of HP1a or other heterochromatin proteins. However, if we add the i) increased compaction, ii) increased heterochromatinisation and iii) reduced size of condensates with less MSR RNA transcripts, we can perceive a model whereby the material state of heterochromatin is dependent on the MSR RNA abundance. Despite all of the avenues where we attempted to link changes to the material state of chromocenters and the genomic instability observed, it is still only an observation (albeit a potentially important one). For these reasons we decided not to include it in the model.

Minor points

4.6: *Would it be possible to determine the interfacial tension from the morphology of the fusion/fission intermediates shown in Fig. 1C? This could tell us something about the “condensate properties” of chromocenters.*

As mentioned in the response 3.11 above, the specifications of our microscopy facility are not sufficient to measure parameters such as viscosity or interfacial tension from the live image acquisition. However, we have extracted more information about the behaviour of chromocenters from the live imaging experiments that we can perform and found that MSR transcript deletion reduces the diffusion coefficient of chromocenters (new Figure 2F) and makes the velocity of chromocenters less varied and more uniform (new Figure 2G). As mentioned in the response to comment 3.11 above, these new analyses suggest that levels of MSR transcripts increase chromocenter mobility, and when depleted by gapmer treatment, a potential change to the material state of chromocenters occurs.

4.7: *It is not uncommon that DNA-binding proteins with low nM Kd values show dynamic FRAP recoveries in cells, see for example mCherry-LacI in Fig. 1E in Chong et al, PMID 29930090 (LacI binds lacO with Kd < 1 nM, but it recovers when bleached in the indicated paper). This might have many reasons, like competition with cellular proteins that bind to the same DNA sequence etc, and I therefore do not see how this is a readout for the “physical properties of chromocenters”, whatever the authors mean with this term. The turnover of a chromatin-associated protein at its target site is certainly not a readout of the local viscosity, as it will depend on many other factors (accessibility of the sequence, active processes counteracting binding, etc).*

We would like to highlight that our conclusions are based on the comparison of the mobility of TALE-MSR-mClover molecules within chromocenters (which we interpret as a readout of the physical properties of chromocenters) in ESCs treated with either MSR-gapmers or GFP-gapmers. To our knowledge, the TALE-MSR-mClover inducible system is currently the most appropriate to probe the dynamic behaviour of specific genomic loci. The strong binding of this reporter to MSR DNA has previously been validated (Miyanari et al., 2013). Additionally, we would like to note that we have validated the legitimacy of TALE-MSR-mClover as a chromocenter reporter by also assessing chromocenter dynamics with a tagged intrinsic component of heterochromatin, the heterochromatin protein 1 α (HP1 α) (Fig. S2F-H). These data confirmed that the mobility of molecules within the heterochromatic environment at chromocenters is reduced following MSR transcript reduction by LNA gapmer treatment, as compared to control conditions.

Also, we would like to clarify that we do not use any of our data as readouts of local viscosity. As the reviewer mentions, effects on viscosity may arise from different types of interactions that could be strongly influenced by the structure of the RNA itself (eg. how compact it is).

In light of these suggestions, we have updated the manuscript to tone down the interpretation of the material properties of the chromocenters in our experiments.

4.8: Why is the FRAP recovery of the TALE construct in J1 ESCs (ref 40) much slower than that measured here?

The TALE construct used in reference 40 (Thanisch, K, et al., 2014) is larger and targets a different major satellite repeat sequence than the TALE used in our current study (first reported by Miyanari et al., 2013). This may lead to differences in the binding ability of these distinct TALEs. For example, Thanisch reported that their TALE was depleted from highly condensed chromatin during mitosis, whereas the TALE protein used by Miyanari remains bound to mitotic chromatin (Miyanari et al., 2013; and confirmed by us). Additionally, TALE molecules are expressed by different vector systems, which might contribute to different availability of TALE-GFP molecules to bind the photo-bleached chromocenters and consequently impact the FRAP results.

4.9: The FRAP curves in Fig. 2B look pretty similar to each other. Although the difference might be significant, it would be interesting to know what functional relevance the authors assign to this small difference.

The recovery of fluorescence after photobleaching individual chromocenters shows a similar pattern in cells treated with either GFP gapmers or MSR gapmers. However, following MSR gapmer treatment, the fluorescence recovery time was doubled at photo-bleached chromocenters, as compared to controls (Fig. 2C), indicating that the movement of TALE-MSR-mClover molecules within chromocenters was significantly reduced. Also, the proportion of mobile TALE-MSR-mClover signal decreased slightly in the MSR RNA depleted cells (Figure 2D), suggesting that the ratio of dynamic (mobile) and stable (immobile) components within chromocenters is also affected by MSR RNA levels. Thus, we interpret this reduction in recovery time and proportion of mobile TALE-MSR-mClover molecules consistent with the TALE proteins being bound within a less dynamic chromatin.

4.10: Figure S3 shows that droplets become smaller with RNAs that have more repeats. What does this mean, and what is the size of these RNAs in the cell? The size seems to be regulated (e.g., PMID: 17984319), and it might be useful to discuss this aspect.

The reviewer raises an important point. However, we would like to point out that the droplet size does not uniformly decrease with increased repeats. The droplets formed with the 2-repeat RNAs are larger than those formed with the 1- and 8-repeat RNAs. Droplet size is thought to be determined by a combination of many different variables arising from the kinds and types of interactions made within droplets, including structural factors of the RNA itself (e.g. how compact the RNA is within a droplet). We therefore speculate that the 2-repeat RNA may promote a structure that allows form larger droplets (see also our response to point # 1.2).

While the droplet sizes do not uniformly track with repeat length, the critical concentration – the lowest concentration at which phase-separation occurs – does get lower with increased RNA length. A similar effect can be observed when DNA length is increased (Keenen et al., 2021). In this context, as the Reviewer points out from Lu and Gilbert's 2007

JBC paper, changing intra-droplet interactions based on RNA length can potentially provide an opportunity to modulate any *in vivo* phase separation. The timing of transcript production related to cell cycle is suggestive and may have additional mechanistic implications.

4.11: In Figure 3B/C, the number of analyzed cells seems much lower than in Figure 3D (based on the points shown in Figure 3C, it would be good to have the number in the legend). Could the large dataset used in Figure 3D to quantify HP1 fluorescence be used to analyze the colocalization that is assessed Figure 3B/C? It would be useful to have better statistics for the colocalization analysis.

In Figures 3B and C, we aimed to acquire higher spatial resolution to examine how the colocalization of H3K9me3 and HP1 α (observed in Figure 3D) occurs at the chromocenters. We have performed super resolution microscopy (3D-SIM) to better understand the structural and organizational pattern of both H3K9me3 and HP1a. As suggested, we have added the numbers and further information to the figure legend and methods section.

4.12: How many chromocenters were used for the analysis in Figure 4A? Also, I wonder why the number of analyzed chromocenters varies so much between panel B (100/105 chromocenters) and panel C (2849/3209 chromocenters).

Figures 4A-C and Figure 4D correspond to different experiments, which explains the difference in the number of chromocenters.

References

- Banáth, J. P., C. A. Bañuelos, D. Klokov, S. M. MacPhail, P. M. Lansdorp, and P. L. Olive. 2009. "Explanation for Excessive DNA Single-Strand Breaks and Endogenous Repair Foci in Pluripotent Mouse Embryonic Stem Cells." *Experimental Cell Research* 315 (8): 1505–20.
- Bulut-Karslioglu, Aydan, Inti A. De La Rosa-Velázquez, Fidel Ramirez, Maxim Barenboim, Megumi Onishi-Seebacher, Julia Arand, Carmen Galán, et al. 2014. "Suv39h-Dependent H3K9me3 Marks Intact Retrotransposons and Silences LINE Elements in Mouse Embryonic Stem Cells." *Molecular Cell* 55 (2): 277–90.
- Camacho, Oscar Velazquez, Carmen Galan, Kalina Swist-Rosowska, Reagan Ching, Michael Gamalinda, Fethullah Karabiber, Inti De La Rosa-Velazquez, et al. 2017. "Major Satellite Repeat RNA Stabilize Heterochromatin Retention of Suv39h Enzymes by RNA-Nucleosome Association and RNA:DNA Hybrid Formation." *eLife* 6:e25293.
- Casanova, Miguel, Michał Pasternak, Fatima El Marjou, Patricia Le Baccon, Aline V. Probst, and Geneviève Almouzni. 2013. "Heterochromatin Reorganization during Early Mouse Development Requires a Single-Stranded Noncoding Transcript." *Cell Reports* 4 (6): 1156–67.
- Efroni, Sol, Radharani Duttgupta, Jill Cheng, Hesam Dehghani, Daniel J. Hoepfner, Chandravanu Dash, David P. Bazett-Jones, et al. 2008. "Global Transcription in Pluripotent Embryonic Stem Cells." *Cell Stem Cell* 2 (5): 437–47.
- Goodarzi, Aaron A., Angela T. Noon, Dorothee Deckbar, Yael Ziv, Yosef Shiloh, Markus Löbrich, and Penny A. Jeggo. 2008. "ATM Signaling Facilitates Repair of DNA Double-Strand Breaks Associated with Heterochromatin." *Molecular Cell* 31 (2): 167–77.
- Hockings, Colin, Chetan Poudel, Kevin A. Feeney, Clara L. Novo, Mehdi S. Hamouda, Ioanna Mela, David Fernandez-Antoran, et al. 2020. "Illuminating Chromatin Compaction in Live Cells and Fixed Tissues Using SiR-DNA Fluorescence Lifetime." bioRxiv. <https://doi.org/10.1101/2020.05.02.073536>.
- Ishiuchi, Takashi, Rocio Enriquez-Gasca, Eiji Mizutani, Ana Bošković, Celine Ziegler-Birling, Diego Rodriguez-Terrones, Teruhiko Wakayama, Juan M. Vaquerizas, and Maria-Elena Torres-Padilla. 2015. "Early Embryonic-like Cells Are Induced by Downregulating Replication-Dependent Chromatin Assembly." *Nature Structural & Molecular Biology* 22 (9): 662–71.
- Keenen, Madeline M., David Brown, Lucy D. Brennan, Roman Renger, Harrison Khoo, Christopher R. Carlson, Bo Huang, Stephan W. Grill, Geeta J. Narlikar, and Sy Redding. 2021. "HP1 Proteins Compact DNA into Mechanically and Positionally Stable Phase Separated Domains," *eLife* 10:e64563.
- Larson, Adam G., Daniel Elnatan, Madeline M. Keenen, Michael J. Trnka, Jonathan B. Johnston, Alma L. Burlingame, David A. Agard, Sy Redding, and Geeta J. Narlikar. 2017. "Liquid Droplet Formation by HP1 α Suggests a Role for Phase Separation in Heterochromatin." *Nature* 547 (7662): 236–40.
- Lai, Fan, Sagar S. Damle, Karen K. Ling, Frank Rigo. 2020. "Directed RNase H Cleavage of Nascent Transcripts Causes Transcription Termination". *Molecular Cell* 77 (5): 1032-1043.
- Lee, Jong-Sun and Joshua T. Mendell. 2020. "Antisense-Mediated Transcript Knockdown Triggers Premature Transcription Termination". *Molecular Cell* 77 (5): 1044-1054.

Lu, Brett Y, Peter C. R. Emtage, Brenda J. Duyf, Aurthur J. Hilliker, and Joel C. Eissenberg. 2000. "Heterochromatin Protein 1 is Required for the Normal Expression of Two Heterochromatin Genes in *Drosophila*." *Genetics* 155 (2): 699-708.

Lu, Junjie, and David M. Gilbert. 2007. "Proliferation-Dependent and Cell Cycle Regulated Transcription of Mouse Pericentric Heterochromatin." *The Journal of Cell Biology* 179 (3): 411-421.

Maison, Christèle, Delphine Bailly, Antoine H. F. Peters, Jean-Pierre Quivy, Danièle Roche, Angela Taddei, Monika Lachner, Thomas Jenuwein, and Geneviève Almouzni. 2002. "Higher-Order Structure in Pericentric Heterochromatin Involves a Distinct Pattern of Histone Modification and an RNA Component." *Nature Genetics* 30 (3): 329-34.

Maison, Christèle, Delphine Bailly, Danièle Roche, Rocio Montes de Oca, Aline V. Probst, Isabelle Vassias, Florent Dingli, et al. 2011. "SUMOylation Promotes de Novo Targeting of HP1 α to Pericentric Heterochromatin." *Nature Genetics* 43 (3): 220-27.

Martens, Joost H. A., Roderick J. O'Sullivan, Ulrich Braunschweig, Susanne Opravil, Martin Radolf, Peter Steinlein, and Thomas Jenuwein. 2005. "The Profile of Repeat-Associated Histone Lysine Methylation States in the Mouse Epigenome." *The EMBO Journal* 24 (4): 800-812.

Meshorer, Eran, Dhananjay Yellajoshula, Eric George, Peter J. Scambler, David T. Brown, and Tom Misteli. 2012. "Hyperdynamic Plasticity of Chromatin Proteins in Pluripotent Embryonic Stem Cells." *Developmental Cell* 10 (1): 105-16.

Miyahari, Yusuke, Céline Ziegler-Birling, and Maria-Elena Torres-Padilla. 2013. "Live Visualization of Chromatin Dynamics with Fluorescent TALEs." *Nature Structural & Molecular Biology* 20 (11): 1321-24.

Muchardt, Christian, Marie Guilleme, Jacob-S Seeler, Didier Trouche, Anne Dejean, and Moshe Yaniv. 2002. "Coordinated Methyl and RNA Binding Is Required for Heterochromatin Localization of Mammalian HP1 α ." *EMBO Reports* 3 (10): 975-81.

Müller, Katharina P., Fabian Erdel, Maiwen Caudron-Herger, Caroline Marth, Barna D. Fodor, Mario Richter, Manuela Scaranaro, Joël Beaudouin, Malte Wachsmuth, and Karsten Rippe. 2009. "Multiscale Analysis of Dynamics and Interactions of Heterochromatin Protein 1 by Fluorescence Fluctuation Microscopy." *Biophysical Journal* 97 (11): 2876-85.

Noon, Angela T., Atsushi Shibata, Nicole Rief, Markus Löbrich, Grant S. Stewart, Penelope A. Jeggo, and Aaron A. Goodarzi. 2010. "53BP1-Dependent Robust Localized KAP-1 Phosphorylation Is Essential for Heterochromatic DNA Double-Strand Break Repair." *Nature Cell Biology* 12 (2): 177-84.

Novo, Clara Lopes, Calvin Tang, Kashif Ahmed, Ugljesa Djuric, Eden Fussner, Nicholas P. Mullin, Natasha P. Morgan, et al. 2016. "The Pluripotency Factor Nanog Regulates Pericentromeric Heterochromatin Organization in Mouse Embryonic Stem Cells." *Genes & Development* 30 (9): 1101-15.

Percharde, Michelle, Aydan Bulut-Karslioglu, and Miguel Ramalho-Santos. 2017. "Hypertranscription in Development, Stem Cells, and Regeneration." *Developmental Cell* 40 (1): 9-21.

Probst, Aline V., Ikuhiro Okamoto, Miguel Casanova, Fatima El Marjou, Patricia Le Baccon, and Geneviève Almouzni. 2010. "A Strand-Specific Burst in Transcription of Pericentric Satellites Is

Required for Chromocenter Formation and Early Mouse Development.” *Developmental Cell* 19 (4): 625–38.

Roach, Ruby J., Miguel Garavís, Carlos González, Geoffrey B. Jameson, Vyacheslav V. Filichev, and Tracy K. Hale. 2020. “Heterochromatin Protein 1 α Interacts with Parallel RNA and DNA G-Quadruplexes.” *Nucleic Acids Research* 48 (2): 682–93.

Schep, Ruben, Eva K. Brinkman, Christ Leemans, Xabier Vergara, Robin H. van der Weide, Ben Morris, Tom van Schaik, et al. 2021. “Impact of Chromatin Context on Cas9-Induced DNA Double-Strand Break Repair Pathway Balance.” *Molecular Cell* 81 (10): 2216–30.e10.

Schuster-Böckler, Benjamin, and Ben Lehner. 2012. “Chromatin Organization Is a Major Influence on Regional Mutation Rates in Human Cancer Cells.” *Nature* 488 (7412): 504–7.

Thanisch, Katharina, Katrin Schneider, Robert Morbitzer, Irina Solovei, Thomas Lahaye, Sebastian Bultmann, and Heinrich Leonhardt. 2014. “Targeting and Tracing of Specific DNA Sequences with dTALEs in Living Cells.” *Nucleic Acids Research* 42 (6): e38.

Tosolini, Matteo, Vincent Brochard, Pierre Adenot, Martine Chebrou, Giacomo Grillo, Violette Navia, Nathalie Beaujean, Claire Francastel, Amélie Bonnet-Garnier, and Alice Jouneau. 2018. “Contrasting Epigenetic States of Heterochromatin in the Different Types of Mouse Pluripotent Stem Cells.” *Scientific Reports* 8 (1): 5776.

Yin, Ping, Dong Deng, Chuangye Yan, Xiaojing Pan, Jianzhong Jeff Xi, Nieng Yan, and Yigong Shi. 2012. “Specific DNA-RNA Hybrid Recognition by TAL Effectors.” *Cell Reports* 2 (4): 707–13.

REVIEWER COMMENTS

Reviewer #1 (Remarks to the Author):

This revised manuscript is much improved. It should be suitable for publication in Nature Communications.

Reviewer #2 (Remarks to the Author):

Please, find my answers to the authors in the attached PDF

2.2: Secondly, if the author believe in the role of MSRtr in organization of chromocenters, they need to demonstrate presence and distribution of the RNA in chromocenters by RNAFISH. This is especially important in the view of the authors' suggestion that MSRtr compete with MSR for HP1 α binding. If their suggestion is correct, then the MSRtr signals must be really abundant and distributed through the entire chromocenter volume in untreated ESC, while this signal will be significantly reduced after MSRtr depletion. In addition, it is important to carry control RNA hybridizations after RNasing.

We agree with the reviewer that the localization of MSR RNA in chromocenters is an important aspect of our model. Multiple studies, including in mouse ESCs, have established using RNA-FISH that the vast majority of MSR transcripts remain associated with pericentromeric regions of mouse chromosomes and that the transcripts are distributed through the entire chromocenter volume (Martens et al., 2005; Lu and Gilbert, 2007; Maison et al., 2011; Bulut-Karslioglu et al., 2014; Ishiuchi et al., 2015; Camacho et al., 2017). The RNA-FISH signal for MSR transcript disappears following RNase treatment (Maison et al., 2011). Additionally, MSR transcripts are essential for chromocenter formation (Probst et al., 2010; Casanova et al., 2013) and MSR RNA has a higher affinity compared to MSR DNA for HP1 α binding (Muchardt et al., 2002; Maison et al., 2011). Importantly, incubating cells with RNase A for 30 minutes prior to fixation results in the loss of HP1 α signal, suggesting that the complete degradation of MSR RNA leads to HP1 α dislodging from chromocenters (Maison et al., 2002). Our model builds on these previous findings to propose that MSR RNA levels are important in modulating the substrate for HP1 α interaction at chromocenters in mouse ESCs, whereby high MSR transcript levels decrease HP1 α binding to chromatin (HP1 α remains at chromocenter foci) and help to maintain an uncompacted local structure.

I am surprise with this answer by the authors. The statement that MSR transcripts are distributed through the entire chromocenter volume in mouse ESCs is absolutely crucial for the authors' conclusions and proposed model. Nevertheless, they rely on data concerning nuclear distribution of MSR transcripts published previously and refer to 6 papers. Since I know some of these papers but could not recall the results the authors refer to, I have specifically checked them again for such important for this MS information:

- in **Martens et al 2005 and Bulut-Karslioglu et al., 2014** – there is not a single microscopic image showing distribution of MSR transcripts or any other RNA-FISH.
- in **Camacho et al 2017** – there is no RNA-FISH experiments but immunofluorescence
- in **Ishiuchi et al., 2015** – the only image with RNA-FISH (see below) shows several small foci that even not quite colocalize with chromocenters:

- in **Lu & Gilbert 2007** - RNA-FISH also clearly shows that MSR transcript signals are detected as small foci on surface of chromocenters and definitely not distributed through them:

- in **Maison et al., 2011** - RNA-FISH also shows reverse and forward MSR transcript signals on surface of chromocenters as small foci or even without connection to chromocenters:

Therefore, unless the authors show distribution of the transcripts through the entire volume of the ESC chromocenters - which is not difficult, since the authors have the probe - they cannot claim that the satellite RNA scaffolds heterochromatin organization in embryonic stem cells and the proposed model has no factual background:

=====
2.1: Firstly, when the authors describe chromocenter structure or their number per nucleus, their assessment is based on DAPI staining and thus can miss small chromocenters (e.g., Fig.4) or immunostaining that stains not only chromocenters (e.g., Fig.3). The methods of choice in such assays are those which specifically detect subcentromeric satellite – either DNA-FISH with MSR probe or targeting MSR with TALE-GFP (see also minor comment 9)

In mouse cells, chromocenters can be easily identified by their DAPI-rich staining that clusters into large nuclear foci, as shown in the manuscript by co-localization with MSR-TALE-mClover (Figures 1A, S1A and S2D) and by DNA-FISH (shown in Reviewer Figure 3 below and new Figure S1C). Additionally, interphasic centromeric DNA forms separate, much smaller foci (Guenatri & Almouzni, 2004; Probst & Almouzni, 2008; please see Reviewer Figure 4 below). Importantly, chromocenters are quantitatively similar when examined using DAPI or TALE-MSR-mClover (please see Figures S1D-E), demonstrating there is strong concordance between the methods.

The authors are right about a possibility to identify mouse chromocenters using simple DAPI counterstain due to preferential intercalation of the dye between AT, in which major satellite repeat enriched. However, this method is not **very exact** because DAPI can highlight other structures as well. Even on the exemplified image from Probst & Almouzni, 2008, the arrow points to a structure, which has the same staining intensities as real chromocenters (arrowheads) but not a chromocenter:

Since number of chromocenters is an important read out of some experiments, I think a more careful scoring has to be done. The same concerns the next answer – see below.

=====
2.3: Supplementary Figure 1C. The histogram shows percentage of nuclei with 1, 2 and 3 chromocenters. As one can see from other figures in the paper, as a rule, ESCs, have more than 3 chromocenters.

We thank the reviewer for highlighting this. The data in the histogram is from analysis that was performed on single optical sections. We have changed the axis label and the figure legend to clarify this point.

In my first reading of the paper, I have not realized that the scoring was done on single sections. Why? This can be very misleading because chromocenters often gathered on either top, or bottom of a nucleus with different frequency, or around a nucleolus. Since the

authors collected image stacks using Nikon SIM microscope and spinning disk confocal microscope, they have stacks in which chromocenters can be scored.

=====
2.4: Supplementary Figure 1D. How the authors define a diameter of a chromocenter? Even on projections, chromocenters have very irregular shape. Besides, to define a border of a chromocenter, one needs to threshold an image. How thresholds were set up for different slides with different acquisition conditions? It is not clear from the legend, e.g. how many nuclei were assessed in the three independent experiments - 3 x 16 or 16 in general? In any case, this statistic is pretty poor.

DAPI linescan analyses were performed in single optical sections where DAPI foci were at the mid-focal point. Fluorescence intensity histograms were generated with a linescan across the nucleus. The background signal (outside of the nucleus) was subtracted from the baseline fluorescence within the nucleus and from the signal within the chromocenter. Chromocenter borders were then defined at the base of each fluorescence peak above the nuclear background. In total, 16 and 25 nuclei were analysed from three independent inductions of TALE-MSR-mClover.

I still do not understand how the authors measured chromocenter diameter. On the example below taking from their Suppl.Fig.1, I indicated that the diameter can be measured pretty arbitrary.

As a more correct estimation, I would suggest to measure an area of a mid section through a chromocenter. Of course, all chromocenters within one nucleus have to be taken in account. And I have to repeat it again: knowing how variable shape of mouse ESCs and how variable number of chromocenters in WT ESCs, 16 and 25 nuclei altogether from 3 experiments are not enough for a proper statistics.

=====

2.8: Page 9 (Figure 3B). To compare overlapping signals of H3K9me3 chromatin and chromatin binding HP1 α , the authors have chosen to compare volumes of the signal. This is not an appropriate method, since volume rendering strongly depends on a threshold, which in this case has to be set separately for two different channels, individual cells and different acquisitions. The colocalization has to be shown and estimated on single optical sections through mid of chromocenters.

HP1 α is a reader of H3K9me3 and thus is highly co-localised when imaged by confocal microscopy, as can be seen in Figure 3D. In Figure 3B, we used the enhanced resolution of structured illumination microscopy (SIM) to have a better understanding of the distribution of each heterochromatin marker within chromocenters. We have clarified in the Methods section that we acquired a series of raw images with 120 nm step intervals that were then reconstructed into the high resolution image shown using a 3D-SIM algorithm. The Nikon 3D-SIM reconstruction algorithm used has minimal parameters and does not require a set threshold to define the limit of the signal and is therefore not subject to the potential bias that the referee suggests. At this higher resolution, we opted for measuring the overlap of H3K9me3 and HP1 α by rendering volumes for each channel with IMARIS software (with the same threshold in each channel) and quantifying how much volume from one marker falls within the volume of the other marker and vice-versa.

I do not know how to interpret this answer. It is a text-book knowledge that 3D reconstruction by any existing software is based on an **image segmentation** and an image segmentation requires a **thresholding**. Therefore, I am confused with the answer about using 3D-SIM algorithm *not requiring* a thresholding and then IMARIS software *requiring* one.

In any case, I can only repeat my comment that 3D volume reconstruction of the two immunostaining signals with not clear cut borders (see Fig.3b) is the most inappropriate method to estimate colocalization. The most straightforward way is estimation of colocalization on single mid sections (see e.g., Ronneberger et al, 2008; 10.1007/s10577-008-1236-4).

=====

2.9: Figure 3D. This figure does not convince that brightness of the fluorescent signal is different in two conditions. Authors have to explain (a) how the HP1 α fluorescence was measured - on projections or single sections, for individual cells or group of cells - and (b) how can they be sure about equal conditions of staining and image acquisition from two different slides, which is essential for such comparison.

We have updated the Methods section to clarify that the image acquisition conditions were maintained between slides (each slide was acquired from independent replicate experiments), so that the technical variation between replicates could be accounted for, including across replicates. The fluorescence intensity was measured on single sections in groups of cells, in chromocenters firstly defined by DAPI (most intense foci).

I still do not grasp why intensities of immunostaining was measured in groups of ca. 30 cells and in one single plain. Such important for the conclusions experiment has to be based on measurements of a sufficient number of individual cells and, probably, on projections.

=====
2.10: Figure 4A. Firstly, how many nuclei were analyzed in this experiment? Secondly, the histogram shows number of chromocenters from 1 to 6. However, the very image on the left shows 8 chromocenters. I am working with various mouse cells types half of my scientific career but I never saw mouse somatic cells with just one chromocenter. I think the authors are mistaking because they rely only on DAPI staining that might be misleading, especially in case of ESCs. To make a correct assessment of chromocenter number, a DNA-FISH with a probe for MSR, specifically detecting subcentromeric heterochromatin, has to be employed.

A total of 55 and 36 nuclei (100 and 105 chromocenters, respectively) were quantified following the transfection of ESCs with GFP or MSR gapmers. The reviewer correctly points out that the x-axis was wrongly labelled (thank you), as it refers to the number of chromocenters per single focal plane rather than per nucleus. We have corrected this in the updated figure.

As a more general note, a previous study showed that the number of chromocenters per nucleus of mouse ESCs ranges between one and nine with a median number of four (Meshorer et al., 2016), which is very similar to the numbers that we report in our study. Also, as described in response to comment 2.1, we show that the number of pericentromeric foci are highly concordant when comparing data obtained from DAPI labelling, MSR DNA-FISH and MSR-TALE-mClover (new Figure 4A and Figure S1A, respectively). We agree with the reviewer that the low numbers of chromocenters are not observed in most mouse somatic cells, but rather it is an interesting feature that is characteristic of mouse pluripotent cells. As described in our manuscript, we believe that the higher levels of MSR transcripts in ESCs compared to somatic cells helps to modulate the properties of pericentromeric heterochromatin and chromocenter organization.

I have already commented about estimation of chromocenter numbers on a single optical section (2.3). As for the mouse cells with a single chromocenter, there is only one well-proven example in the literature - rod cells in mouse retina (10.1016/j.cell.2009.01.052). In these cells the major satellite repeat **of all chromosomes** merge in one blob. Do the authors claim the same case for mouse ESCs? That would be a discovery in itself. It would be interesting to see such examples of ESCs after FISH with the satellite probe.

I also would disagree that stem cells have low number of chromocenters compared to differentiated cells. There are examples of terminally differentiated cells (e.g., neurons) with only several chromocenters – the number of chromocenters strongly depends on shape and size of cells. As for Meshorer et al., 2006, which I looked up, unfortunately, they do not describe how number of chromocenters was estimated – in an entire cell or also on an optical section.

=====
And as the last general comment concerning microscopy images:

I would like to advise the authors to follow a good publishing practice and always present images as grey-scale keeping RGB only for overlays (as it is actually done for Fig.5C). Then readers do not have to copy images from a manuscript into Photoshop to consider single channels.

Reviewer #3 (Remarks to the Author):

Lopes Novo et al have significantly improved their manuscript since initial submission. The authors have added insightful experiments including additional in vitro and in vivo measurements to support their original manuscript. Additionally, they have satisfactorily amended the manuscript text and figure legends for clarity and further interpretation. The SiR-DNA fluorescence lifetime experiments to measure compaction are elegant and well-executed, I look forward to seeing similar follow-up work. In vitro, I'm also looking forward to further understanding of dsRNA polyA/U promoting HP1 phase separation, and the potential re-entrant behavior mentioned in the reviewer response (1 vs 2 vs 8 MSR repeat lengths).

I now enthusiastically recommend the manuscript for publication in Nature Communications.

Reviewer #4 (Remarks to the Author):

The revised manuscript by Novo et al presents almost the same data as the initially submitted version, and my concerns remain therefore the same. In a nutshell, the authors characterize different properties of chromocenters in mouse embryonic stem cells that have been transfected with major satellite repeat (MSR) gapmers or not, and they find moderate differences. Chromocenters in untreated cells show 1.5-times more cleavage/coalescence events than those in MSR gapmer-transfected cells, HP1a and a TALE-MSR-mClover fusion dissociate a few seconds faster from chromocenters in untreated cells, the HP1a and H3K9me3 intensities at chromocenters are slightly lower (about 1.1-times and 1.4-times) in untreated cells, and chromocenters are larger in untreated cells. Furthermore, the authors show that HP1 phase-separates when mixed with high concentrations of MSR RNA, which is a potentially interesting result but which is a scenario that is different from the cell where HP1 localizes to chromocenters that contain a large amount of MSR DNA and probably a much lower amount of MSR RNA (estimated based on published data, as the authors unfortunately did not quantify the MSR RNA content, which would be needed to interpret the in vitro assays and to better evaluate the suggested model).

Unfortunately, based on the manuscript in its current form, we still do not know if any of the rather small changes in the properties of chromocenters with/without MSR gapmers, which seem to be in many cases similar to the standard deviation observed in a single condition, have any impact on function. They neither bring us much closer to understanding phase separation of heterochromatin: for example, it is unclear if FRAP recoveries become slower because proteins bind stronger to their binding sites (due to a changed local chromatin state after MSR gapmer transfection) or because a liquid condensate is formed/alterd. Furthermore, the authors do not link the hallmarks of genomic instability that they observe after MSR gapmer transfection to the changes of chromocenters they describe. Therefore, despite the potentially interesting results, I am not convinced that this manuscript is very well suitable for publication in a multi-disciplinary journal.

Comments about individual (major) points:

4.1 I appreciate that the authors have quantified these data.

4.2 The result that TALE-MSR and HP1 dissociate more slowly from chromocenters after MSR gapmer transfection seems to be perfectly consistent with a model in which binding of these proteins to their binding sites is affected by the local chromatin state (histone marks, proteins bound to chromatin, chromatin compaction/accessibility), which changes upon MSR gapmer transfection. This change is documented in Figs. 3/4, showing that H3K9me3/HP1 signals increase when MSR gapmers are transfected, and this change could occur independently from affecting phase separation. RNA

Polymerase, which I mentioned in the first revision round, was only one example to refer to an altered local chromatin state. I think the authors should also consider such an "altered binding" model to explain their FRAP data.

As one difference between an "altered binding" model and a "phase separation" model would be that the first is effectively 1-dimensional while the second is effectively 3-dimensional (the condensate produced by phase separation is a 3-dimensional object), studying an independent sequence that localizes within this putative 3-dimensional object could be helpful. As the authors can see in their image (Reviewer Figure 6), some chromocenters overlap with MinSat sequences, making them a potentially useful control even if they do not always overlap with chromocenters. In a phase separation model, the population of MinSat sequences in chromocenters should be affected by changes in the 3D "physical environment", in a binding model they should not.

4.3 It is unclear to me how the authors can be sure that MSR transcripts operate in cis, given that they have observed in a previous paper that their overexpression has an effect, and given that the authors do not visualize MSR transcripts in the cell, for example via RNA FISH as suggested by reviewer #1. This experiment could also provide useful information about the abundance of MSR transcripts, which the authors decided not to quantify.

4.4 I think that it is impossible to estimate the RNA concentration from the DNA concentration, as the authors imply in their response. The RNA concentration will depend on the transcription rate, the half-life of the transcripts etc. I think it would be very important to measure the MSR RNA concentration in the cell to interpret the in vitro experiments. Furthermore, it would be closer to the cellular situation to study HP1 + DNA/nucleosomes + RNA in vitro (using the conditions in Keenen et al. 2021 + RNA), as chromocenters contain a high concentration of DNA/nucleosomes that affect phase separation.

REVIEWER COMMENTS

Reviewer #1

This revised manuscript is much improved. It should be suitable for publication in Nature Communications.

Reviewer #3

Lopes Novo et al have significantly improved their manuscript since initial submission. The authors have added insightful experiments including additional in vitro and in vivo measurements to support their original manuscript. Additionally, they have satisfactorily amended the manuscript text and figure legends for clarity and further interpretation. The SiR-DNA fluorescence lifetime experiments to measure compaction are elegant and well-executed, I look forward to seeing similar follow-up work. In vitro, I'm also looking forward to further understanding of dsRNA polyA/U promoting HP1 phase separation, and the potential re-entrant behavior mentioned in the reviewer response (1 vs 2 vs 8 MSR repeat lengths).

I now enthusiastically recommend the manuscript for publication in Nature Communications.

We thank the two reviewers for their supportive and encouraging comments.

Reviewer #2

2.2

Comment from the first round of review:

Secondly, if the author believe in the role of MSRtr in organization of chromocenters, they need to demonstrate presence and distribution of the RNA in chromocenters by RNAFISH. This is especially important in the view of the authors' suggestion that MSRtr compete with MSR for HP1 α binding. If their suggestion is correct, then the MSRtr signals must be really abundant and distributed through the entire chromocenter volume in untreated ESC, while this signal will be significantly reduced after MSRtr depletion. In addition, it is important to carry control RNA hybridizations after RNasing.

Response from the first round of review:

We agree with the reviewer that the localization of MSR RNA in chromocenters is an important aspect of our model. Multiple studies, including in mouse ESCs, have established using RNA-FISH that the vast majority of MSR transcripts remain associated with pericentromeric regions of mouse chromosomes and that the transcripts are distributed through the entire chromocenter volume (Martens et al., 2005; Lu and Gilbert, 2007; Maison et al., 2011; Bulut-Karslioglu et al., 2014; Ishiuchi et al., 2015; Camacho et al., 2017). The RNA-FISH signal for MSR transcript disappears following RNase treatment (Maison et al., 2011). Additionally, MSR transcripts are essential for chromocenter formation (Probst et al., 2010; Casanova et al., 2013) and MSR RNA has a higher affinity compared to MSR DNA for HP1 α binding (Muchardt et al., 2002; Maison et al., 2011). Importantly, incubating cells with RNase A for 30 minutes prior to fixation results in the loss of HP1 α signal, suggesting that the complete degradation of MSR RNA leads to HP1 α dislodging from chromocenters (Maison et al., 2002). Our model builds on these previous findings to propose that MSR RNA levels are important in modulating the substrate for HP1 α interaction at chromocenters in mouse ESCs, whereby high MSR transcript levels decrease HP1 α binding to chromatin (HP1 α remains at chromocenter foci) and help to maintain an uncompact local structure.

Comment from the second round of review:

I am surprised with this answer by the authors. The statement that MSR transcripts are distributed through the entire chromocenter volume in mouse ESCs is absolutely crucial for the authors' conclusions and proposed model. Nevertheless, they rely on data concerning nuclear distribution of MSR transcripts published previously and refer to 6 papers. Since I know some of these papers but could not recall the results the authors refer to, I have specifically checked them again for such important information:

- in Martens et al 2005 and Bulut-Karslioglu et al., 2014 – there is not a single microscopic image showing distribution of MSR transcripts or any other RNA-FISH.
- in Camacho et al 2017 – there is no RNA-FISH experiments but immunofluorescence
- in Ishiuchi et al., 2015 – the only image with RNA-FISH (see below) shows several small foci that even not quite colocalize with chromocenters
- in Lu & Gilbert 2007 - RNA-FISH also clearly shows that MSR transcript signals are detected as small foci on surface of chromocenters and definitely not distributed through them
- in Maison et al., 2011 - RNA-FISH also shows reverse and forward MSR transcript signals on surface of chromocenters as small foci or even without connection to chromocenters

Therefore, unless the authors show distribution of the transcripts through the entire volume of the ESC chromocenters - which is not difficult, since the authors have the probe – they cannot claim that the satellite RNA scaffolds heterochromatin organization in embryonic stem cells and the proposed model has no factual background.

Response to the second round of review:

We would like to apologise for incorrectly stating that the papers referred to in our previous reply showed major satellite repeat (MSR) RNA throughout the entire chromocenter volume. As the reviewer correctly points out, upon closer examination of these and other papers there is no published data showing RNA-FISH signal for MSR RNA through the entire volume of chromocenters. Our answer was based in part on the location of satellite repeats within chromocenters and the assumption that transcripts are retained in close proximity to their sites of transcription. So to directly address the reviewer's comment we have now performed RNA-FISH as suggested. This has enabled us to investigate the presence and distribution of MSR transcripts in ESCs (Figure 4F and Supplementary Figure 4E). These experiments showed that MSR transcript foci are highly variable in size across different cells, which we speculate could be a reflection of cell-cycle effects and perhaps associated functions (*cis* vs *trans*?). Notably, strong MSR RNA-FISH signal was rarely observed within chromocenters. Instead, the most common pattern was the presence of large foci on the periphery of chromocenters, and in some cells these foci seem to border multiple chromocenters (Supplementary Figure 4E). Importantly, we show that the RNA-FISH signal is sensitive to RNaseA and RNaseH treatments, and that the signal is strongly reduced following MSR-gapmer treatment (Figure 4F). These interesting new data (which are corroborated by RNA-FISH images recently published by the Guttman lab - PMID 34739832 and the Bonnet-Garnier lab - PMID: 35048992) have allowed us to further develop and strengthen our model (new Figure 6). In particular, we appreciate that the current evidence does not adequately support our previous suggestion that MSR transcripts compete with MSR sequences for HP1 α binding, and therefore we have removed this part from our model and conclusions. The new data raise the possibility that MSR transcripts might be involved in the coalescence of neighbouring chromocenters, which would be consistent with the observed decrease in chromocenter fusion events following MSR RNA depletion, but we have opted to be cautious and to not emphasise this point in the revised manuscript. We have re-written several sections of the discussion to take into account the new RNA-FISH data and associated implications.

2.1

Comment from the first round of review:

Firstly, when the authors describe chromocenter structure or their number per nucleus, their assessment is based on DAPI staining and thus can miss small chromocenters (e.g., Fig.4) or immunostaining that stains not only chromocenters (e.g., Fig.3). The methods of choice in such assays are those which specifically detect subcentromeric satellite – either DNA-FISH with MSR probe or targeting MSR with TALE-GFP (see also minor comment 9)

Response from the first round of review:

In mouse cells, chromocenters can be easily identified by their DAPI-rich staining that clusters into large nuclear foci, as shown in the manuscript by co-localization with TALE-MSR-mClover (Figures 1A, S1A and S2D) and by DNA-FISH (shown in Reviewer Figure 3 below and new Figure S1C). Additionally, interphasic centromeric DNA forms separate, much smaller foci (Guenatri & Almouzni, 2004; Probst & Almouzni, 2008; please see Reviewer Figure 4 below). Importantly, chromocenters are quantitatively similar when examined using DAPI or TALE-MSR-mClover (please see Figures S1D-E), demonstrating there is strong concordance between the methods.

Comment from the second round of review:

The authors are right about a possibility to identify mouse chromocenters using simple DAPI counterstain due to preferential intercalation of the dye between AT, in which major satellite repeat enriched. However, this method is not very exact because DAPI can highlight other structures as well. Even on the exemplified image from Probst & Almouzni, 2008, the arrow points to a structure, which has the same staining intensities as real chromocenters (arrowheads) but not a chromocenter. Since number of chromocenters is an important read out of some experiments, I think a more careful scoring has to be done. The same concerns the next answer – see below.

Response to the second round of review:

We have followed the reviewer's recommendation and have performed new experiments that demonstrate that scoring chromocenter number is highly consistent when measured using different methods. We have now directly compared and scored DAPI signals with major satellite DNA-FISH signals in the same images (Supplementary Figure 4B-C). The data show that the two methods give quantitatively very consistent results, and so we think it is unlikely that our measurements using DAPI counterstain are recording non-chromocenter structures. We have also visualised chromocenters using TALE-MSR-mClover, together with DAPI and HP1a immunofluorescence (examples shown in Supplementary Figure 4A). Together, these results provide firm evidence of the high concordance between methodologies, and provide sufficient confidence in the scoring of chromocenters.

2.3

Comment from the first round of review:

Supplementary Figure 1C. The histogram shows percentage of nuclei with 1, 2 and 3 chromocenters. As one can see from other figures in the paper, as a rule, ESCs, have more than 3 chromocenters.

Response from the first round of review:

We thank the reviewer for highlighting this. The data in the histogram is from analysis that was performed on single optical sections. We have changed the axis label and the figure legend to clarify this point.

Comment from the second round of review:

In my first reading of the paper, I have not realized that the scoring was done on single sections. Why? This can be very misleading because chromocenters often gathered on either top, or bottom of a nucleus with different frequency, or around a nucleolus. Since the authors collected image stacks

using Nikon SIM microscope and spinning disk confocal microscope, they have stacks in which chromocenters can be scored.

Response to the second round of review:

We have now scored chromocenter numbers in whole nuclei using both DAPI (Figure 4A) and DNA-FISH (Supplementary Figure 4C). We have moved the linescan analysis based on DAPI to Supplementary Figure 4D. Importantly, as before, there is a significant change in chromocenter numbers following MSR gapmer treatment, and this difference is detected when analysing chromocenters in either whole nuclei or in single sections.

2.4

Comment from the first round of review:

Supplementary Figure 1D. How the authors define a diameter of a chromocenter? Even on projections, chromocenters have very irregular shape. Besides, to define a border of a chromocenter, one needs to threshold an image. How thresholds were set up for different slides with different acquisition conditions? It is not clear from the legend, e.g. how many nuclei were assessed in the three independent experiments – 3 x 16 or 16 in general? In any case, this statistic is pretty poor.

Response from the first round of review:

DAPI linescan analyses were performed in single optical sections where DAPI foci were at the mid focal point. Fluorescence intensity histograms were generated with a linescan across the nucleus. The background signal (outside of the nucleus) was subtracted from the baseline fluorescence within the nucleus and from the signal within the chromocenter. Chromocenter borders were then defined at the base of each fluorescence peak above the nuclear background. In total, 16 and 25 nuclei were analysed from three independent inductions of TALE-MSR-mClover.

Comment from the second round of review:

I still do not understand how the authors measured chromocenter diameter. On the example below taking from their Suppl.Fig.1, I indicated that the diameter can be measured pretty arbitrary.

As a more correct estimation, I would suggest to measure an area of a mid section through a chromocenter. Of course, all chromocenters within one nucleus have to be taken in account. And I have to repeat it again: knowing how variable shape of mouse ESCs and how variable number of chromocenters in WT ESCs, 16 and 25 nuclei altogether from 3 experiments are not enough for a proper statistics.

Response to the second round of review:

We have now measured the area and the major axis length through the mid-section of chromocenters using the CellProfiler Measurement module. As requested, we have also increased the number of nuclei scored to 774 for cells treated with GFP gapmers and 1084 for MSR gapmers. The new data are presented in Figures 4B and C. Due to large numbers, we were advised by our statistician (Anne Segonds-Pichon) present the data as binned quartiles and compare the differences after MSR gapmer treatment to the GFP controls as each set of biological replicates (GFP_I vs MS_I, etc) were statistically different.

For one experiment, we were unable to increase the 'n' number for the data in Supplementary Figures 1, which show that chromocenter organisation is unaffected by the expression of TALE-MSR-GFP. However, this specific point (absence of affecting chromocenters) has been previously reported by the Torres-Padilla group (PMID 24096363) and in our earlier publication (PMID 27125671). Importantly, the Torres-Padilla group also generated mouse embryos that expressed the same TALE-MSR-GFP transgene and which developed normally, thereby providing

further evidence that the TALE-MSR-GFP is unlikely to disrupt pericentromeric heterochromatin. We now cite these studies in the results section of the manuscript.

2.8

Comment from the first round of review:

Page 9 (Figure 3B). To compare overlapping signals of H3K9me3 chromatin and chromatin binding HP1a, the authors have chosen to compare volumes of the signal. This is not an appropriate method, since volume rendering strongly depends on a threshold, which in this case has to be set separately for two different channels, individual cells and different acquisitions. The colocalization has to be shown and estimated on single optical sections through mid of chromocenters.

Response from the first round of review:

HP1 α is a reader of H3K9me3 and thus is highly co-localised when imaged by confocal microscopy, as can be seen in Figure 3D. In Figure 3B, we used the enhanced resolution of structured illumination microscopy (SIM) to have a better understanding of the distribution of each heterochromatin marker within chromocenters. We have clarified in the Methods section that we acquired a series of raw images with 120 nm step intervals that were then reconstructed into the high resolution image shown using a 3D-SIM algorithm. The Nikon 3D-SIM reconstruction algorithm used has minimal parameters and does not require a set threshold to define the limit of the signal and is therefore not subject to the potential bias that the referee suggests. At this higher resolution, we opted for measuring the overlap of H3K9me3 and HP1 α by rendering volumes for each channel with IMARIS software (with the same threshold in each channel) and quantifying how much volume from one marker falls within the volume of the other marker and vice-versa.

Comment from the second round of review:

I do not know how to interpret this answer. It is a text-book knowledge that 3D reconstruction by any existing software is based on an image segmentation and an image segmentation requires a thresholding. Therefore, I am confused with the answer about using 3D-SIM algorithm not requiring a thresholding and then IMARIS software requiring one. In any case, I can only repeat my comment that 3D volume reconstruction of the two immunostaining signals with not clear cut borders (see Fig.3b) is the most inappropriate method to estimate colocalization. The most straightforward way is estimation of colocalization on single mid sections (see e.g., Ronneberger et al, 2008; 10.1007/s10577-008-1236-4).

Response to the second round of review:

We have followed the reviewer's recommendation and have now measured the colocalisation of H3K9me3 and HP1a signals on single optical sections through the mid-sections of chromocenters. These new findings are shown in Figure 3C. Consistent with our previous results, the new data show that there is increased colocalisation of the two heterochromatin-associated signals following MSR gapmer treatment. We performed this analysis on confocal images in order to have a sufficient 'n' number. We have therefore removed the criticised and now unnecessary quantification of volume overlap from the 3D-SIM images, although we have retained the 3D-SIM images in the figure as they nicely exemplify the compaction of chromocenter heterochromatin and increased overlap between H3K9me3 and HP1a.

2.9

Comment from the first round of review:

Figure 3D. This figure does not convince that brightness of the fluorescent signal is different in two conditions. Authors have to explain (a) how the HP1a fluorescence was measured – on projections or

single sections, for individual cells or group of cells – and (b) how can they be sure about equal conditions of staining and image acquisition from two different slides, which is essential for such comparison.

Response from the first round of review:

We have updated the Methods section to clarify that the image acquisition conditions were maintained between slides (each slide was acquired from independent replicate experiments), so that the technical variation between replicates could be accounted for, including across replicates. The fluorescence intensity was measured on single sections in groups of cells, in chromocenters firstly defined by DAPI (most intense foci).

I still do not grasp why intensities of immunostaining was measured in groups of ca. 30 cells and in one single plain. Such important for the conclusions experiment has to be based on measurements of a sufficient number of individual cells and, probably, on projections.

Response to the second round of review:

Following the reviewer's comment, we have removed the comparison between HP1a fluorescence intensity in cells treated with GFP or MSR gapmers. The finding that there is an increased recruitment of HP1a to major satellite DNA following MSR gapmer treatment is better evidenced by the CHIP data shown in Figure 3D.

2.10

Comment from the first round of review:

Figure 4A. Firstly, how many nuclei were analyzed in this experiment? Secondly, the histogram shows number of chromocenters from 1 to 6. However, the very image on the left shows 8 chromocenters. I am working with various mouse cells types half of my scientific career but I never saw mouse somatic cells with just one chromocenter. I think the authors are mistaking because they rely only on DAPI staining that might be misleading, especially in case of ESCs. To make a correct assessment of chromocenter number, a DNA-FISH with a probe for MSR, specifically detecting subcentromeric heterochromatin, has to be employed.

Response from the first round of review:

A total of 55 and 36 nuclei (100 and 105 chromocenters, respectively) were quantified following the transfection of ESCs with GFP or MSR gapmers. The reviewer correctly points out that the x-axis was wrongly labelled (thank you), as it refers to the number of chromocenters per single focal plane rather than per nucleus. We have corrected this in the updated figure.

As a more general note, a previous study showed that the number of chromocenters per nucleus of mouse ESCs ranges between one and nine with a median number of four (Meshorer et al., 2016), which is very similar to the numbers that we report in our study. Also, as described in response to comment 2.1, we show that the number of pericentromeric foci are highly concordant when comparing data obtained from DAPI labelling, MSR DNA-FISH and MSR-TALE-mClover (new Figure 4A and Figure S1A, respectively). We agree with the reviewer that the low numbers of chromocenters are not observed in most mouse somatic cells, but rather it is an interesting feature that is characteristic of mouse pluripotent cells. As described in our manuscript, we believe that the higher levels of MSR transcripts in ESCs compared to somatic cells helps to modulate the properties of pericentromeric heterochromatin and chromocenter organization.

Comment from the second round of review:

I have already commented about estimation of chromocenter numbers on a single optical section (2.3). As for the mouse cells with a single chromocenter, there is only one wellproven example in the

literature - rod cells in mouse retina (10.1016/j.cell.2009.01.052). In these cells the major satellite repeat of all chromosomes merge in one blob. Do the authors claim the same case for mouse ESCs? That would be a discovery in itself. It would be interesting to see such examples of ESCs after FISH with the satellite probe. I also would disagree that stem cells have low number of chromocenters compared to differentiated cells. There are examples of terminally differentiated cells (e.g., neurons) with only several chromocenters – the number of chromocenters strongly depends on shape and size of cells. As for Meshorer et al., 2006, which I looked up, unfortunately, they do not describe how number of chromocenters was estimated – in an entire cell or also on an optical section.

Response to the second round of review:

We have now analysed chromocenter numbers in entire nuclei (Figure 4A). Additionally, the new data demonstrate there is a high agreement in the number of chromocenters measured by DAPI or by DNA-FISH (Supplementary Figure 4C). We believe there is now sufficient evidence to support the measured changes that we observe in chromocenter organisation upon GFP vs MSR gapmer treatment.

Additional comment

Comment from the second round of review:

And as the last general comment concerning microscopy images:

I would like to advise the authors to follow a good publishing practice and always present images as grey-scale keeping RGB only for overlays (as it is actually done for Fig.5C). Then readers do not have to copy images from a manuscript into Photoshop to consider single channels.

Response to the second round of review:

We have changed all non-merged panels to grayscale (Figures 1D, 3C, 3E, SF2E, SF2D, SF4A, SF4C).

Reviewer #4

The revised manuscript by Novo et al presents almost the same data as the initially submitted version, and my concerns remain therefore the same. In a nutshell, the authors characterize different properties of chromocenters in mouse embryonic stem cells that have been transfected with major satellite repeat (MSR) gapmers or not, and they find moderate differences. Chromocenters in untreated cells show 1.5-times more cleavage/coalescence events than those in MSR gapmer-transfected cells, HP1a and a TALE-MSR-mClover fusion dissociate a few seconds faster from chromocenters in untreated cells, the HP1a and H3K9me3 intensities at chromocenters are slightly lower (about 1.1-times and 1.4-times) in untreated cells, and chromocenters are larger in untreated cells. Furthermore, the authors show that HP1 phase-separates when mixed with high concentrations of MSR RNA, which is a potentially interesting result but which is a scenario that is different from the cell where HP1 localizes to chromocenters that contain a large amount of MSR DNA and probably a much lower amount of MSR RNA (estimated based on published data, as the authors unfortunately did not quantify the MSR RNA content, which would be needed to interpret the in vitro assays and to better evaluate the suggested model).

Unfortunately, based on the manuscript in its current form, we still do not know if any of the rather small changes in the properties of chromocenters with/without MSR gapmers, which seem to be in many cases similar to the standard deviation observed in a single condition, have any impact on function. They neither bring us much closer to understanding phase separation of heterochromatin: for example, it is unclear if FRAP recoveries become slower because proteins bind stronger to their

binding sites (due to a changed local chromatin state after MSR gapmer transfection) or because a liquid condensate is formed/altered. Furthermore, the authors do not link the hallmarks of genomic instability that they observe after MSR gapmer transfection to the changes of chromocenters they describe. Therefore, despite the potentially interesting results, I am not convinced that this manuscript is very well suitable for publication in a multi-disciplinary journal.

Comments about individual (major) points:

4.1 I appreciate that the authors have quantified these data.

Thank you.

4.2 The result that TALE-MSR and HP1 dissociate more slowly from chromocenters after MSR gapmer transfection seems to be perfectly consistent with a model in which binding of these proteins to their binding sites is affected by the local chromatin state (histone marks, proteins bound to chromatin, chromatin compaction/accessibility), which changes upon MSR gapmer transfection. This change is documented in Figs. 3/4, showing that H3K9me3/HP1 signals increase when MSR gapmers are transfected, and this change could occur independently from affecting phase separation. RNA Polymerase, which I mentioned in the first revision round, was only one example to refer to an altered local chromatin state. I think the authors should also consider such an “altered binding” model to explain their FRAP data.

As one difference between an “altered binding” model and a “phase separation” model would be that the first is effectively 1-dimensional while the second is effectively 3-dimensional (the condensate produced by phase separation is a 3-dimensional object), studying an independent sequence that localizes within this putative 3-dimensional object could be helpful. As the authors can see in their image (Reviewer Figure 6), some chromocenters overlap with MinSat sequences, making them a potentially useful control even if they do not always overlap with chromocenters. In a phase separation model, the population of MinSat sequences in chromocenters should be affected by changes in the 3D “physical environment”, in a binding model they should not.

We thank the reviewer for this comment. While we agree that chromatin changes alone could explain our FRAP data, our *in vitro* demonstration that MSR RNA promotes phase-separation of HP1a makes us still favour the ‘phase separation’ model. Nevertheless, we also agree that it would be helpful to present both models as providing plausible interpretations of our data. We have therefore now added the “altered binding” model to Figure 6, and have further emphasised this model in the discussion (page 11). These additions have helped to strengthen the manuscript and present a more balanced and interesting outlook. We also thank the reviewer for the suggestion of using MinSat sequences as controls. We have tried to do these experiments, however our attempts to use MinSat-FISH did not provide sufficiently reliable data to address this point.

4.3 It is unclear to me how the authors can be sure that MSR transcripts operate in *cis*, given that they have observed in a previous paper that their overexpression has an effect, and given that the authors do not visualize MSR transcripts in the cell, for example via RNA FISH as suggested by reviewer #1. This experiment could also provide useful information about the abundance of MSR transcripts, which the authors decided not to quantify.

In our previous work (PMID: 27125671), we induced the overexpression of MSR RNA in *cis* by recruiting a TALE-MSR activator fusion protein. We opted for this strategy specifically to induce MSR

expression locally. Importantly MSR RNA induction in this way was sufficient to induce rapid chromocenter reorganisation (within 24h). We have now clarified this point in the manuscript.

As suggested by the reviewer, we have now performed RNA-FISH in ESCs. Our results show that MSR RNA form foci of different sizes, and the foci are most commonly located at the periphery of chromocentres and in some nuclei these foci seem to border multiple chromocenters (Fig. 4F and Supplementary Fig. 4E). Although it is possible that some of the RNA-FISH signal is lost due to the harsh treatments of the RNA-FISH protocol, the results in general indicate that the majority of MSR transcripts are retained close to chromocenters and therefore could be operating in *cis*. Importantly, the RNA-FISH signal is sensitive to RNaseA and RNaseH treatments, and the signal is strongly reduced following MSR-gapmer treatment.

The RNA-FISH images unfortunately do not provide quantitative information about the abundance of MSR transcripts in a cell. This might require single-molecule approaches, but the repetitive sequences of MSR makes this approach technically unfeasible at the moment. We have now highlighted this limitation in the revised Discussion.

4.4 I think that it is impossible to estimate the RNA concentration from the DNA concentration, as the authors imply in their response. The RNA concentration will depend on the transcription rate, the half-life of the transcripts etc. I think it would be very important to measure the MSR RNA concentration in the cell to interpret the in vitro experiments. Furthermore, it would be closer to the cellular situation to study HP1 + DNA/nucleosomes + RNA in vitro (using the conditions in Keenen et al. 2021 + RNA), as chromocenters contain a high concentration of DNA/nucleosomes that affect phase separation.

We agree with the reviewer that measuring the MSR RNA concentration in the cell would be very informative to intercalate in vitro and in vivo data. Unfortunately, due in part to the repetitive nature of MSR transcripts, we cannot think of a technically feasible and reliable method to address this at the moment. We have included the following sentence in the Discussion to mention this point:

Page 10: "A limitation of our work is that the concentration of MSR RNA within cells is currently unknown, and obtaining this information would help to better understand and model how the transcripts might contribute to phase separation processes."

REVIEWER COMMENTS

Reviewer #2 (Remarks to the Author):

The authors did a lot of efforts to bring their imaging approaches to a more standard levels and I think their work has now gained more strength and credibility.

My points of concern - 2.1, 2.3, 2.4, 2.8, 2.9, 2.10 – are now reasonably addressed. I am also glad to see that the authors draw conclusions more cautiously and have removed too speculative part about competing between RNA and DNA for HP1 binding.

I am also happy to learn that the authors perform RNA-FISH, an experiment requested by two reviewers and of a high importance for their work. Unfortunately, the two figures showing RNA signals (Fig 4F and Suppl Fig 4E) are wholly unacceptable for publication. The DAPI channel shows that there are no nuclei left: there are no nuclear borders, no chromocenters and on what the authors base their nuclear outlining is the most enigmatic – compare clear nuclear images on Suppl Fig 4D and dim clouds of something on Suppl Fig 4E. If the shown images are “representative” as indicated in the legend, how all the others look like?! Regrettably, this image does not illustrate size and distribution of RNA signals and I am not convinced that shown signals are not a background.

Reviewer #4 (Remarks to the Author):

In the second revision of this manuscript by Novo et al, the authors have now added RNA-FISH data to show where MSR transcripts are localized. Unfortunately, the authors did not quantify the abundance of MSR transcripts, with the argument that their repetitive nature poses technical problems. However, I believe that it would have been feasible to quantify the MSR content of the cells, either in bulk using dot/slot blots with an appropriate standard, or in single cells using flow cytometry (flow-FISH) or smRNA-FISH. Approaches to estimate the abundance of repetitive sequences from deep sequencing data have also been published (for example for telomeric DNA, should also be feasible for RNA). I am somewhat disappointed that the authors do not provide this information to put their in vitro experiments into context.

The RNA-FISH data themselves look not so impressive compared to what is published in refs 19, 26, 100 and 101. Where does the DAPI signal outside of the nuclei, which are surrounded with the white lines in Fig. 4F, come from? Nevertheless, in conjunction with the four references above, it is clear that MSR transcripts do not co-localize with chromocenters, which is inconsistent with the model that MSR transcripts “scaffold heterochromatin into dynamic condensates”, as the authors write in the Abstract. Such scaffolding would obviously imply an interaction between these molecules, which is only possible if molecules co-localize at steady state (even if molecules/structures are “dynamic”). The authors have rewritten the text to tone down some of the claims and point out some limitations, but they still make the “scaffolding” claim in the Abstract and present a “phase separation” model in their final Fig. 6, which I find misleading as HP1 and MSR transcripts seem to make a “phase” surrounded by nucleosomes in the figure, although HP1 actually co-localizes with the nucleosomes and the DAPI signal while the MSR transcripts localize mostly between chromocenters. I recognize the subtle changes compared to the original version, in which some “MSR RNA clusters” have changed their place, but also in its present form it does not reflect the differential localization of the different components (MSR, HP1, nucleosomes) observed in this paper and other papers.

At the end, I am still left wondering if the effects documented in Fig. 5 are a direct consequence of MSR transcripts being present or not, if MSR transcripts have a function in cis, if they directly affect chromocenter coalescence/separation, and what stabilizes the MSR RNA clusters outside of chromocenters. I unfortunately do not find the answers to these questions in this paper, so I am basically missing the mechanistic insight and the convincing model that would connect the different results in the paper.

Response to reviewers

Reviewer #2:

The authors did a lot of efforts to bring their imaging approaches to a more standard levels and I think their work has now gained more strength and credibility. My points of concern - 2.1, 2.3, 2.4, 2.8, 2.9, 2.10 – are now reasonably addressed. I am also glad to see that the authors draw conclusions more cautiously and have removed too speculative part about competing between RNA and DNA for HP1 binding.

Thank you.

I am also happy to learn that the authors perform RNA-FISH, an experiment requested by two reviewers and of a high importance for their work. Unfortunately, the two figures showing RNA signals (Fig 4F and Suppl Fig 4E) are wholly unacceptable for publication. The DAPI channel shows that there are no nuclei left: there are no nuclear borders, no chromocenters and on what the authors base their nuclear outlining is the most enigmatic – compare clear nuclear images on Suppl Fig 4D and dim clouds of something on Suppl Fig 4E. If the shown images are “representative” as indicated in the legend, how all the others look like?! Regrettably, this image does not illustrate size and distribution of RNA signals and I am not convinced that shown signals are not a background.

We have optimised the RNA-FISH protocol and used a different microscope system, and have obtained better quality images. The DAPI signal is now much clearer and shows the nuclei borders and chromocenters. The new images confirm and improve our previous observations: MSR RNA foci are heterogeneous in size and localisation within nuclei, being observed in chromocenters, at their periphery, and in between adjacent chromocenters. The new images and quantification results are shown in Figs. 4F-G and Supplemental Fig. 4E.

MSR RNA foci are sensitive to RNaseA and RNaseH treatments, and the proportion of nuclei with MSR RNA FISH foci is strongly reduced in cells that have been treated with MSR gapmers (new Figs. 4F-G and Reviewer Fig. 1 below). In addition, this effect is sequence-specific because telomeric RNA foci (TERRA) are not affected (Reviewer Fig. 1). These results validate the MSR RNA FISH signals.

Reviewer Fig. 1: Detection of MSR RNA and telomeric RNA (TERRA) by RNA-FISH in mouse ESC colonies following treatment with either GFP (upper) or MSR (lower) gapmers.

Reviewer #4:

In the second revision of this manuscript by Novo et al, the authors have now added RNA-FISH data to show where MSR transcripts are localized. Unfortunately, the authors did not quantify the abundance of MSR transcripts, with the argument that their repetitive nature poses technical problems. However, I believe that it would have been feasible to quantify the MSR content of the cells, either in bulk using dot/slot blots with an appropriate standard, or in single cells using flow cytometry (flow-FISH) or smRNA-FISH. Approaches to estimate the abundance of repetitive sequences from deep sequencing data have also been published (for example for telomeric DNA, should also be feasible for RNA). I am somewhat disappointed that the authors do not provide this information to put their in vitro experiments into context.

We still cannot see a way to quantify the number of MSR transcripts in a cell. The transcripts vary in terms of their number of unit repeats (234bp each), which means that the signal measured by a probe targeting a repeat sequence will read out a combination of transcript abundance and the number of sequence repeats – we would not be able to derive an accurate estimate of RNA content from these measurements, only relative comparisons. The only way that we can think of would be by long-read Nanopore sequencing which could measure copy

number and repeat length. This is something we have looked into, but it is well known that sequencing non-polyadenylated transcripts (such as major satellite RNA) on the Nanopore is extremely challenging. There are potential methods emerging, at least for short non-polyA RNAs, such as NERD-seq (DOI: 10.1101/2021.05.06.442990) and TERA-Seq (DOI: 10.1093/nar/gkab713), but these technologies are still very much in development. For the time-being, we hope that our new RNA-FISH data (Fig. 4F), together with the RT-qPCR analysis of relative transcript abundance (Figs. 2A and Supplemental Fig. 2A), will suffice in the current manuscript.

The RNA-FISH data themselves look not so impressive compared to what is published in refs 19, 26, 100 and 101. Where does the DAPI signal outside of the nuclei, which are surrounded with the white lines in Fig. 4F, come from? Nevertheless, in conjunction with the four references above, it is clear that MSR transcripts do not co-localize with chromocenters, which is inconsistent with the model that MSR transcripts “scaffold heterochromatin into dynamic condensates”, as the authors write in the Abstract. Such scaffolding would obviously imply an interaction between these molecules, which is only possible if molecules co-localize at steady state (even if molecules/structures are “dynamic”). The authors have rewritten the text to tone down some of the claims and point out some limitations, but they still make the “scaffolding” claim in the Abstract and present a “phase separation” model in their final Fig. 6, which I find misleading as HP1 and MSR transcripts seem to make a “phase” surrounded by nucleosomes in the figure, although HP1 actually co-localizes with the nucleosomes and the DAPI signal while the MSR transcripts localize mostly between chromocenters. I recognize the subtle changes compared to the original version, in which some “MSR RNA clusters” have changed their place, but also in its present form it does not reflect the differential localization of the different components (MSR, HP1, nucleosomes) observed in this paper and other papers.

We have repeated the RNA-FISH experiments with an optimized protocol and used a different microscope system, which has much improved the DAPI staining, allowing us to better localise MSR RNA foci in regards to chromocenters. We detect a variety of MSR RNA foci distributions (new Figs. 4F-G). Although most of the MSR probe signal does localise at the periphery of chromocenters, particularly between adjacent chromocenters, there are also a subset of chromocenters that contain MSR FISH signal in their interior, as shown in new Fig. 4G and Supplemental Fig. 4E. These new data highlight the complexity of the MSR RNA foci formation, which is likely regulated through the cell cycle.

In addition, we believe that RNA-FISH may under represent the signal within chromocenters (perhaps because of insufficient probe penetration/binding, or due to the denaturing nature of

formamide included in the hybridization buffer) because previous sequencing-based data reported that 80% of MSR transcripts were chromatin-associated versus 20% in the nucleoplasm (DOI: 10.7554/eLife.25293) and there must clearly be some RNA transcripts within chromocenters at their sites of transcription (which we include in our model). We believe that the HP1a and MSR RNA localisations are not so black and white, and that there is ample opportunity for HP1a and MSR RNA to interact particularly at the periphery of chromocenters.

We have changed the wording of the abstract so that it no longer says that the RNA is scaffolding the condensates.

At the end, I am still left wondering if the effects documented in Fig. 5 are a direct consequence of MSR transcripts being present or not, if MSR transcripts have a function in cis, if they directly affect chromocenter coalescence/separation, and what stabilizes the MSR RNA clusters outside of chromocenters. I unfortunately do not find the answers to these questions in this paper, so I am basically missing the mechanistic insight and the convincing model that would connect the different results in the paper.

We hope that the findings presented in this manuscript will sufficiently advance the field, and allow further elucidation of the complexity of these mechanisms in the next few years.

REVIEWERS' COMMENTS

Reviewer #2 (Remarks to the Author):

The authors did a good job improving their images and modifying conclusions and I am happy with the implemented changes. I do support the publication in its present state.